# Sox8 is essential for vertebrate gastrulation

Sofia Moreira [1,2], Artemis G Korovesi [3] & Elias H Barriga [1,2 ✉]

## Abstract

**Gastrulation is a fundamental developmental process during which germ layers are formed and the body axes are defined by the precise orchestration of cell movements and fate specification. Here, we identify the SOXE transcription factor Sox8 as a pivotal regulator of *Xenopus laevis* gastrulation. We show that Sox8 is expressed in the ventrolateral mesoderm, and that its depletion—via CRISPR-DiCas7-11—leads to blastopore closure defects and impaired AP axis elongation. Transcriptomic analysis reveals that Sox8 modulates Wnt signalling, in part by directly activating transcription of *kremen2*, a Wnt inhibitor. Indeed, chromatin immunoprecipitation confirms direct binding of Sox8 to the *kremen2* promoter. Consequently, Sox8 or Kremen2 knockdown results in an abnormal ventral expansion of *wnt11b* mRNA that was consistent with increased nuclear β-catenin and reduced BMP signalling. These treatments also led to disruptions in axial and paraxial mesodermal patterning. Together, our data provide new insights into the molecular control of vertebrate gastrulation and invite researchers to assess whether this Sox8/Kremen2 regulatory axis is involved in other biological processes.**

**Keywords** Gastrulation; Ventrolateral Mesoderm; Sox8; Kremen2; *Xenopus laevis*
**Subject Categories** Chromatin, Transcription & Genomics; Development

## Introduction

Gastrulation is one of the earliest and most relevant developmental processes requiring the orchestration of cell movements, divisions and fate acquisition. This high degree of coordination is achieved through the concerted activity of transcriptional activities and signaling pathways, which in turn define the border between the dorsal and ventral territories (De Robertis, 2009; Gao and Zhu, 2012). A key experiment performed 100 years ago by Spemann and Mangold provided an essential platform for understanding amphibian gastrulation (Spemann and Mangold, 1924, 2003). In that experiment, the dorsal lip of the blastopore was grafted onto the opposite side of a host embryo, causing a non-cell-autonomous effect in which this small piece of tissue induced the formation of dorsal structures in otherwise ventral territories. Since this seminal work, a large body of evidence concerning the dorsalizing roles of signaling pathways and transcription factors has been published (Steinbeisser et al, 1995; Piccolo et al, 1996; De Robertis and Wessely, 2004; Sander et al, 2007; Kumar et al, 2021). In brief, the organizer is characterized by strong dorsal activity of the Wnt/β-catenin signaling pathway, which antagonizes the ventralizing activities of the BMP pathway. Although most attention has focused on understanding the gene regulatory networks of the dorsal organizer center, recent progress has also been made in identifying transcription factors with ventralizing activities (Ding et al, 2017a). For example, *ventx1*, a BMP-induced gene that shares homology with Nanog, was shown to play a role as a transcriptional repressor of dorsal gene expression in the ventral gastrula (Gawantka et al, 1995; Sander et al, 2007; Kumar et al, 2022, 2023). Despite these advances, we are still far from fully understanding the convoluted transcriptional networks underlying the formation of ventral territories during gastrulation. Therefore, we investigated transcriptional regulators of the ventral *Xenopus laevis* gastrula, a well-established model of gastrulation. In our RNA sequencing experiments, we focused on SOXE transcription factor 8 (Sox8), as its expression in the ventral territories of the *Xenopus laevis* gastrula was previously detected (O'Donnell et al, 2006; Hong et al, 2008; Ding et al, 2017a, 2017b). However, whether and how Sox8 regulates the establishment of ventral territories and consequently gastrulation remains largely unexplored. By combining RNA-seq experiments with a novel CRISPR Cas7–11 knockdown approach, we discovered that Sox8 is required for gastrulation by allowing the expression of the Wnt inhibitor Kremen2 in ventral regions. This mechanism contributes to reinforcing the ventralizing activity of BMP signaling in ventral tissues by biasing Wnt activity to dorsal tissues.

## Results

### *sox8* is expressed in the ventrolateral mesoderm of *Xenopus laevis* gastrula

To explore the molecular mechanisms involved in the ventral territories, we performed transcriptomic analysis (RNAseq) of isolated ventral tissues at stage 11.5 (Nieuwkoop and Faber, 1994) (Fig. 1A,B). We first validated the quality of our RNA libraries by PCR against the general mesodermal marker *brachyury* (*xbra*) and a specific marker of the ventral mesoderm, *ventx1*. The expression

[1]Mechanisms of Morphogenesis Lab, Cluster of Excellence Physics of Life (PoL), TU Dresden, Dresden, Germany. [2]Mechanisms of Morphogenesis Lab, Gulbenkian Institute of Science (IGC), Oeiras, Portugal. [3]Patterning and Morphogenesis Lab, Gulbenkian Institute for Molecular Medicine (GIMM), Lisbon, Portugal.
✉E-mail: elias.barriga@tu-dresden.de

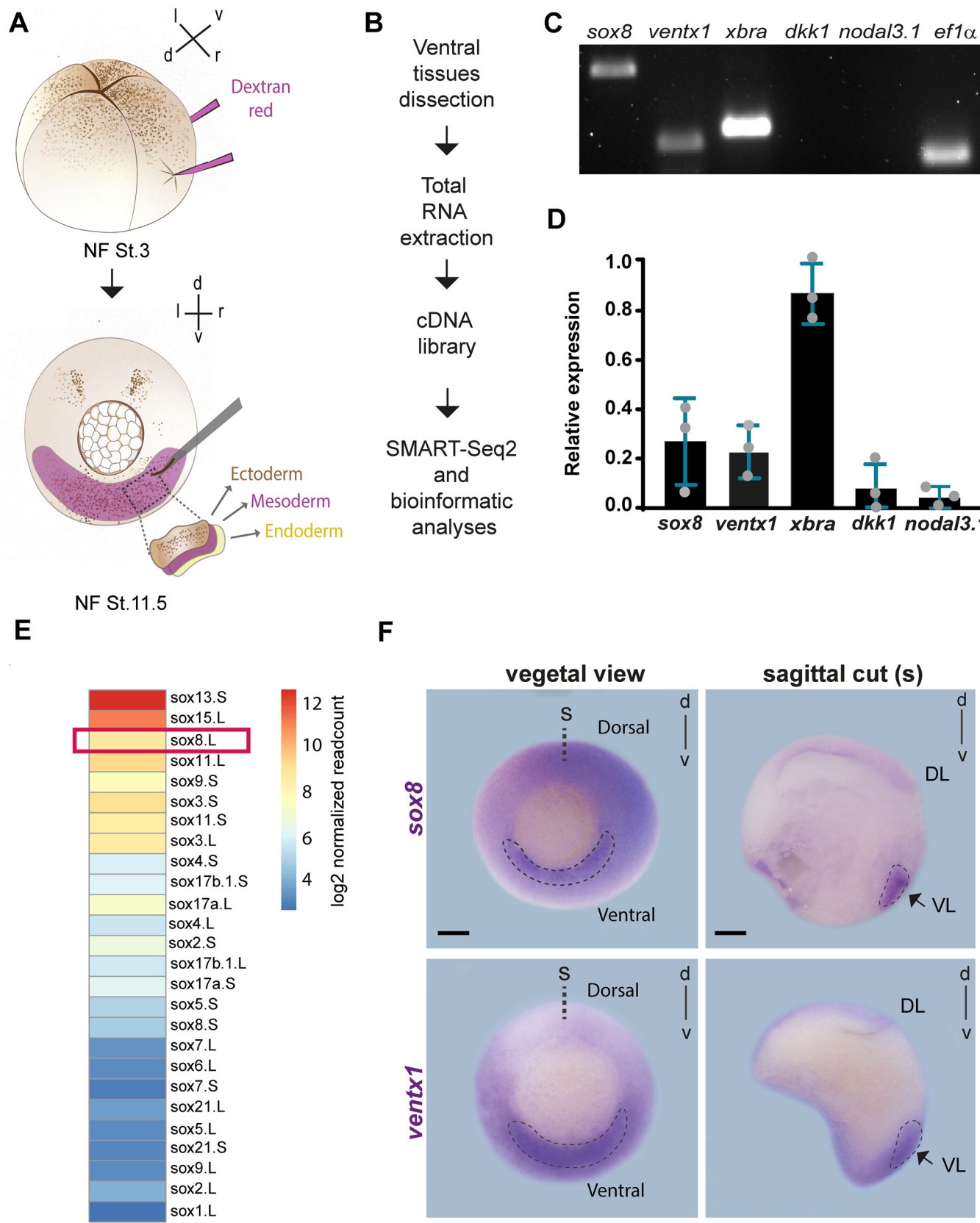

**Figure 1.** *sox8* **is expressed in the ventrolateral mesoderm during gastrulation.**

(A) Schematic representation of targeted ventrolateral mesoderm injections (detected by Dextran Red dye) at four cell-stage (Nieuwkoop and Faber, NF, stage 3), followed by ventral tissue dissection at NF stage 11.5 for RNA-seq. (B) Workflow of the RNA-seq pipeline, including ventral tissue isolation and SMART-seq2 processing. (C) Semi-quantitative RT-PCR (sqRT–PCR) gel image showing the expression of *sox8*, genes related to ventral mesoderm tissues (*xbra* and *ventx1*) and the absence of dorsal markers (*dkk1* and *nodal3.1*). (D) Quantification of relative gene expression normalized to *ef1a* ($n = 3$ three independent biological experiments). Error bars: mean ± SD. (E) Heatmap showing the mean regularized log-transformed read counts of *sox* gene expression from two independent RNA-seq datasets of mesoderm explants. (F) in situ hybridization (ISH) of *sox8* and *ventx1*. A sagittal section is shown to reveal internal tissues and confirm *sox8* expression in the ventrolateral mesoderm (arrows). DL dorsal lip, VL ventral lip, s sagittal. Scale bars: 250 µm. Source data are available online for this figure.

of these markers was enriched in our libraries as compared to the expression of dorsal mesoderm markers such as *dickkopf-1* (*dkk1*) and *nodal3.1*, which were lower in our isolates, confirming enrichment in ventral tissues (Fig. 1C,D). Further RNA-seq and g:Profiler (Raudvere et al, 2019) analyses revealed a list of candidates in which mRNAs encoding transcription factors and other transcriptional regulators were highly enriched (Appendix Fig. S1A). SOX transcription factors, which are related to tissue morphogenesis in multiple species (Okuda et al, 2010; Hong and Saint-Jeannet, 2005; Schock and LaBonne, 2020; Buzzi et al, 2022; Kamachi and Kondoh, 2013; Cizelsky et al, 2013; McGary et al, 2010), were highly represented in our RNAseq analysis (Fig. 1E). Therefore, we next generated a list of the five most highly expressed SOX transcription factors in ventral tissues (Appendix Fig. S1B). The expression pattern of these SOX transcription factors is ubiquitous in gastrulas, with some of them being expressed mostly in dorsal tissues (Rorick et al, 2007; de Almeida et al, 2008; Satow et al, 2006; Nitta et al, 2004; Marchak et al, 2017). In contrast, the expression of the SOXE transcription factor Sox8 was previously reported in the ventrolateral domain of *Xenopus laevis* gastrula (O'Donnell et al, 2006; Hong et al, 2008; Ding et al, 2017b, 2017a) (Appendix Fig. S1B). Nevertheless, whether and how Sox8 participates in gastrulation remains largely unknown. Therefore, we next focused on the role of Sox8 in gastrulation. We first performed whole-mount in situ hybridization against *sox8* at stage 11.5, followed by tissue sectioning. Our in situ hybridizations revealed that *sox8* is specifically expressed in the ventrolateral mesoderm, as shown by comparison with the expression of *ventx1*, a well-known ventrolateral mesoderm marker (Ault et al, 1996) (Fig. 1F). We also confirmed the ventral expression of *sox8* by PCR analyses of isolated ventral mesoderm libraries (Fig. 1C,D). These results suggest that Sox8 plays a role in gastrulation by operating in the ventrolateral mesoderm.

## Sox8 is required for *Xenopus laevis* gastrulation

We next assessed the requirement of Sox8 during gastrulation by using an RNA-targeting knockdown tool described as CRISPR-DiCas7-11 (Özcan et al, 2021; Moreno-Sánchez et al, 2025) and hereafter referred to as *sox8* CRISPR 7–11 (Fig. 2A). Further details about the implementation of the CRISPR 7–11 methodology in *Xenopus laevis* can be found in the Methods section (Appendix Figs. S4–S7). We first determined that *sox8* CRISPR 7–11 injection targets the ventral gastrula (Fig. 2B,C) and that it effectively decreases *sox8* mRNA expression in that territory, as observed by in situ hybridization (Fig. 2D). Morphologically, the inhibition of *sox8* in the ventral mesoderm elicited an evident delay in the closure of the blastopore (Figs. 2E–G and EV1A,B). These defects

were rescued by co-injection of *sox8* mRNA along with *sox8* CRISPR 7–11 (Fig. 2E–G), confirming the specificity of our results. These observations were complemented by the use of morpholino knockdown, which was previously used to assess the role of Sox8 in neural crest induction (O'Donnell et al, 2006). By targeting this morpholino to the ventral mesoderm, we recapitulated the blastopore closure defects observed by using CRISPR 7–11 knockdown (Fig. EV2A–C). Conversely, the injection of *sox8* CRISPR 7–11 knockdown into the dorsal territories affected *sox8* expression in the neural crest (Fig. EV2D–H). Together, our results indicate the requirement of Sox8 in ventral territories to ensure correct gastrulation and validated CRISPR 7–11 as an effective new mRNA-knockdown tool in *Xenopus laevis*. Blastopore closure defects have been previously associated with defects in cell movements, which are required for axis elongation (Cervino et al, 2021). Indeed, when analysed at later tailbud stages, we observed that embryos exhibited open blastopores, and a significant percentage of embryos appeared shortened (Fig. EV1C–E), suggesting defects in anterior-posterior axis elongation. Thus, to further characterize the observed effects on blastopore closure, we performed live imaging of control and *sox8* CRISPR embryos. Trajectory analyses of involuting ventral cell movements (Fig. 3B–D; Methods) revealed that while the velocity of migrating cells was slightly affected, the persistence of cell movement was drastically reduced in *sox8* CRISPR-treated embryos (Fig. 3C,D). Consistently, sagittal sections revealed increased accumulation of ventral tissues in *sox8* knockdown-treated embryos compared with control embryos (Fig. 3A). In addition, in situ hybridization revealed that the axial and paraxial mesoderm markers *chordin*, *brachyury*, and *myoD* are affected by Sox8 loss-of-function (Fig. 3E–M). These observations indicate that Sox8 contributes to gastrulation by supporting correct blastopore closure and associated axis elongation.

## Transcriptomic analysis reveals that Sox8 influences ventral markers and Wnt inhibitors

To gain molecular insights into how Sox8 regulates gastrulation, we performed RNA-seq analysis of ventral tissue from control and *sox8* CRISPR 7–11-treated embryos (Fig. 4A). The quality of our RNA libraries was determined by PCR analyses of *xbra*, a pan mesodermal marker; *msfGFP* from Cas7-11, for correct targeting; and *sox8*, to confirm the efficiency of its depletion (Fig. EV3A). Principal component analysis (PCA) was also performed to confirm the overall sample variability (Appendix Fig. S2). A list of 7273 upregulated and 6933 downregulated candidates was generated and further filtered by grouping these candidates on the basis of the biological processes and signaling pathways in which they are

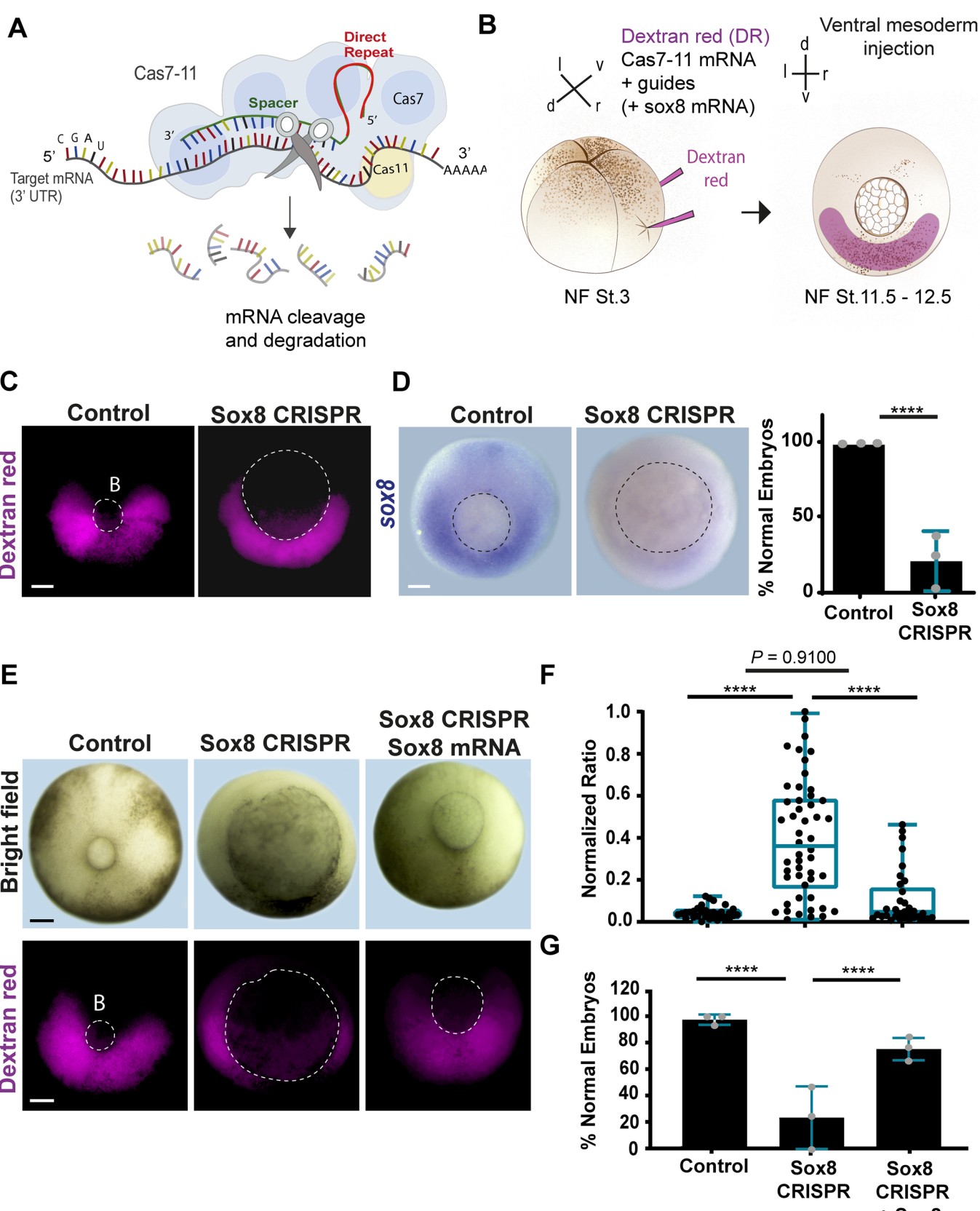

**Figure 2. CRISPR-mediated knockdown of *sox8* in the ventrolateral mesoderm induces gastrulation defects.**

(A) Schematic of the CRISPR Cas7–11 technique, highlighting the endonuclease Cas7–11, the target mRNA and guide binding to the 3′UTR mRNA through a complementary spacer. For simplicity, we represent just one guide, although three guides targeting distinct regions of the 3′ UTR were used in the injection mixture. (B) Schematic of targeted injections at the four-cell stage (NF stage 3) specifically directed to the ventrolateral mesoderm. (C) Injection control showing successful ventrolateral targeting of Dextran Red dye (in magenta); B blastopore. (D) Percentage of embryos displaying a normal *ISH* signal of *sox8*. Statistical significance was assessed by two-sided Fisher's exact test, ****$P < 0.0001$. Error bars: mean ± SD; $N = 3$ independent biological experiments ($n = 9$ control embryos; $n = 22$ *sox8* CRISPR embryos). (E) Representative phenotypes for the control, *sox8* CRISPR Cas7–11 and rescue (co-injection of *sox8* mRNA). B blastopore. (F) Box plot showing the normalized ratio of blastopore area to embryo area under the indicated conditions. Boxes represent the interquartile range (IQR; 25th–75th percentile), with the center line indicating the median and whiskers extending to the minimum and maximum values. Statistical significance was assessed by Kruskal–Wallis multiple comparison test, ****$P < 0.0001$; ns non-significant: $P = 0.9100$. (G) Percentage of embryos exhibiting normal blastopore size per condition (embryos displaying values below the maximum control ratio value are classified as normal). Two-sided Fisher's exact test, ****$P < 0.0001$; Error bars: mean ± SD. In (C–E), dotted lines delineate the blastopore. For panels (F, G), $N = 3$ independent biological experiments ($n = 32$ control embryos (injected with DR and Cas7–11); $n = 50$ *sox8* CRISPR embryos; $n = 33$ *sox8* CRISPR + *sox8* mRNA embryos). Scale bars: 250 μm. Source data are available online for this figure.

involved. For this purpose, we performed functional enrichment analysis in g:Profiler, including Gene Ontology (GO) biological process and Kyoto Encyclopedia of Genes and Genomes (KEGG) pathway analyses. As observed in the Manhattan plot for "Biological Processes" (Fig. 4B), significant results were obtained for GO terms related to cytoskeleton organization, cell migration, and axial pattern formation (among others) (Fig. 4C). These findings align with earlier observations of phenotypes related to morphogenetic movements and axis elongation in *sox8* CRISPR (Figs. 3 and EV1). KEGG pathway analysis revealed overrepresented terms such as "Wnt signaling pathway" and "TGF-β signaling pathway" (in which BMP signaling is included) (Fig. 4D,E). Gene expression analyses revealed that the expression of ventral genes, such as *ventx*, *bmp* and *kremen2*, was downregulated in our libraries (Fig. 4F). On the other hand, dorsal genes such as *wnt11b* and *frzd* as well as dorsal organizer genes, such as *gsc*, *not*, and *chrd*, are upregulated in ventral tissues (Fig. 4F). This strongly suggests an expansion of Wnt activity to ventral territories. Consequently, we performed in situ hybridization against *wnt11b* —a ligand whose zygotic expression is found in dorsal territories after the onset of gastrulation (Castro Colabianchi et al, 2024; Tada and Smith, 2000; Smith, 2001). We observed that *wnt11b* expression domain expanded to ventral tissues in *sox8*-knockdown embryos (Fig. 4G). This ventral expansion was consistent with an increase in nuclear β-catenin, a readout of Wnt activity, which is normally increased in dorsal tissues (Fagotto and Brown, 2009; Schohl and Fagotto, 2002) (Fig. 4I,J). Similarly, this ventral expansion of Wnt activity was accompanied by an inhibition of ventralizing BMP signaling, as observed by a decrease in pSmad protein (Fig. 4K,L). These results indicate that Sox8 may control gastrulation by modulating the activity of Wnt and BMP in ventral tissues.

## Sox8 directly regulates *kremen2* transcription

The expansion of Wnt activity and, in turn, *wnt11b* expression into the ventral territories is consistent with the downregulation of well-known inhibitors of the Wnt signaling pathway, such as *kremen2*, which is typically expressed in the ventrolateral region of control embryos (Hassler et al, 2007; Rothbächer and Lemaire, 2002) (Fig. 5A). Thus, we hypothesize that Sox8 might regulate gastrulation, at least in part, by repressing Wnt activity in the ventral domain. Since the interaction between Sox8 and Kremen2 has not been previously described, we first focused on the regulation of this Wnt

inhibitor by *sox8*. We first validated the requirement of Sox8 for *kremen2* expression by in situ hybridization and PCR (Figs. 5A,B and EV3B). In light of this regulation, we next investigated whether Sox8 regulates *kremen2* expression through direct binding to its promoter or in a non-direct manner. To address this, we performed chromatin immunoprecipitation (ChIP) against Sox8-GFP from gastrulating embryos. We used in silico approaches to identify potential binding sites of Sox8 in the regulatory elements of *kremen2* and identified three putative binding sites in the proximal promoter region of *kremen2* (Fig. 5C, Methods). PCR analyses revealed that Sox8-GFP was significantly enriched in one of these regions compared with that in samples obtained from control embryos (Fig. 5D–F; Appendix Fig. S3A). Moreover, this increase was not observed for another putative binding site (Appendix Fig. S3B), suggesting specific binding. Next, we evaluated whether the recruitment of Sox8 to this proximal binding site is associated with the recruitment of the transcriptional machinery. Indeed, we detected Pol II enrichment in control ChIP samples, but PoL II recruitment was decreased in samples obtained from *sox8*-depleted embryos (Fig. 5D,G,H; Appendix Fig. S3C). Together, these results indicate that Sox8 directly regulates *kremen2* transcription in the ventral regions of the *Xenopus* gastrula.

## *kremen2* controls gastrulation by restricting Wnt activity to dorsal tissues

Next, we assessed the requirement of *kremen2* for both gastrulation and *wnt11b* expression by targeting *kremen2* CRISPR 7–11 to ventral tissues (Fig. 6; depletion controls in Fig. EV3B). Morphological analysis revealed that *kremen2* knockdown recapitulates the blastopore closure defects observed upon *sox8* depletion (Fig. 6A–C), which are defects that persist at a later stage (Fig. EV4). In addition, we also found that *kremen2* inhibition affected the localization of mesoderm markers such as *myoD*, *chordin* and *brachyury* (Fig. EV5A–I), suggesting a requirement for axial patterning. Indeed, double-knockdown experiments for *sox8* and *kremen2* elicited a stronger effect (Fig. EV5A–I), which makes sense with drastic, but not full, depletion of *kremen2* expression in *sox8* knockdown (Fig. 5A,B). Furthermore, in situ hybridization analyses revealed that, as observed with *sox8* CRISPR 7–11, *wnt11b* localization also expanded to ventral tissues upon *kremen2* knockdown (Figs. 6D and EV3E–H for further details). Blastopore closure and the wnt11b expression pattern were rescued by co-

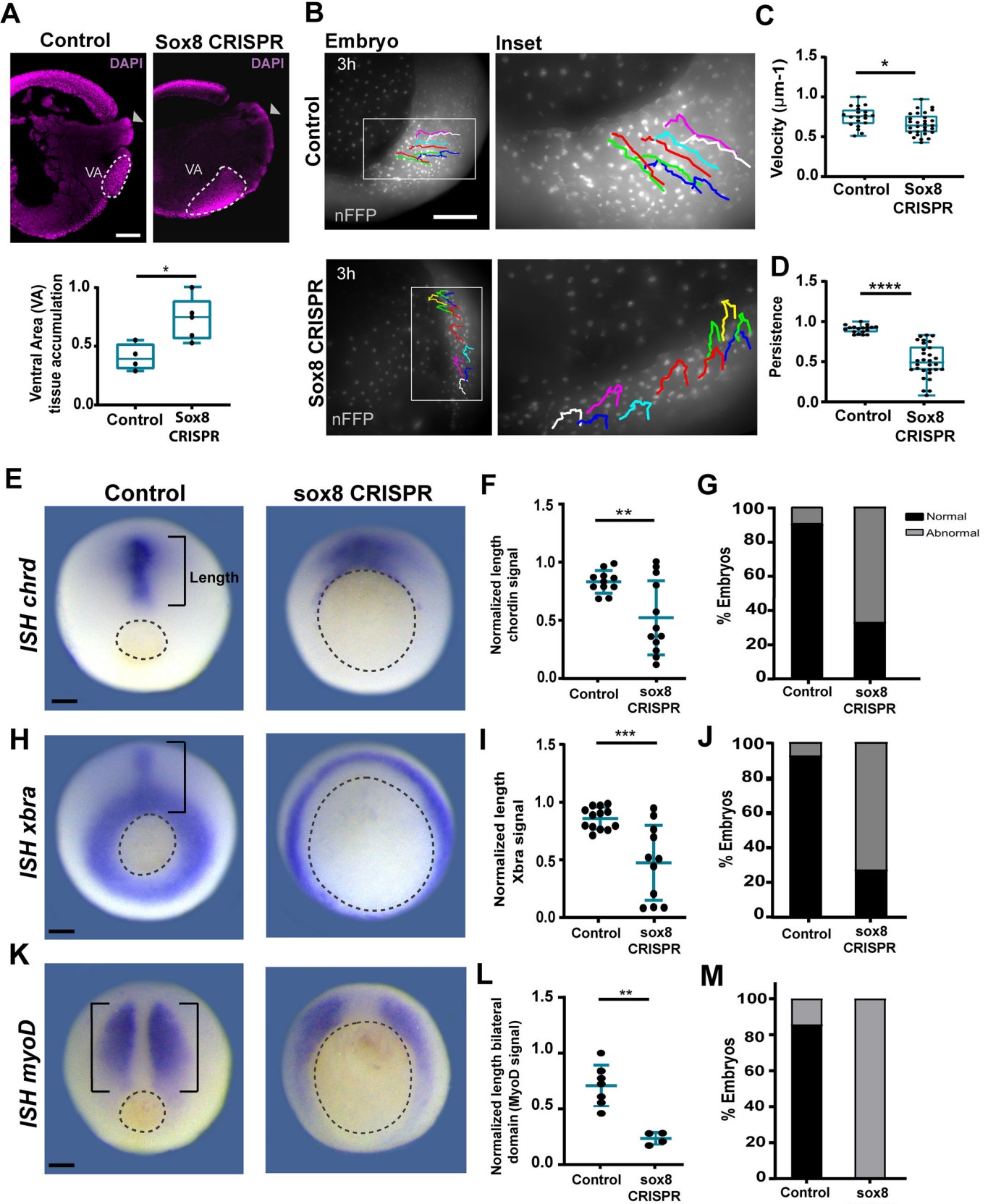

**Figure 3.  sox8 CRISPR disrupts internal tissue organization and ventral cell movement during gastrulation.**

(A) Sagittal histological sections of control and sox8 CRISPR gastrulae show impaired internalization of the yolk plug (arrowhead) and abnormal accumulation of ventral tissues in sox8 CRISPR conditions. Box plot showing the quantification of ventral tissue area revealed a significant increase in sox8 CRISPR gastrulae compared with controls. Statistical significance was assessed by Mann–Whitney test, *$P = 0.0317$. $n = 4$ embryos per group (control and sox8 CRISPR). (B–D) Tracking of superficial involuting cell movements on the ventral side using nuclear RFP. (C) Box plot showing cell velocity. Two-tailed t-test, *$P = 0.0219$. (D) Box plot showing migration persistence in sox8 CRISPR embryos compared with controls. Two-tailed *t*-test, ****$P < 0.0001$. For (A, C, D), boxes represent the interquartile range (IQR; 25th–75th percentile), with the center line indicating the median and whiskers extending to the minimum and maximum values. For (C, D), $n = 20$ cells (control) and $n = 30$ cells (sox8 CRISPR). (E, H, K) Whole-mount in situ hybridization (ISH) for chordin (chrd) (E), brachyury (xbra) (H), and myogenic differentiation 1 (myoD) (K) shows abnormal expression patterns in sox8 CRISPR knockdown embryos compared with controls. (F, I, L) Scatter dot plot showing the quantification of axial mesoderm expression length reveals significant reductions under CRISPR conditions. Error bars: mean ± SD. (F) Two-tailed t-test, **$P = 0.0057$. (I) Two-tailed t-test, ***$P = 0.0005$. (L) Two-tailed Mann–Whitney test, **$P = 0.0061$. (G, J, M) Percentages of normal and abnormal embryos in each condition (embryos displaying higher values than the baseline control length value are classified as normal). Sample sizes: For ISH, chordin: $n = 10$ control; $n = 12$ sox8 CRISPR; for Xbra ISH: $n = 13$ control; $n = 11$ sox8 CRISPR; for ISH, MyoD: $n = 7$ control; $n = 4$ sox8 CRISPR. Scale bars: 250 µm. Source data are available online for this figure.

injection of *kremen2* mRNA with *kremen2* CRISPR 7–11 (Fig. 6A–D), confirming the specificity of the observed effects. To corroborate whether Wnt expansion towards the ventral territories was functional, we performed immunostaining for β-catenin. We detected greater accumulation of nuclear signals in the ventral tissue of *kremen2* CRISPR-treated embryos than in control embryos, indicating an increase in Wnt signaling (Fig. 6E,F). In our analyses, we also observed effects on the transcription of BMP components in *sox8* CRISPR embryos, as revealed by RNA-seq and ISH (Figs. 4F and EV6A,B). Additionally, pSmad protein expression, a readout of BMP signaling, was reduced upon *sox8* and *kremen2* knockdowns (Figs. 4K,L and EV6C,D). These observations raised the possibility that the observed defects are a consequence of BMP signaling defects. However, co-injection of *bmp* mRNA was not sufficient to rescue the blastopore defects caused by *sox8* CRISPR (Figs. 6H,J and EV3C,D). On the other hand, the co-injection of *sox8* CRISPR 7–11 alongside *kremen2* mRNAs was sufficient to rescue both blastopore closure and the *wnt11b* expression pattern (Figs. 6G–I,M–O and EV3C–H). In contrast, *sox8* mRNA co-injection failed to rescue the defects observed in *kremen2* CRISPR 7–11 treated embryos (Figs. 6K,L,P,Q and EV3C–H). These findings confirm that *kremen2* works downstream of *sox8* and strongly suggest that BMP downregulation may not be the primary cause of the observed defects upon Sox8 knockdown but rather a secondary effect of Wnt signaling modulation via Kremen2.

Taken together, our results indicate that Sox8 contributes to gastrulation by directly regulating the expression of the Wnt inhibitor Kremen2, thereby limiting the expansion of Wnt-related genes, such as *wnt11b*, into ventral territories.

## Discussion

Most SOX transcription factors have been reported to be involved mainly in cell pluripotency and neural crest development in *Xenopus* (Schock and LaBonne, 2020), but less is known about their role in gastrulation. The Sox F-group transcription factor Xsox17 was reported to control gastrulation in *Xenopus* by defining the endodermal territory and inhibiting Wnt signaling by repressing the transcriptional activity of β-catenin (Howard et al, 2007; Zorn et al, 1999). In zebrafish, *sox8* is not expressed during gastrulation (https://zfin.org), but the B1 SOX transcription factor SOX2/3/19

reportedly regulates gastrulation by controlling BMP-related genes and Wnt to coordinate cell fate specification with patterning and morphogenesis (Okuda et al, 2010). In chick and mouse embryos, SOX3 was reported to play a role in gastrulation by defining embryonic territories (Acloque et al, 2011). These findings collectively support a conserved role for SOX transcription factors in morphogenesis.

Here, we demonstrate that direct transcriptional regulation of *kremen2* by sox8 is required for repressing Wnt activity in ventral territories, thereby ensuring proper gastrulation and anterior-posterior (AP) patterning. Since Kremen2 is involved in the Wnt/β-catenin (canonical) pathway (Rothbächer and Lemaire, 2002), we expected that downregulation of Sox8 or Kremen2 would lead to increased nuclear β-catenin in ventral tissues. Such an increase was indeed observed in our conditions. Nevertheless, we cannot exclude the contribution of non-canonical Wnt signaling. In fact, involvement of non-canonical Wnt pathways could account for fluctuations observed in the localization of different gastrulation markers upon double Sox8/Kremen2 knockdowns or upon individual Kremen2 and Sox8 knockdowns. This possibility warrants further investigation.

Beyond gastrulation, AP patterning is also required in other contexts, such as in neural crest and limb development, but whether Sox8 regulates *kremen2* in these contexts remains to be determined. Both Kremen2 and Sox8 are involved in limb development and osteogenesis in mice (Schmidt et al, 2005; Ellwanger et al, 2008; Schulze et al, 2010; González Alvarado and Aprato, 2025), as well as in neural crest development in *Xenopus* (O'Donnell et al, 2006; Hassler et al, 2007) but whether they operate through the same regulatory axis remains an open question. In addition, investigating whether Sox8 also mediates *kremen2* expression in these contexts would contribute to exploring the extent to which this regulatory axis can be generalized. In addition to their role in embryonic development, Wnt and SOX interactions are associated with many types of cancer (Kormish et al, 2010); interestingly, Kremen2 has also been reported to be involved in gastric, colon and lung cancers (Chen et al, 2021; Sun et al, 2023; Long et al, 2023). Thus, our finding that Sox8 regulates Kremen2 expression could expand our understanding of pathological scenarios in which this regulatory axis might be involved.

In our data, we additionally observed that *sox8* downregulation also affects BMP expression. One possibility is that the observed

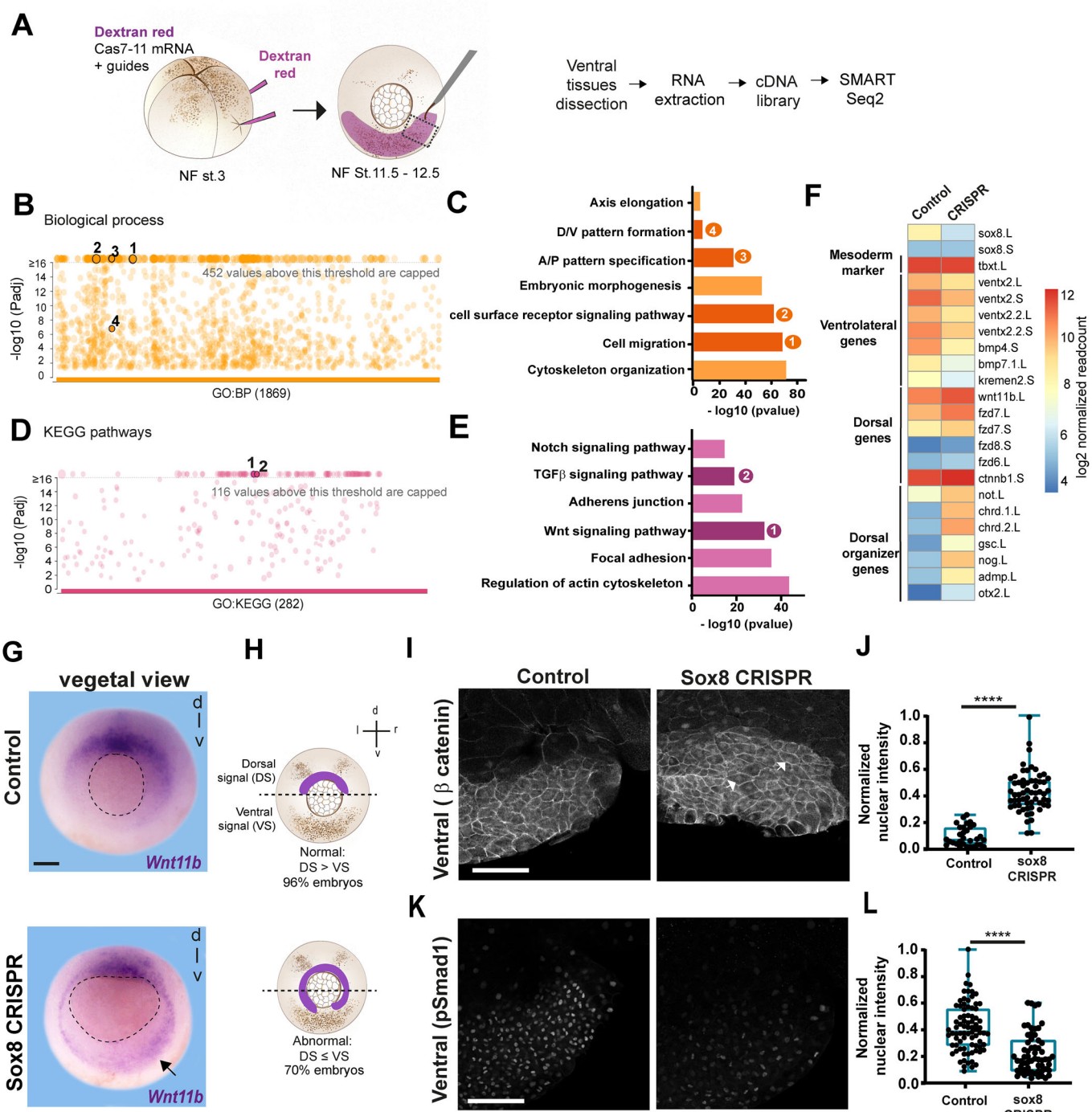

effect could be due to a parallel role of BMP. However, injection of *bmp* mRNA was not able to rescue the effects of *sox8* loss-of-function. Accordingly, depletion of ventral markers such as *bmp* or *ventx1* often leads to a dorsalisation of embryos at later stages, which is characterized by the lack of ventral features (Sander et al, 2007; Reversade et al, 2005). In contrast, *sox8* and *kremen2* depletion resulted in marked failure of blastopore closure and defects in axis elongation, consistent with defective convergence movements and altered patterning of axial and mesodermal markers. These results suggest that the role of *sox8* in gastrulation is not related to the establishment of the ventral structure itself, but rather to AP axis patterning. Previous studies suggest that gastrulation and AP axis patterning are linked, sharing common mechanisms such as convergent extension movements (Shook et al, 2018). On the other hand, blastopore closure can also occur independently of AP axis patterning in *Xenopus* (Nutt et al, 2001; Shook et al, 2022). Nonetheless, further research is necessary to distinguish between these possibilities.

**Figure 4. Transcriptomic analysis following *sox8* knockdown reveals the involvement of the Wnt and BMP pathways.**

(A) Schematic representation of targeted ventrolateral mesoderm injections and RNA-seq workflow. (B) Manhattan plot generated by g:Profiler showing functional enrichment of gene ontology (GO) BP (biological process) terms from up- and downregulated genes. (C) Bar chart of selected enriched GO BP (biological process) terms with corresponding numbers in the Manhattan plot (B): (1) cell migration (GO:0016477); (2) "cell surface receptor signaling pathways (GO:0007166); (3) anterior/posterior pattern specification (GO:0009952), and (4) dorsal/ventral pattern formation (GO:0009953). (D) Manhattan plot generated by g:Profiler showing functional enrichment using gene ontology KEGG analysis from a list of up- and down-regulated genes. (E) Bar chart of selected KEGG (Kyoto Encyclopedia of Genes and Genomes) terms highlighting significant enrichment of (1) "Wnt signaling pathway" and (2) "TGF-β signaling pathway". (F) Heatmap showing the mean regularized log-transformed read counts for genes identified via RNA-seq analysis, both in the control and *sox8* CRISPR samples. (G) In situ hybridization of *wnt11b* showing expanded ventrolateral expression in *sox8* CRISPR embryos (arrow). Dotted lines delineate the blastopore. (H) Schematic illustration of the predominant *wnt11b* expression phenotypes in each condition. Dashed lines indicate the embryo's equatorial region, dividing the dorsal (above) and ventral (below) regions. The shaded purple area denotes *wnt11b* expression. See EV3E, F for quantifications. (I) β-catenin immunofluorescence showing an increased nuclear β-catenin signal (arrows) on the ventral side of *sox8* CRISPR embryos compared with controls. Scale bar: 100 μm. (J) Box plot showing the normalized nuclear β-catenin intensity in ventral cells. Statistical significance was assessed by a two-tailed Mann–Whitney test, ****$P < 0.0001$. $n = 4$ control embryos (31 cells); $n = 8$ sox8 CRISPR (60 cells). (K) pSmad staining levels are decreased in *sox8* CRISPR conditions. (L) Box plot showing the quantification of nuclear pSmad signal in ventral regions; Statistical significance was assessed by two-tailed Mann–Whitney test, ****$P < 0.0001$. $n = 9$ control embryos (80 cells); $n = 6$ sox8 CRISPR (60 cells). Scale bar: 250 μm. For box plots in (J, L), boxes represent the interquartile range (IQR; 25th–75th percentile), with the center line indicating the median and whiskers extending to the minimum and maximum values. Source data are available online for this figure.

Other intriguing questions are related to the upstream signals that promote and restrict Sox8 expression to ventrolateral regions of the mesoderm. Notch1, for example, has been demonstrated to be the earliest sign of ventral development in vertebrates, positively controlling, directly or indirectly, the expression of genes from the ventral program (Acosta et al, 2011; Castro Colabianchi et al, 2018; López, 2022). Investigating whether Notch1 affects *sox8* expression and localization in the ventrolateral mesoderm, thereby influencing the expression of other ventral genes, is important for future research. Additionally, Notch was reported to antagonize Wnt signaling by destabilizing transcriptionally active β-catenin on the ventral side of the early embryo (Acosta et al, 2011). Thus, the regulation of Sox8 by Notch could also represent an additional regulatory layer through which Notch inhibits Wnt activity in the ventral territory. In addition to Notch, several other mechanisms can also regulate *sox8* localization. A role for Wnt signaling in mediating the transcriptional regulation of *sox8* through epigenetic modification of its promoter has also been described in spermatogonial cells (Kayyar et al, 2022). Hif1α was also demonstrated to regulate Sox9 expression in early skeletogenesis (Amarilio et al, 2007), but whether similar mechanisms could act on Sox8 during *Xenopus* gastrulation remains to be shown. Additionally, recent studies have shown that mechanical forces influence Sox9 expression during craniofacial development in zebrafish (Subramanian et al, 2023). Given the relevance of cell mechanics in gastrulation (Keller, 1981; Wilson and Keller, 1991; Shook et al, 2018) and the role of mechanics in controlling gene transcription (Dupont and Wickström, 2022; Miroshnikova et al, 2017), investigating whether mechanical forces during gastrulation could also play a role in restricting *sox8* spatial expression in the ventrolateral mesoderm would be interesting. Later in development, SoxE factors are involved in neural crest development, a process that is highly influenced by mechanical signals (Barriga et al, 2018; Marchant et al, 2022). This illustrates the potential broad relevance of studying the mechanical control of *sox8* expression and its related *kremen2* regulation in other biological contexts.

Finally, our study highlights Sox8 as a key factor that contributes to gastrulation by allowing the expression of *kremen2* in ventral tissues and biasing Wnt activity to dorsal territories. These results provide new insights into the mechanisms of gastrulation, a process that is central across the animal kingdom.

# Methods

### Reagents and tools table

| Reagent/resource | Reference or source | Identifier or catalog number |
|---|---|---|
| **Experimental models** | | |
| *Xenopus laevis* (wildtype) | Xenopus1 (https://xenopus1.com) | N/A |
| **Recombinant DNA** | | |
| pCS2⁺ nRFP | Barriga et al, 2018 | N/A |
| pDF0159 pCMV-huDisCas7-11 | Addgene | Cat# 172507 |
| pCS2⁺ msfGFP | This study | N/A |
| pCS2⁺ huDisCas7-11-msfGFP | This study | N/A |
| pCS2⁺8CmCherry | Barriga et al, 2018 | N/A |
| pCS2⁺ sox8 | Provided by Saint-Jeannet lab (published in O'Donnell et al, 2006) | N/A |
| pCS2⁺ kremen2S | This study | N/A |
| pCS2⁺ bmp7.1 L | This study | N/A |
| pCS2⁺8CeGFP | Addgene | Cat# #34952 |
| pCS2⁺ Sox8-GFP | This study | N/A |
| Plasmid for Snail2/Slug | Provided by Eric Thevenau lab (published in Mayor et al, 1995) | N/A |
| pBSK-ventx1 | Provided by Philipp Vick Laboratory | N/A |
| pCS2⁺ Chordin | Provided by Saint-Jeannet lab | N/A |
| pBSK-MyoD | Provided by Saint-Jeannet lab | N/A |
| pSP73-Xbra | Provided by Saint-Jeannet lab | N/A |
| BMP7-CS110K | European Xenopus Resource Center (EXRC) | HAR310 |
| **Antibodies** | | |
| pSmad1/5/8 | Millipore | Cat# AB3848-I |

| Reagent/resource | Reference or source | Identifier or catalog number |
|---|---|---|
| β-CAT active | Sigma-Aldrich | Cat# 05-665 |
| Anti-mouse Alexa Fluor 647 | Invitrogen | Cat# A32728 |
| Anti-rabbit Alexa Fluor 647 | Invitrogen | Cat# A31573 |
| Anti-Pol II | Diagenode | Cat #C15200004 |
| Anti-GFP | Abcam | Cat # ab6556 |
| **Oligonucleotides and other sequence-based reagents** | | |
| Sox8 MO | GeneTools (O'Donnell et al, 2006) | N/A |
| PCR primers | This study or others | Table EV1 |
| gRNAs | This study | Table EV2 |
| **Chemicals, enzymes and other reagents** | | |
| Human chorionic gonadotrophin (hCG) | MSD Animal Health, Chorulon | 5x1500UI |
| Ficoll | Sigma | Cat# P7798 |
| Dextran Texas Red | Life Technologies | Cat# D1863 |
| mMESSAGE mMACHINE SP6 | Thermo Fisher | Cat# AM1340 |
| mMESSAGE mMACHINE T7 | Thermo Fisher | Cat# AM1344 |
| Gibson Assembly® Master Mix | New England Biolabs (NEB) | Cat# E2611 |
| Q5® Hot Start High-Fidelity DNA Polymerase | New England Biolabs (NEB) | Cat# M0493 |
| RNeasy kit | Qiagen | Cat# 74104 |
| Monarch RNA Cleanup Kit | New England Biolabs (NEB) | Cat# T2040L |
| MinElute Gel Extraction Kit | Qiagen | Cat# 28604 |
| Promega Riboprobe® SP6 | Promega | Cat# P1085 |
| Promega Riboprobe® T7 | Promega | Cat# P2075 |
| Qubit RNA BR (Broad-Range) assay kit | Invitrogen | Cat# Q10210 |
| DIG RNA labeling mix | Roche | Cat# 11277073910 |
| Proteinase K | GRiSP | Cat# GE010.0100 |
| SuperScript IV | Invitrogen | Cat# 18090050 |
| Oligo (dT) Primer 50 μM | Invitrogen | Cat# AM5730G |
| Dynabeads™ Protein G | Thermo Fisher Scientific | Cat# 10004D |
| M-280 Dynabeads | Thermo Fisher Scientific | Cat# 11203D |
| Triethylamine | Millipore | Cat# T0886 |
| Acetic anhydride | Millipore | Cat# A6404 |

| Reagent/resource | Reference or source | Identifier or catalog number |
|---|---|---|
| Benzyl benzoate | Sigma | Cat# B6630-250ML |
| Benzyl alcohol | Sigma | 8.22259.0100 |
| BCIP | Roche | 11383221001 |
| NBT | Roche | 11383213001 |
| Low-melting agarose | Sigma | A9414-5G |
| **Software** | | |
| Leica's software | Leica, IM50 | Leica, IM50 |
| Graph Pad Prism 10.2.2 | GraphPad Software (La Jolla, CA, USA) | N/A |
| FIJI | Schindelin et al, 2012 | N/A |
| SnapGene® | Dotmatics | |
| **Other** | | |
| NextSeq500 Sequencer | Illumina | N/A |
| Stereo LUMAR stereoscope | Zeiss | N/A |
| Leica Stellaris 5 | Leica | N/A |
| Thunder Imager 3D | Leica | N/A |

## Methods and protocols

The procedures related to animal handling described in this work were performed according to the Ethics Committee and Animal Welfare Body (ORBEA) of the Instituto Gulbenkian de Ciência (IGC) and complied with Portuguese (Decreto-Lei n° 113/2013) and European (Directive 2010/63/EU) legislation.

### *Xenopus laevis manipulation and embryo collection*
Adult frogs were maintained in a temperature- and light-controlled environment (18 °C and 12 h light/12 h obscurity). Embryos were obtained by in vitro fertilization using a sperm mixture containing 500 μl of Marc's modified Ringer 0.1x medium (MMR: 10 mM NaCl, 0.2 mM CaCl₂·2H₂O, 0.2 mM KCl, 0.1 mM MgCl₂·6H₂O, and 0.5 mM HEPES, pH 7.1–7.2). Ovulation was induced by injecting human chorionic gonadotropin (MSD Animal Health, Chorulon). After fertilization, the embryos were maintained at 0.1x MMR at 12 or 14 °C. Subsequent embryo staging was performed following previously established developmental methods (Nieuwkoop and Faber, 1994).

### *Microinjection, cloning, and synthesis of MO and mRNA*
Pulled glass needles were calibrated and mounted in a cell microinjector (MDI, PM1000), which was programmed to inject 10 nl on a gas pulse of 20 psi for 0.2 s. After 5 min of de-jelly using L-cysteine (1 g in 50 ml 0.1x MMR with 500 μl NaOH 5 M), with consequent rinses in 0.1x MMR, the embryos were transferred to 5% Ficoll (Sigma, P7798)/0.45x MMR (w/v). The neural crest was targeted by injecting dorsal and ventral animal blastomeres at the eight-cell stage; the ventrolateral mesoderm was injected at the

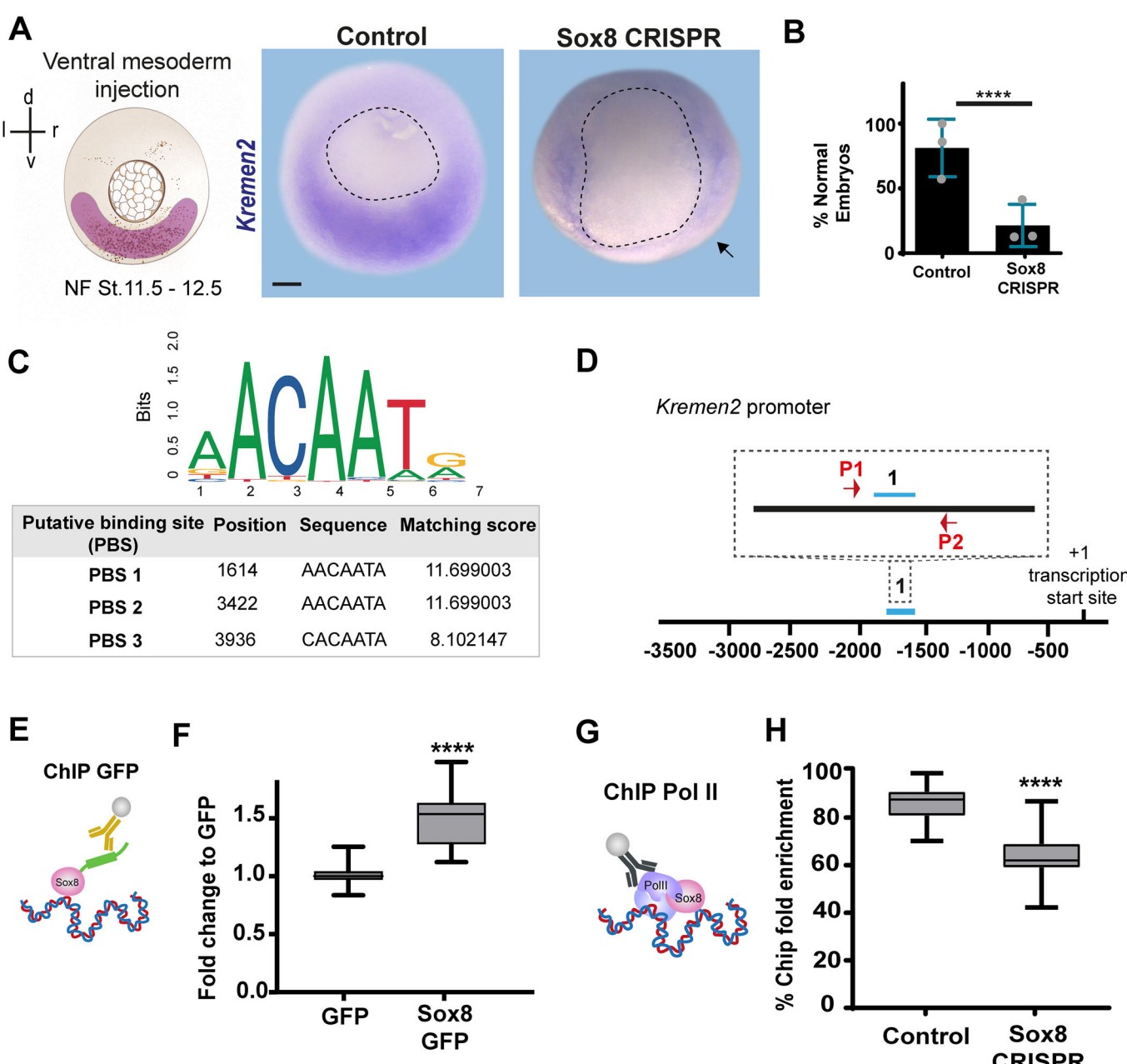

**Figure 5. kremen2 expression is directly regulated by Sox8.**

(A, B) kremen2 is downregulated in sox8 CRISPR Cas7–11 embryos, as shown by (A) representative *ISH* images and (B) quantification of the percentage of normal embryos (with kremen2 ventral signal). The arrow indicates the reduction in the kremen2 signal in the ventral region. Dotted lines delineate the blastopore. Statistical significance was assessed by two-sided Fisher's exact test, ****$P < 0.0001$. Error bars: mean ± SD; $N = 3$ independent experiments ($n = 30$ controls injected with Cas7–11 and DR; $n = 29$ sox8 CRISPR embryos). (C) Enrichment analysis of the Sox8 consensus motif, showing the localization of three putative binding sites with higher matching scores in the proximal promoter of kremen2. (D) Schematic representation of the localization of the putative binding site 1 motif in the kremen2 proximal promoter and primers used for ChIP–PCR. (E) Schematic representation of the ChIP assay using an anti-GFP antibody. (F) Box plot showing ChIP–PCR enrichment at putative binding site 1 in the kremen2 promoter, following ChIP with GFP or Sox8-GFP. Enrichment is shown relative to the GFP-only control. Statistical significance was assessed by a two-tailed Mann–Whitney test, ****$P < 0.0001$. (G) Schematic representation of the ChIP experiment using an anti-Pol II antibody to assess transcriptional activity at the kremen2 promoter in control and sox8 CRISPR embryos. (H) Box plot showing Pol II ChIP enrichment at the kremen2 promoter in the control vs sox8 CRISPR conditions. Statistical significance was assessed by a two-tailed Mann–Whitney test, ****$P < 0.0001$. $N = 3$ independent biological experiments. Scale bar: 250 μm. For box plots in (F, H), boxes represent the interquartile range (IQR; 25th-75th percentile), with the center line indicating the median and whiskers extending to the minimum and maximum values. Source data are available online for this figure.

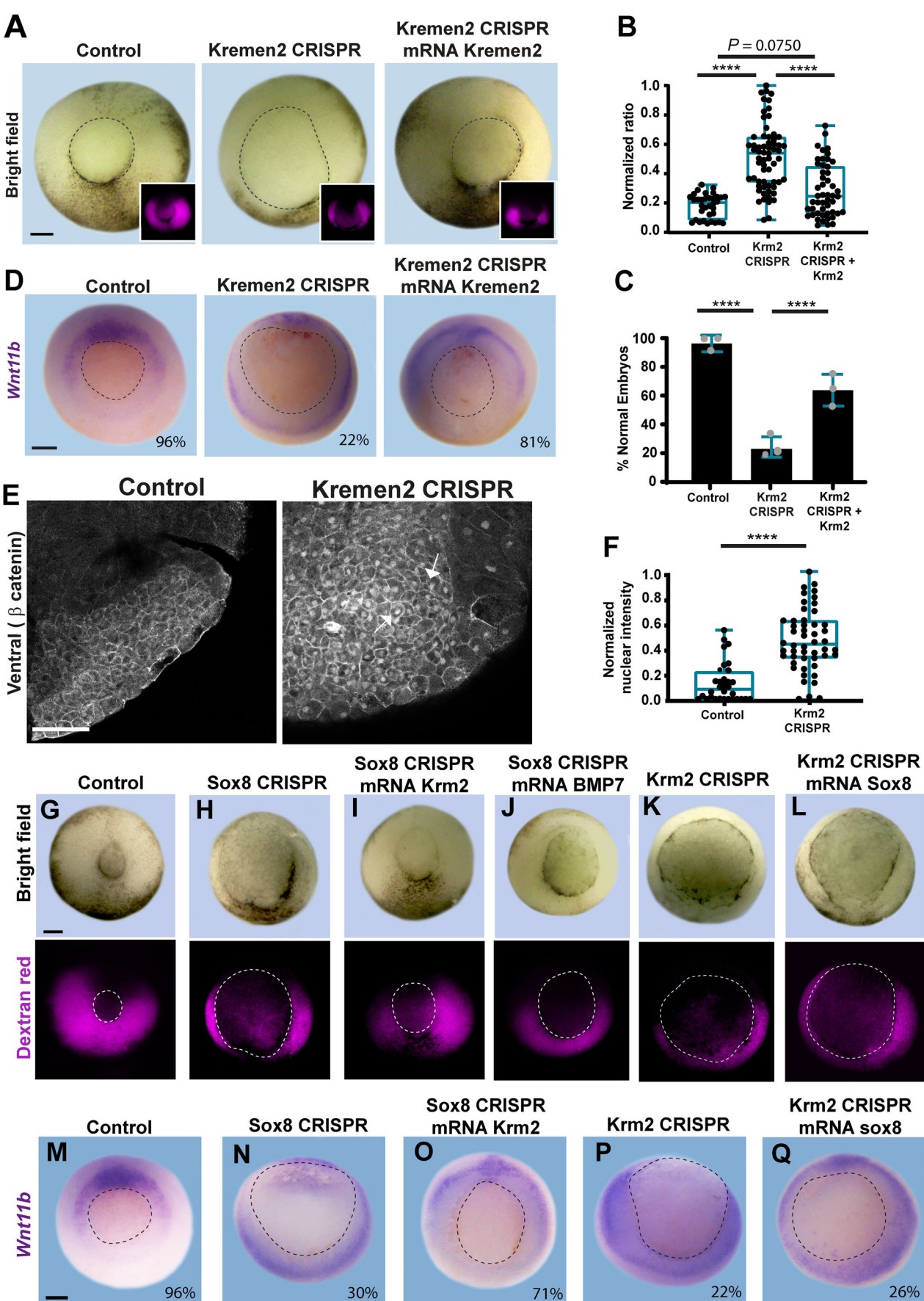

**Figure 6. CRISPR-mediated knockdown of *kremen2* recapitulates the gastrulation defects and ventral Wnt expansion observed upon *sox8* downregulation.**

(A) Representative images of gastrulation phenotypes in *kremen2* CRISPR embryos. (B) Box plot showing the normalized ratio of blastopore area to embryo area under the indicated conditions. Statistical significance was assessed by Kruskal–Wallis multiple comparisons test, ****$P < 0.0001$; ns, non-significant: $P = 0.0750$. $N = 3$ independent experiments ($n = 34$ control embryos; $n = 63$ *kremen2* CRISP; $n = 50$ *kremen2* CRISPR + *kremen2* mRNA). Scale bars: 250 µm. (C) Percentage of embryos exhibiting normal blastopore size per condition (embryos with values below the maximum control ratio value were classified as normal). Statistical significance was assessed by two-sided Fisher's exact test, ****$P < 0.0001$. $N = 3$ to 4 independent experiments; Error bars: mean ± SD. (D) ISH of *wnt11b* in the control, *kremen2* CRISPR and rescue (co-injection of *kremen2* CRISPR with mRNA *kremen2*); see Fig. EV3C, D for quantifications of the percentage of embryos displaying normal expression *wnt11b*. Scale bars: 250 µm. (E, F) β-catenin immunofluorescence showing increased nuclear β-catenin signal (arrows) on the ventral side of *kremen2* CRISPR embryos compared with controls. Scale bar: 100 µm. (F) Box plot of normalized nuclear β-catenin intensity in ventral cells. Statistical significance was assessed by a two-tailed Mann–Whitney test, ****$P < 0.0001$. $n = 4$ control embryos (32 cells); $n = 5$ embryos *kremen2* CRISPR embryos (50 cells). (G–L) Representative images of gastrulation phenotypes from epistasis experiments and respective controls for injection with Dextran Red dye. Scale bars: 250 µm. (M–Q) In situ hybridization (ISH) for *wnt11b* expression under different conditions. See Fig. EV3E–H for quantification of the percentage of embryos displaying normal *wnt11b* expression. Scale bar: 250 µm. For (A, D, G–Q), dotted lines delineate the blastopore. For box plots in (B, F), boxes represent the interquartile range (IQR; 25th–75th percentile), with the center line indicating the median and whiskers extending to the minimum and maximum values. Source data are available online for this figure.

four-cell stage into the marginal (equatorial) region of both ventral blastomeres (Kreis et al, 2021). To label injected tissue, 250 pg of nRFP mRNA or 10 nl of 0.62 mg/ml Dextran Texas Red (Life Technologies, D1863) was injected per blastomere under all the injection conditions.

A previously validated MO against *Xenopus* Sox8 (O'Donnell et al, 2006) was synthesized by GeneTools and injected at 0.03 nM/blastomere. Transcripts for injection were synthesized in vitro by using mMESSAGE mMACHINE SP6 or T7 kits (Thermo Fisher, AM1340 or AM1344). *sox8* mRNA was generated by digestion of the plasmid pCS2⁺ Sox8 (O'Donnell et al, 2006) with *Not*I, and the linearized, purified plasmid was used as a template for SP6 transcription. mRNA from *sox8* was injected at 50–250 pg/blastomere; mRNAs from *kremen2. S* and *bmp7.1 L* were generated from *Not*I linearized plasmids cloned by Gibson Assembly® Master Mix (E2611) using ventrolateral mesoderm cDNA to amplify *kremen2. S* and *bmp7.1 L* CDS. Primers 1–8 listed in Table EV1 were used to amplify CDS (insert) and pCS2⁺8CmCherry (vector) using the following PCR conditions: (i) 98 °C for 30 s; (ii) 98 °C for 10 s; (iii) 30 s at an adequate Ta temperature (Table EV1); 72 °C for 1.5 min for inserts and 2.5 min for vectors (24x cycles, steps ii–iv); and (vi) 72 °C for 2 min; mRNA from *kremen2. S* and *bmp7.1 L* were injected at a concentration of 100–250 pg/blastomere. Sox8-GFP mRNA was generated from *Not*I linearized plasmids cloned by Gibson Assembly® Master Mix using the plasmid pCS2⁺ Sox8 (O'Donnell et al, 2006) to amplify *sox8* (insert) and a PCR product generated from pCS2⁺8CeGFP (vector) using the primers 38–41 listed in Table EV1, and the following PCR conditions were used: (i) 98 °C for 30 s; (ii) 98 °C for 10 s; (iii) 30 s at adequate Ta temperature (Table EV1); 72 °C for 1 min for inserts and 3 min for vectors (24x cycles, steps ii–iv); and (vi) 72 °C for 2 min. mRNA for GFP was generated from *Not*I-linearized pCS2⁺8CeGFP and transcribed using the SP6 kit. mRNA for GFP and Sox8-GFP was injected at 300 pg/blastomere in the ventrolateral mesoderm at the four-cell stage. HuDisCas7-11 (mentioned as Cas7–11 throughout the manuscript) (Özcan et al, 2021) was cloned and inserted into a pCS2⁺ plasmid via Gibson Assembly® Master Mix. For this purpose, we performed PCR using the plasmid pDF0159 pCMV-huDisCas7-11 (Addgene, 172507) as a template for insert amplification and the plasmid pCS2⁺msfGFP (generated in this study) as a template for vector amplification using Q5® Hot Start High-Fidelity DNA Polymerase (NEB, M0493). The primers used are listed in Table EV1 (primers 46–49), and the cloning strategy used is shown in Appendix Fig. S7. The PCR conditions used were as follows: (i) 98 °C

for 30 s; (ii) 98 °C for 10 s; (iii) 30 s at an adequate Ta temperature (Table EV1); 72 °C for 2 min (19–29 cycles, steps ii–iv); and (vi) 72 °C for 2 min. The pCS2⁺ huDisCas7-11-msfGFP-generated plasmid was digested with *Not*I, and Cas7–11-msfGFP was transcribed with SP6. mRNA from Cas7–11 msfGFP was injected at a concentration of 800 pg/blastomere under all the conditions. The previous mentioned in vitro transcribed mRNAs were purified using an RNeasy kit (Qiagen) and eluted in RNase-free $H_2O$.

## CRISPR/Cas7–11 tool for RNA targeting in *Xenopus laevis*

Here we provide a workflow outlining the implementation of this tool in *Xenopus* embryos, especially focused on guide design, synthesis, Cas7–11 mRNA, microinjection and knockdown validation. We describe some important tips and troubleshooting for efficient, specific, and reliable downregulation of genes in this model organism (Appendix Fig. S4). A detailed protocol is provided in protocols.io (https://doi.org/10.17504/protocols.io.n92ld67zxg5b/v1)

### 1. Designing gRNAs for the 3′ UTR of the mRNA target sequence (in silico)
Estimated timing: 1–2 h.

1.1 Search for your gene in Xenbase (https://www.xenbase.org) (Fisher et al, 2023). In "Nucleotides", select all the transcripts and paralogues for your gene and download the complete mRNA sequences (including UTR's). We selected *kremen2. S* sequences NM_001088883.1 and XM_041577699.1, and for *sox8*, we selected *sox8. L* (NM_001090495.1) and *sox8. S* (XM_018239299);

1.2 Perform an alignment between different transcript variants and/or paralogues and identify the 3′UTR;

1.3 To design the most promising guides, use a platform (https://cas13design.nygenome.org/) that provides optimized spacers to target transcripts in the human transcriptome, model organisms and viral RNA genomes using CRISPR-Cas13 (Wessels et al, 2020; Guo et al, 2021); This platform relies on a machine learning-based "on-target" model that takes into consideration several parameters, such as the crRNA-fold energy, the local target C context and the upstream target U context (Wessels et al, 2020);

a) Go to "Design custom gRNAs", since *Xenopus laevis* is not yet included in the list of model organisms;

b) We started by designing guides for the transcript with the longest 3'UTR. For sequences <500 nt, copy and paste the sequence where you want to design the guides. If a sequence is larger than 500 nt, use "upload multiple single-entry fasta files", and the results will arrive at the email within a few minutes;

c) From the list provided, select top three spacers positioned in quartile 4 (Q4) with a minimum distance of 50 nt between them to avoid potential competition between gRNAs for target binding (Hernandez-Huertas). Each spacer will have 23 nt;

d) We recommend evaluating the most promising spacer sequences by aligning them with various transcript variants and/or paralogues and choosing those that show conservation across these sequences;

e) Optional but recommended: Analyse if the selected spacers have potential off-target effects. For this purpose, a BLAST search of the spacers was performed against the RNA-seq data of the *Xenopus laevis* database in NCBI. Consider a direct off-target if it has up to four mismatches with the target (Kushawah et al, 2020). To further evaluate the off-target risk, we analysed whether the potential off-target was expressed in the tissue of interest/developmental stage.

Note 5: According to a study using CRISPR-Cas13 (Kushawah et al, 2020), guides for the CDS are more efficient at inducing a phenotype than guides for the UTRs. However, a great disadvantage in designing guides for the CDS is the impossibility of performing rescue experiments by reintroducing the mRNA of the gene being targeted. In this case, you can perform the rescue with a version that is not targeted by your guides (for instance, creating a CDS resistant to your guides using silent mutations or expressing the CDS of another species that cannot be targeted by your guides).

**2 Preparation of the template PCR for gRNA transcription**
Estimated timing: 3 h.

2.1 The DNA template used to generate the gRNA was produced by fill-in PCR (Hernandez-Huertas) using the following primers:

a) Universal primer: containing the direct repeat (DR) for the CAS7–11 interaction used in Özcan et al, 2021 (Özcan et al, 2021) and the T7 minimal promoter (Appendix Fig. S5).

b) Specific primer: This primer contains the complementary region that binds the linker and a sequence that should correspond to the mRNA target region (the same as the reverse complement of the guides) (Appendix Fig. S5).

Note 6: We strongly recommend to simulate the PCR with your designed primers using to SnapGene® software (Go to "Actions" and then "PCR") or other tool that allows to simulate your PCR to avoid any error in the design of the specific primer.

2.2 A mixture of the universal primer and specific primers (see primers 9–15 in Table EV1) was used. In our case, we performed PCR using a pool of three different gRNA-specific primers at equimolar concentrations (3.3 µM of each primer per 50 µl of PCR, which corresponds to ~10 µM in total).

Note 7: Alternatively, gRNAs can be individually produced by performing separate PCRs for each gRNA template using 10 µM of the specific primer and 10 µM of the universal primer. In the injected solution, guides should be mixed at equal concentrations.
Note 8: We recommend performing at least three PCRs and mixing them to increase the amount of DNA produced after purification. Negative controls with universal or specific primers alone can be run to ensure that the PCR bands are correct.

2.3 We performed PCR using Q5® Hot Start High-Fidelity DNA Polymerase (NEB, M0493), but other DNA polymerases can be used (just following the proper manufacturer's instructions). In our case, the PCR conditions were as follows: (i) 98 °C for 30 s; (ii) 98 °C for 10 s; (iii) 30 s at 61 °C; (iv) 72 °C for 30 s (29× cycles, steps ii–iv); and (vi) 72 °C for 2 min.

2.4 All three PCR products were mixed, and 5 µl was run on a 2% agarose gel (Appendix Fig. S6). A band less than 100 bp (75 bp) should be observed;

2.4 The remaining PCR products were purified using a MinElute Gel Extraction Kit (Qiagen Cat# 28604) following the manufacturer's instructions. This kit allows a good yield to be obtained in a small volume of elution.

Note 9: The minimum amount of nuclease-free water recommended for elution was used. This will increase the concentration of the sample for the posterior step of in vitro transcription.

**3 In vitro transcription of the gRNAs**
Estimated time: 6 h.

3.1 We optimized the protocol for gRNA synthesis and purification as follows:

a) Use 500 ng of the purified PCR template per in vitro transcription;

b) A mMESSAGE mMACHINE T7 kit (Thermo Fisher, AM1344) was used following the manufacturer's instructions. We performed transcription at 37 °C for 4 h;

c) We added 1 µl of TURBODNase (2 U/mL) (provided with the transcription kit) and incubated at 37 °C for 1 h;

Note 10: CRITICAL: gRNAs should be purified via small RNA purification kits. We used a Monarch RNA Cleanup Kit (NEB) to purify the RNA following the instructions, but with the adjustment for smaller RNAs and RNA with secondary structures. Thus, we diluted our samples with two volumes of ethanol instead of one in Step 2 of the manufacturer's protocol. Elution should also be performed in the minimum volume of nuclease-free water recommended;

Note 11: <u>CRITICAL:</u> The use of a spectrophotometer (Nanodrop) for gRNA quantification will overestimate the gRNA concentration (Hernandez-Huertas). A Qubit RNA BR (Broad-Range) assay kit (Invitrogen) is crucial for accurate quantification of these small RNA fragments.

d)  Always run a bleach gel (1% commercial bleach/2% agarose) to check for the integrity of the gRNAs (Appendix Fig. S6). The use of parallel samples whose concentrations were previously known could also be a good quality control.

Note 12: Do always three to four transcriptions and mix them to have enough amount for all the experiments, using the same batch of gRNA. Expected yields can vary because of their composition and secondary structures (Hernandez-Huertas), but in general, concentrations between 100 and 500 ng/µl can be obtained using this method.

Note 13: We always recommend storing gRNAs at −80 °C in several aliquots (e.g., 5 µl) to avoid freeze-thaw cycles.

### 4 In vitro transcription of Cas7–11-msfGFP mRNA
Estimated time: 2 days.

4.1 We used 6 µg of the pCS2$^+$ huDisCas7-11-msfGFP plasmid produced in this study (Appendix Fig. S7 and see Methods for further details) to digest the mixture for 4 h with NotI-HF (NEB, # R3189S).

Note 14: We usually use a 100 µl digestion volume to increase the chance of obtaining good HuDisCas7-11-msfGFP concentrations after purification.

4.2 The products were purified with a MinElute Gel Extraction Kit (Qiagen, Cat# 28604) and transcribed (2 µg was used for transcription via the mMESSAGE mMACHINE™ SP6 Transcription Kit); incubation was performed at 37 °C for 4 h.

Note 15: After purification, 1 µl of the digested plasmid and the same amount of non-digested plasmid were run to check for digestion efficiency.

4.3 Proceed with digestion with TURBO DNase (1 h at 37 °C);
4.4 Proceed with the cleanup using an RNeasy kit (Qiagen);
4.5 The mRNA was quantified via a NanoDrop™ or Qubit RNA BR (Broad-Range) assay kit (Invitrogen) and run on a gel bleach gel (as described previously).

### 5 Microinjection of embryos
Estimated time: 1–2 h.

5.1 In this study, we injected the dorsal and ventral blastomeres (right or left) of eight cell-stage embryos to target the neural crest and the two ventral vegetal blastomeres of four cell-stage embryos to target the ventral mesoderm, as described in the Methods section.
5.2 Guides (Table EV2) were injected at 160 pg/blastomere and mRNA from Cas7–11-msfGFP was injected at a concentration of 800 pg/blastomere under all the conditions.

Note 16: We recommend performing an initial titration to determine the concentration of gRNA and Cas7–11-msfGFP mRNA that should be used without affecting the viability of the embryos or the biological process.

### 6 Validation of mRNA knockdown
Estimated time for sqRT-PCR and ISH - 2 weeks. **Estimated time for RNA-seq: variable (months)**

6.1 To validate the efficiency of the knockdown, we propose the following complementary or alternative experiments:
  a)  SqRT-PCR or qPCR: Analysing the mRNA levels of a target gene through semi-quantitative RT-PCR analysis is a fast way to address the knockdown efficiency if the depletion is successful and evident. For a more quantitative analysis, use qPCR with specific primers for the target gene;
  b)  In situ hybridization: This is a complementary way to check for the efficiency of depletion and requires designing specific probes for the gene you want to analyse or using probes already published and validated:
  b)  RNA-seq: This type of analysis will reveal not only the efficiency of depletion but also how the whole transcriptome is affected.
6.2 Phenotype validation should always be performed with rescue experiments for the gene you are targeting. If guides are designed to target UTRs, a rescue experiment can be performed by simply reintroducing the coding sequence of a gene; titration of the amount of mRNA should be performed to avoid dominant effects caused by overexpression.

### 7 Troubleshooting

#### 7.1 Problem: Low gRNA and Cas7–11 mRNA concentrations

a)  If a low gRNA concentration is obtained, check the design of the primers, ensuring that the T7 minimal promoter is included in the "universal primer";
b)  For Cas7–11 mRNA, check the linearization efficiency of the plasmid;
c)  If low gRNA and/or Cas7–11 mRNA levels are obtained, the amount and purity of the template used for in vitro transcription should be checked, as these aspects are crucial for successful in vitro transcription;
d)  For gRNA purification, use a kit that allows the purification of low-quality fragments.

e) Alternatively, commercially-synthetized gRNAs (Moreno-Sánchez et al, 2025) can be used to reduce the variability related to guide concentration and quantification, as well as associated toxicity effects. This approach can also save time and lower overall costs.

### 7.2 Problem: Deficient RNA depletion and weak/absent phenotype

a) Check the design of your guides;

b) Do not use poly(A) tailing in your gRNAs, as this can critically alter the structure of your guides;

c) For linearization of the plasmid to produce Cas7–11 mRNAs, a single-cut restriction that cuts after poly(A) is used; in this case, poly(A) is essential to confer stability and enhance translation efficiency in vivo;

d) The concentration of each gRNA must be checked via a reliable method (with a Qubit fluorometer). As discussed before, this is a critical aspect to consider because the guide concentration can be overestimated by a NanoDrop spectrophotometer;

e) The samples (e.g., 500 ng) were run in 1% commercial bleach/2% agarose as suggested before to check the integrity of the gRNAs and Cas7–11 mRNA; if possible, a sample of gRNAs or Cas7–11 mRNA that worked previously was run in parallel;

f) Always ensure that you are using RNAse-free material during the transcription and purification processes and avoid freeze–thaw cycles;

g) If an embryo has a low survival rate, a lower concentration of gRNAs and/or Cas7–11 mRNAs should be tested;

h) The efficiency of targeted injections should always be carefully checked. Use a fluorescent dye (e.g., dextran red) to confirm the efficiency of the injection (as the msfGFP of the Cas7–11 signal can sometimes be faint owing to embryo autofluorescence and/or the ubiquitous distribution of Cas7–11–msfGFP in cells);

i) If the signal of an injection marker is faint, it might be possible that the injection conditions (e.g., technical aspects related to the needle) are not optimal. Additionally, a high amount of mRNA may be toxic to cells, resulting in death, fewer injected cells and consequently a less penetrant phenotype.

### In situ hybridization

In this study, we used the following in situ hybridization probes: (1) *sox8* (O'Donnell et al, 2006) (provided by Jean-Pierre Saint-Jeannet); (2) *snail2/slug* provided by Eric Thevenau laboratory (Mayor et al, 1995); (3) *ventx1* (provided by Philipp Vick laboratory); (4) *kremen2* (Janesick et al, 2018); (5) *wnt11b* (this study); (6) *chordin* (provided by Jean-Pierre Saint-Jeannet laboratory); (7) *myoD* (provided by Jean-Pierre Saint-Jeannet laboratory); (8) *xbra* (provided by Jean-Pierre Saint-Jeannet laboratory); and (9) *bmp7* (this study). In brief, digoxigenin–labeled antisense template probes were generated as follows: for the *sox8* and *chordin* probes, the plasmid was linearized using *EcoRI* (NEB) and used as a template for T7 RNA polymerase; for *snail2/slug*, the plasmid was linearized with *BlgII* (NEB) and transcribed using the probe using SP6 RNA polymerase; for *ventx1*, the plasmid pBSK-ventx1 was digested with *SaII* (NEB) and used as a template for T7 RNA polymerase; for *kremen2*, an antisense PCR

fragment of approximately 800 bp containing the T7 promoter was amplified using cDNA from ventrolateral mesoderm and the primers indicated in the Table EV1 (primers 16 and 17); for the *wnt11b* probe, an antisense PCR fragment of 841 bp containing the T7 promoter was amplified using the primers indicated in the Table EV1 (primers 18 and 19), and it was used as a template for transcribing the antisense probe via the T7 promoter; for the *myoD* probe, the plasmid was linearized with *HindIII* (NEB) and used as a template for the T7 RNA polymerase; for the *xbra*, the plasmid was linearized using *BlgII* (NEB) and used as a template for T7 RNA polymerase; and finally for *bmp7.1*, the plasmid BMP7-CS110K was used as a template for antisense PCR fragment amplification using primers indicated in Table EV1 (primers 20 and 21). For all the probes, we used the standard protocols from the Promega Riboprobe® in vitro transcription systems (P1085 for the SP6 and P2075 for the T7 RNA polymerase kit), but used DIG RNA labeling mix (Roche, 11277073910) instead of ribonucleotides from the kit (Barriga et al, 2019). RNA cleanup was performed using the RNeasy kit (Qiagen, 74104), and the RNA was eluted in RNase-free $H_2O$. Probes were used at a concentration of 3 µg/ml in hybridization buffer solution.

In situ hybridization was performed following a previously described protocol for neural crest markers (Barriga et al, 2019) with additional steps of permeabilization and acetylation after the rehydration steps to improve penetration and specific probe binding in gastrulation (Willsey, 2021).

### Immunofluorescence

The immunofluorescence protocol for β-catenin and pSmad was based on a published protocol (Plouhinec et al, 2013). The embryos were fixed in 1x MEMFA for 4 h at RT and washed in 1x PBS for 5 min. Sagittal sections of the embryos were obtained using a scalpel blade, and half of the embryos were dehydrated in increasing concentrations of methanol (MeOH): 25% (v/v), 50% (v/v), and 70% (v/v) in $H_2O$ for 5 min each and washed twice in 100% MeOH. To remove autofluorescence, the embryos were rinsed with 100% MeOH in 2:1 MeOH:30% $H_2O_2$ for 2 h and rehydrated in 70%, 50%, or 25% MeOH in $H_2O$ for 5 min each. After this step, the embryos were washed twice in PBS 0.1% Tween 20 containing 0.2% BSA for 10 min and incubated in blocking solution (PBS 0.1% Tween 20 containing 0.2% BSA) for 1 h at RT. The embryos were incubated with primary antibody diluted in blocking solution overnight at 4 °C (rabbit pSmad1/5/8 antibody at 1:50; Millipore, AB3848-I; mouse β-CAT active at 1:50; Sigma-Aldrich 05-665). Excess primary antibodies were removed by three washes of 30 min each in PBS-Tween 0.1% containing 0.2% BSA. After washing, the embryos were incubated in blocking solution for an additional 1 h at RT and then incubated with anti-mouse or anti-rabbit Alexa Fluor 647 secondary antibodies (1:100) and DAPI (1:500) in blocking solution at 4 °C. The excess secondary antibody was removed by three washes of 30 min each in PBS-Tween 0.1% containing 0.2% BSA, in 1× PBS for 10 min, washing in 70% MeOH in $H_2O$ for 5 min and 100% MeOH quickly, followed by 2 h at RT. Finally, the embryos were flat-mounted for imaging in a clearing reagent mix (two volumes of benzyl benzoate and one volume of benzyl alcohol; Sigma-Aldrich).

### sqRT–PCR

RNA extraction from the ventrolateral mesoderm was performed as described in the RNA-seq procedures and analysis section, whereas

RNA extraction from the heads (stage 21) was performed using the RNeasy Mini Kit (Qiagen, 74104). For head samples, cDNA was generated with SuperScript IV (Invitrogen, 18090050) using Oligo (dT) Primer 50 μM (Invitrogen, AM5730G). Amplification by PCR was performed using the Q5 Hot Start High-Fidelity DNA Polymerase with the primers indicated in Table EV1 (primers 22–37). The PCR steps were as follows: (i) 98 °C for 30 s; (ii) 98 °C for 10 s; (iii) 30 s (annealing) at the temperatures indicated in Table EV1; (iv) 72 °C for 20 or 30 s (32–35x cycles, steps ii–iv); and 72 °C for 2 min. Amplicons were resolved on 0.8–2% (w/v) agarose gels. For semi-quantitative analysis, after background subtraction in an adjacent area, the band intensity of each gene was quantified using three different regions of interest (ROIs), and the ratio of each gene to *ef1α* was determined. The ratios were then normalized to the maximum value.

### RNA-seq procedures and analysis

The RNAseq experiment was performed and optimized by the Genomic Unit of Instituto Gulbenkian Ciência (IGC), Oeiras. mRNA libraries from ventrolateral mesoderm tissues were prepared using the Smart-seq2 protocol adapted from Macaulay et al, 2016. RNA quality was assessed using a HS RNA Screen Tape Analysis (Agilent Technologies), and Illumina libraries were generated according to the Nextera protocol (Baym et al, 2015). The libraries were quantified and assessed for quality using the Agilent Fragment Analyser (AATI) in combination with an HS next-generation sequencing (NGS) kit (Agilent Technologies). Sequencing was carried out on a NextSeq500 Sequencer (Illumina) using 100 SE high-throughput kit. Sequence information was extracted in FastQ format using the bcl2fastq v2.19.1.403 (Illumina).

FastQC v0.12.1 and multiqc v1.14. Adapter sequences were removed, and reads with low-quality or length less than ten bases were discarded using cutadapt v4.4 (Martin, 2011) (with the following parameters: -b CTGTCTCTTATACACATCT-b GTATC AACGCAGAGTACT -b TTTTTTTTTTTTTTTTTTTTTTTTT TT -b AAAAAAAAAAAAAAAAAAAAA -m 10 -q 20). The rRNA sequences were filtered out with SortMeRNA v4.3.6 (Kopylova et al, 2012) using eukaryotic 18S and 28S rRNA as reference sequences. The trimmed, rRNA-depleted sequences were therefore aligned to the reference genome of *X. laevis* v10.1 using HISAT2 v2.2.1 with default parameters (Kim et al, 2019). SAM files were converted into BAM files, sorted via SAMtools v1.5, and transcript abundance was quantified using featureCounts v2.0.6 (Liao et al, 2014), resulting in a raw count matrix of mapped reads per gene. Genes presenting a mean expression lower than 5 across all samples were filtered, and the resulting raw count matrix was normalized to account for technical variations, including library size, using the "DESeq2" package in R (Love et al, 2014). Differential gene expression analysis identified genes with statistically significant expression changes between conditions, applying a threshold of |Log2FC| >0.58. Additionally, the normalized counts for each condition were subjected to a "regularized log" transformation for downstream visualization. The transformed counts of the intersecting genes are displayed in a heatmap generated with the "heatmap" package in R.

### g: Profiler analysis

The differentially expressed genes identified via RNA-seq were used for functional enrichment analysis. Functional enrichment analyses were performed using g:GOSt from g:Profiler (Raudvere et al, 2019) against the reference genome *Mus musculus*. Gene Ontology (GO) analyses were performed using two reference databases: molecular function (MF), biological process (BP), and KEGG biological pathway. Terms with a *p* value <0.05 with g:SCS correction were used for analysis. The results are represented by Manhattan plots provided by g:Profiler (with small modifications to improve visualization quality) and by plot bars showing the most relevant BP- or KEGG-enriched terms and their respective $\log^{10}$ *p* values.

### ChIP–PCR procedures and motif enrichment analysis

Chromatin immunoprecipitation (ChIP) was performed following established protocols for *X. laevis* embryos (Akkers et al, 2012; Gentsch and Smith, 2014). Following injection and embryo pilling, the ventral halves were dissected at NF stages 11–11.5 using a scalpel, collected, fixed for 30 min (room temperature), and quenched. After washing, sonication was performed at 30% power (30 s on 30 s off) for 40 min (ON time) in a Qsonica Q800R3 sonicator. For immunoprecipitation, 5 μg of Pol II (Diagenode, C15200004) or 2.5 μg of anti-GFP antibodies (Abcam, ab6556) were incubated ON at 4 °C with Dynabeads™ Protein G (Thermo Fisher Scientific, 10004D) or M-280 Dynabeads (Thermo Fisher Scientific, 11203D), respectively. Equal amounts of sonicated chromatin from the control and experimental conditions were then incubated ON with the antibody-bound beads to precipitate Pol II- or GFP-associated DNA. Following immunoprecipitation, the DNA was eluted, decrosslinked and purified via phenol:chloroform extraction, as previously described.

To find putative binding sites for Sox8 in the *Kremen2* genomic region, 3.5 kb upstream of the transcription start site (TSS) region was manually extracted using the reference genome of *X. laevis* v10.1 and scanned for the SOX MA0868.3 motif of Sox8 from JASPAR2022 CORE, using a custom python script. For motif enrichment analysis, the simple enrichment analysis (SEA) tool from the MEME suite (Bailey and Grant, 2021) was used, with the JASPAR2022 CORE vertebrate non-redundant v2 database used as the reference for motif discovery.

The primer pair for the selected putative binding site was designed to analyse chromatin enrichment via PCR and is indicated in Table EV1 (primers 42 and 43). PCR was performed under the following conditions: 98 °C for 30 s; 98 °C for 10 s; 67 °C for 30 s; 72 °C for 10 s for 34–39 cycles; and 72 °C for 2 min. For the other putative binding site, the primers and annealing temperature used are also listed in Table EV1 (primers 44–45).

The intensities of the PCR bands were measured using the pixel intensity distribution (plot profile) in FIJI, and the background for each band was subtracted by using an adjacent ROI. Each dataset was normalized to the maximum value. For each experiment, we used three biological replicates with approximately 70–80 embryos per condition.

### Microscopy

For imaging of the immunofluorescences, z-stack series were acquired on a Leica Stellaris 5 upright system using an HC PL APO CS2 10x/0.40 dry and 405 and 647 nm laser lines. The system was controlled by LAS X (Leica). For live imaging, embryos were embedded in 0.7% low-melting agarose (Sigma, A9414-5G) and imaged in a Thunder Imager 3D Cell Culture system (Leica) using the HC PL APO 10x/0.40-NA objective (Leica) in time-lapse mode every 10 min.

### Imaging of whole embryos

All whole embryo images were captured using agarose dishes with small depressions to accommodate the embryos. For ISH embryos, MAB-T was used to fill the dishes, whereas 0.3x MMR was used for living embryos. For living embryos and some ISH (Fig. EV2), Flexacam C1 or C3 (Leica) mounted onto stereoscopes were used at 2.5x magnification using Leica's software (Leica, IM50). Images from other ISHs of gastrulation were obtained using a Zeiss Stereo LUMAR stereoscope equipped with a Hamamatsu Orca-Flash 2.8 camera controlled with MicroManager v1.14 software.

### Data processing and analysis

Rotations and lookup images (magenta) were obtained via Fiji (Schindelin et al, 2012). To adjust the contrast and brightness, we used both Fiji and Adobe Photoshop CS6. The scale bars were added via Fiji or Adobe Illustrator 2021. In the in situ hybridization images, the background was pseudocolored in Adobe Photoshop CS6 for clarity purposes without interfering with the embryos.

Quantification of the in situ hybridization *wnt11b* signal and area: The ventral expansion of the *wnt11b* signal in CRISPR embryos was quantified by designing an equatorial line in the embryo image and using the manual ROI tool from Fiji to select the dorsal and ventral signal above and below the line, respectively. The background quantified in a nonstained region of the embryo was subtracted from the mean gray signal intensity. The ratios between the ventral and dorsal intensities were normalized against the maximum intensity value. The *wnt11b* signal area was quantified by manually selecting regions of interest (ROIs) using Fiji and calculating the ratio of the *wnt11b* signal area to the total area of the corresponding embryos. These ratios were then normalized to the maximum observed value.

Quantification of pSmad and β-catenin nuclear intensity: To quantify the nuclear signals of pSmad and β-catenin, a single Z plane was selected. Using the DAPI channel, a circular ROI around the nucleus of 10 randomly chosen cells was used. Mitotic cells (ellipsoid nuclei) were excluded from the quantification. The background for each cell was quantified using a circular ROI in the cytoplasm and subtracted from each intensity value from the nucleus. Normalization to the maximum intensity value was performed.

### Statistical analysis

The experiments were not blinded due to their nature, as only viable embryos and well-injected embryos were considered for the analysis. After selection, the embryos were randomly analysed. The exact $P$ values, sample sizes, statistical tests employed and number of biological replicates are stated in the figures' legends. For all the experiments, outliers were identified and removed via the statistical Grubb's test and/or the ROUT method in Prism 10.2.2. The statistics used for each experiment are annotated in the figure legends. For all the datasets, normality was tested using the D'Agostino-Pearson and/or Shapiro-Wilk tests in Prism 10.2.2 (GraphPad). For data that did not pass the normality tests, we used the Mann-Whitney test (two-tailed, unequal variances) or the Kruskal-Wallis test with Dunn's correction for multiple comparisons in Prism 10.2.2. For data that passed the normality tests, we used ordinary one-way ANOVA (Tukey's multiple comparisons test) for multiple comparison datasets or two-tailed *t*-tests to compare two conditions. Contingency analyses were performed using the two-tailed Fisher's exact test. For all the experiments, a 95% confidence interval was used. Data analysis and charts were performed with Graph Pad Prism 10.2.2 and formatted in Adobe Illustrator 25.0.

## Data availability

All the data that support our findings are displayed in manuscript figures or in the expanded view. All other information will be available upon reasonable request to the corresponding author. The RNA-seq datasets produced in this study are available at the Gene Expression Omnibus GSE GSE302963 (https://www.ncbi.nlm.nih.gov/geo/query/acc.cgi?acc=GSE302963)

The source data of this paper are collected in the following database record: biostudies:S-SCDT-10_1038-S44319-025-00617-z.

## Peer review information

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

## Acknowledgements

The authors thank Barriga laboratory members for fruitful discussions and acknowledge services from the IGC's Advanced Imaging, Genomics, and Aquatic animal facilities. We also thank Professor Jean-Pierre N. Saint-Jeannet (NYU Dentistry Research Faculty) for the plasmids used for in situ hybridization of several genes (*sox8*, *chordin*, *myoD*, and *xbra*) and Doctor Philipp Vick (University of Hohenheim) for the *ventx1* plasmid. We also thank the Harnos Laboratory (Brno, Czech Republic) for providing the β-catenin antibody. Work at Barriga Lab is supported by grants from the European Research Council Starting Grant (ERC-StG) under the European Union's

Horizon 2020 research and innovation program, Grant agreement No. 950254 (to EHB); the European Molecular Biology Organization (EMBO) Installation Grant, Project No. 4765 (to EHB); and EMBO Young Investigator Program, Project No. 5248 (to EHB); La Caixa Junior Leader Incoming, No. 94978 (to EHB); and the Deutsche Forschungsgemeinschaft (DFG, German Research Foundation) under Germany's Excellence Strategy (EXC 2068, 390729961 Cluster of Excellence Physics of Life of TU Dresden). EHB also acknowledges support from Instituto Gulbenkian de Ciência (IGC) and Fundação Calouste Gulbenkian (FCG). SM was funded by a Fundação para a Ciência e a Tecnologia (FCT) postdoctoral contract, 2020.00759. CEECIND (to SM); AGK holds a PhD fellowship 36/8 I-D/21 from Fundação Calouste Gulbenkian and Gulbenkian Institute for Molecular Medicine (GIMM).

## Author contributions

**Sofia Moreira**: Conceptualization; Data curation; Formal analysis; Funding acquisition; Validation; Investigation; Visualization; Methodology; Writing—original draft; Writing—review and editing. **Artemis, G Korovesi**: Data curation; Formal analysis; Validation; Visualization; Methodology; Writing—review and editing. **Elias, H Barriga**: Conceptualization; Resources; Data curation; Supervision; Funding acquisition; Validation; Investigation; Visualization; Methodology; Writing—original draft; Project administration; Writing—review and editing.

Source data underlying figure panels in this paper may have individual authorship assigned. Where available, figure panel/source data authorship is listed in the following database record: biostudies:S-SCDT-10_1038-S44319-025-00617-z.

## Funding

## Disclosure and competing interests statement

The authors declare no competing interests.

# Expanded View Figures

**Figure EV1.** *sox8* **knockdown by Cas7–11 in the ventrolateral mesoderm impairs embryonic development.** ▶

(A, B) Time-course analysis of the blastopore area shows delayed closure in *sox8* CRISPR embryos compared with controls. Dotted lines delineate the blastopore. Statistical significance was assessed using the two-tailed Mann–Whitney test, ****$P < 0.0001$. Error bars: mean ± SD. (**C**) Representative images of *sox8* CRISPR embryos at the neurula and tailbud stages showing open blastopores (arrow) and shortened anteroposterior axis. Dotted lines delineate the blastopore. (**D**) Percentage of embryos with normal morphology at the neurulation stage. Statistical significance was assessed using a two-sided Fisher's exact test, **** $P < 0.0001$. Error bars represent mean ± SD. $N = 3$ independent experiments ($n = 32$ control, $n = 56$ *sox8* CRISPR Cas7–11). (**E**) Quantification of embryos with normal morphology at the tailbud stage, showing a significant reduction in normal morphology following *sox8* knockdown. Two-sided Fisher's exact test, **** $P < 0.0001$. Error bars represent mean ± SD; $N = 3$ independent experiments ($n = 33$ control, $n = 55$ *sox8* CRISPR). Scale bars: 250 μm.

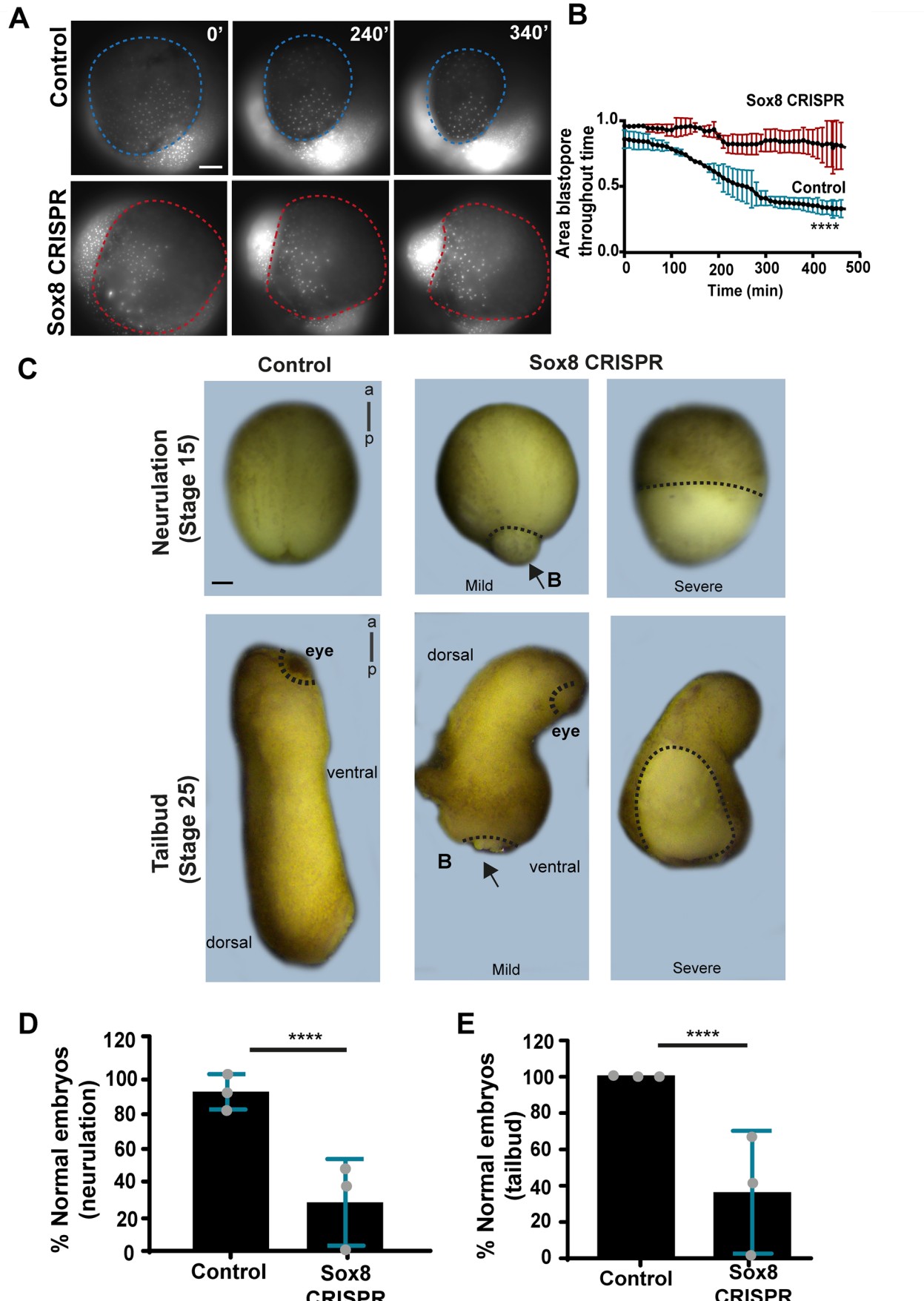

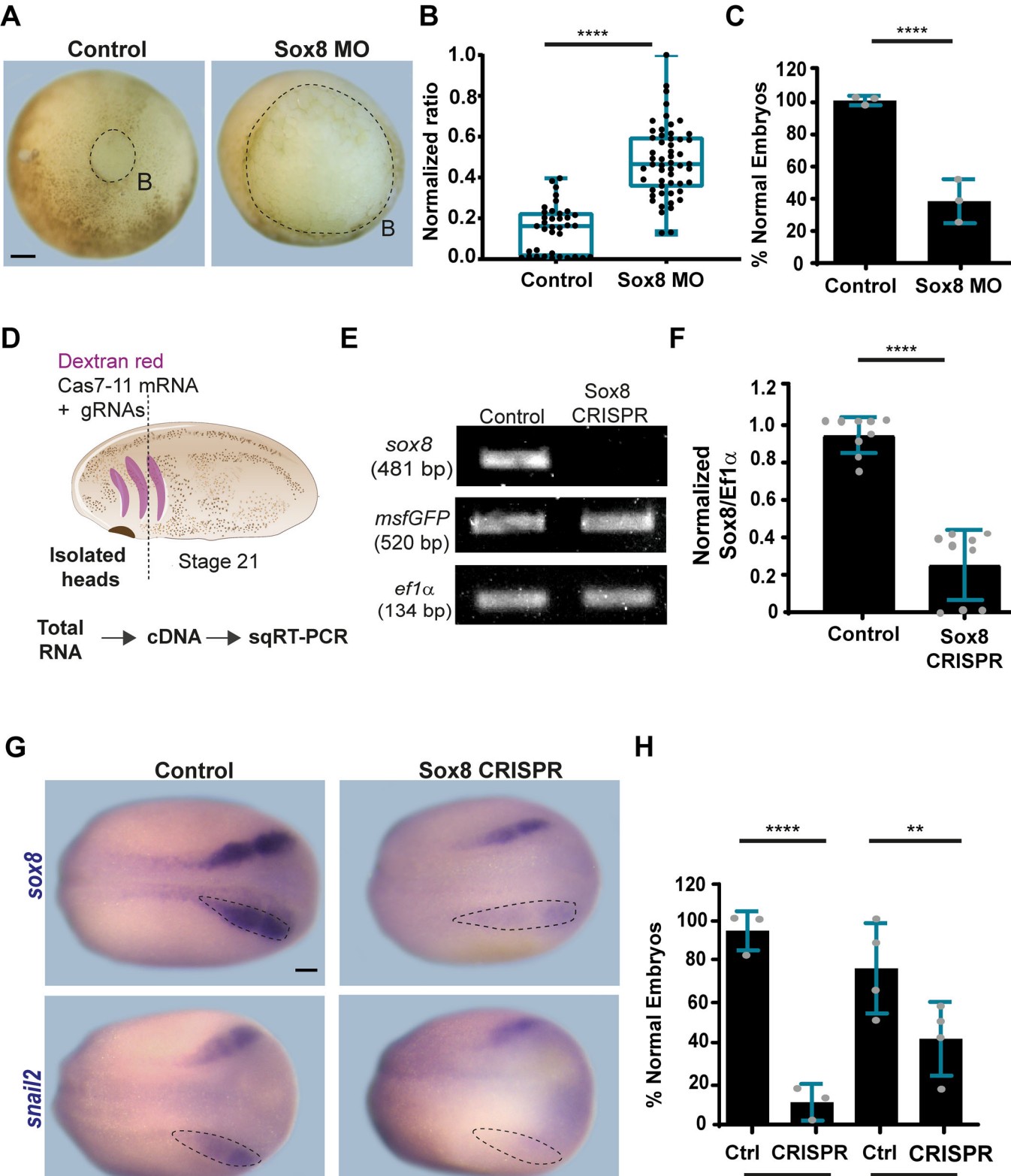

**Figure EV2.  *sox8* CRISPR recapitulates Sox8 MO phenotypes during the gastrulation and neurula stages.**

(A–C) Injection of a validated Sox8 morpholino (MO) into the ventrolateral mesoderm induces defects in blastopore closure, as illustrated by representative examples. Dotted lines delineate the blastopore (B). (B) Box plot showing the normalized ratio of blastopore area to embryo area. Boxes represent the interquartile range (IQR; 25th–75th percentile), with the center line indicating the median and whiskers extending to the minimum and maximum values. Statistical significance was assessed using the two-tailed Mann–Whitney test, ****$P < 0.0001$; $N = 3$ independent experiments ($n = 37$ control embryos; n = 54 sox8 MO). Scale bars: 250 μm. (C) Percentage of embryos exhibiting normal blastopore size per condition (embryos displaying values below the maximum control ratio value are classified as normal). Two-sided Fisher's exact test, ****$P < 0.0001$; Error bars: mean ± SD. $N = 3$ independent experiments ($n = 37$ control; $n = 52$ *sox8* MO). (D–F) Semi-quantitative RT–PCR of stage 21 heads (early neurulation), confirming the depletion of *sox8*. The purple stripes represent target injections in the neural crest. Statistical significance was assessed using a two-tailed Mann–Whitney test, ****$P < 0.0001$; Error bars: mean ± SD. $N = 3$ independent experiments. (G) CRISPR Cas7–11-mediated depletion of *sox8* in the neural crest phenocopies previously reported *sox8* MO phenotypes in the neural crest, such as a delay in neural crest induction (as detected by *ISH* for *snail2*). Dotted lines delineate the neural crest region. (H) Quantification of the percentage of embryos with normal signals for *sox8* or *snail2*. Two-sided Fisher's exact test, ****$P < 0.0001$; **$P = 0.0033$. Error bars: mean ± SD. $N = 3$ independent experiments; $n = 17$ control embryos (*ISH snail2*); $n = 18$ s*ox8* CRISPR Cas7–11 (*ISH snail2*); $n = 38$ control embryos (*ISH sox8*); $n = 41$ *sox8* CRISPR (*ISH Sox8*). Scale bars: 250 μm.

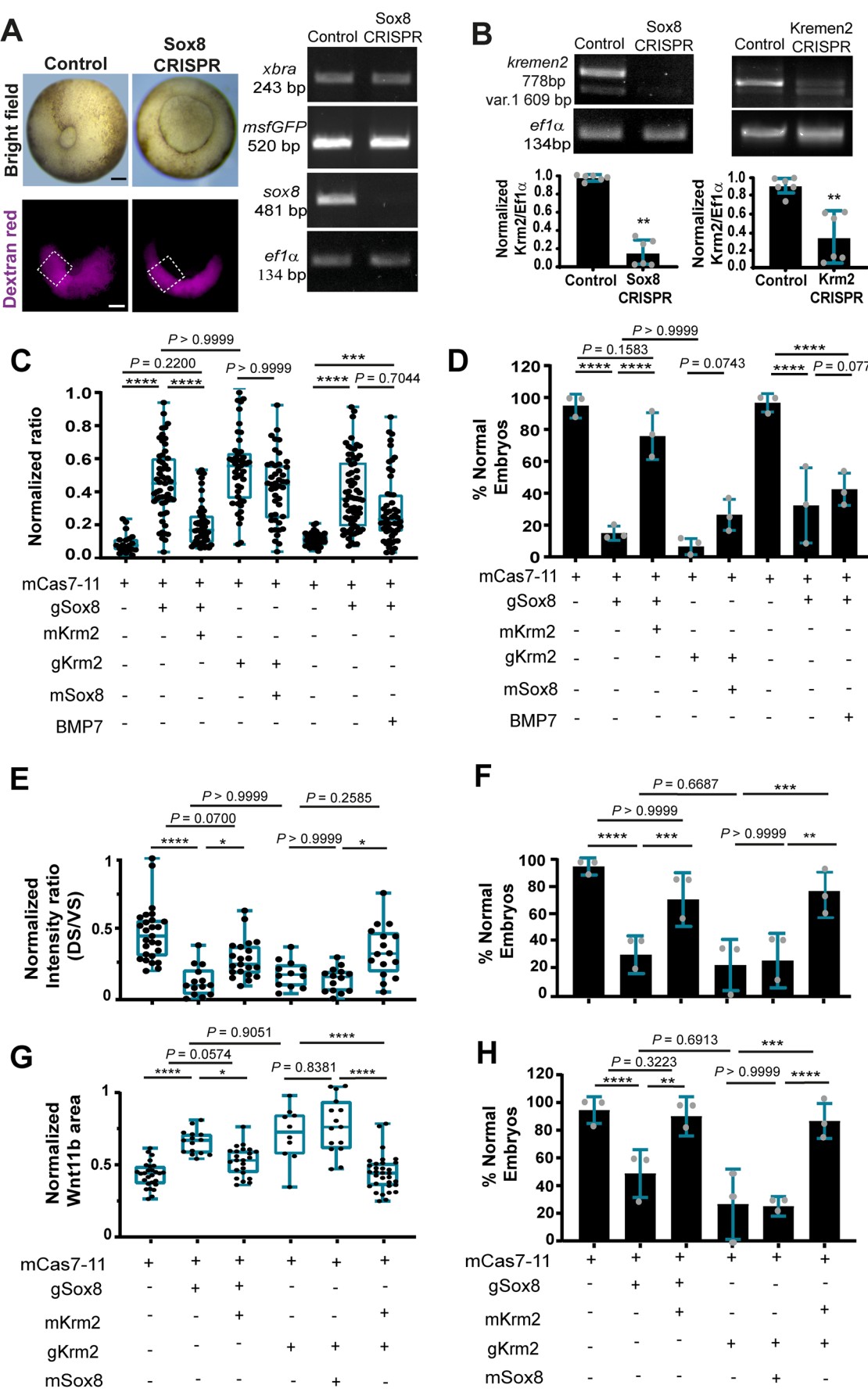

**Figure EV3.** *sox8* and *kremen2* downregulation impairs blastopore closure and alters the *wnt11b* expression pattern.

(A) SqRT–PCR was performed as a quality control for samples sent for RNA-seq. (B) SqRT–PCR and quantification of samples sent for RNA-seq revealed that *kremen2* is downregulated. Statistical significance was assessed by a two-tailed Mann–Whitney test, \*\*$P = 0.0022$. Control for *kremen2* depletion in the context of *kremen2* CRISPR is also shown. Mann–Whitney test, \*\*$P = 0.0022$; Error bars: mean ± SD. (C) Box plot showing the normalized ratio of blastopore area to embryos. Statistical significance was assessed by Kruskal–Wallis multiple comparisons test, \*\*\*\*$P < 0.0001$; \*\*\*$P = 0.0002$ (control vs *sox8* CRISPR + *bmp7* mRNA); ns non-significant: $P = 0.2200$ (control vs *sox8* CRISPR + *kremen2* mRNA); $P > 0.9999$ (*sox8* CRISPR vs kremen2 CRISPR); $P > 0.9999$ (*kremen2* CRISPR vs *kremen2* CRISPR + *sox8* mRNA); $P = 0.7044$ (*sox8* CRISPR vs *sox8* CRISPR + *bmp7* mRNA). (D) Percentage of embryos exhibiting normal blastopore size per condition (embryos displaying values below the maximum control ratio value are classified as normal). Two-sided Fisher's exact test, \*\*\*\*$P < 0.0001$; ns non-significant: $P = 0.0774$ (*sox8* CRISPR vs *sox8* CRISPR + *bmp7* mRNA); $P = 0.1583$ (control vs *sox8* CRISPR + *kremen2*); $P > 0.9999$ (*sox8* CRISPR vs kremen2 CRISPR); $P = 0.0774$ (*sox8* CRISPR vs *sox8* CRISPR + *bmp7* mRNA); $P = 0.0743$ (*kremen2* CRISPR vs *kremen2* CRISPR + *sox8* mRNA). Error bars: mean ± SD; For (C, D): $n = 20$ control Cas7–11; $n = 55$ *sox8* CRISPR; $n = 51$ *sox8* CRISPR + *kremen2* mRNA; $n = 45$ *kremen2* CRISPR; $n = 45$ *kremen2* CRISPR + *sox8* mRNA; $n = 39$ control (for rescue with *bmp7* mRNA); $n = 68$ *sox8* CRISPR (for rescue with *bmp7* mRNA); $n = 51$ *sox8* CRISPR + *bmp7* mRNA. $N = 3$ independent experiments. (E) Box plot showing the *wnt11b* intensity ratio between dorsal and ventral signals in different conditions. Statistical significance was assessed by Kruskal–Wallis multiple comparisons test, \*\*\*\*$P < 0.0001$; \*$P = 0.0184$ (*kremen2* CRISPR + *sox8* mRNA vs *kremen2* CRISPR + *kremen2* mRNA); \*$P = 0.0472$ (*sox8* CRISPR vs *sox8* CRISPR + *kremen2* mRNA); ns non-significant: $P = 0.0700$ (control vs *sox8* CRISPR + *kremen2* mRNA); $P > 0.9999$ (*kremen2* CRISPR + *kremen2 sox8* mRNA); $P > 0.9999$ (*sox8* CRISPR vs kremen2 CRISPR); $P = 0.2585$ (*kremen2* CRISPR vs *kremen2* CRISPR + *kremen2* mRNA). (F) Percentage of embryos displaying a normal *wnt11b* expression pattern is shown; percentage of normal and abnormal embryos in each condition (embryos displaying higher values than the baseline control intensity ratio value are classified as normal). Two-sided Fisher's exact test, \*\*\*\*$P < 0.0001$ (control vs *sox8* CRISPR); \*\*\*$P = 0.0001$ (*sox8* CRISPR vs *sox8* CRISPR + *kremen2* mRNA); \*\*$P = 0.0092$ (*kremen2* CRISPR + *sox8* mRNA vs *kremen2* CRISPR); $P > 0.9999$ (Control vs *sox8* CRISPR + *kremen2* mRNA); $P = 0.6687$ (*sox8* CRISPR vs kremen2 CRISPR); $P > 0.9999$ (*kremen2* CRISPR vs *kremen2* CRISPR + *sox8* mRNA). Error bars: mean ± SD. $N = 3$ independent experiments; $n = 27$ control embryos; $n = 14$ sox8 CRISPR; $n = 21$ sox8 CRISPR + *kremen2* mRNA; $n = 8$ kremen2 CRISPR; $n = 14$ kremen2 CRISPR + sox8 mRNA and $n = 15$ kremen2 CRISPR + kremen2 mRNA; (G) Box plot showing area of the wnt11b intensity in different conditions. Statistical significance was assessed by ordinary one-way ANOVA, \*\*\*\*$P < 0.0001$; \*$P = 0.0147$ (*sox8* CRISPR vs sox8 CRISPR + *kremen2* mRNA); $P = 0.0574$ (control vs *sox8* CRISPR + *kremen2* mRNA); $P = 0.9051$ (*sox8* CRISPR vs kremen2 CRISPR); $P = 0.8381$ (*kremen2* CRISPR vs *kremen2* CRISPR + *sox8* mRNA); (H) Percentage of embryos displaying a normal *wnt11b* expression pattern is shown; percentage of normal embryos in each condition (embryos displaying higher values than the maximum control area value are classified as normal). Two-sided Fisher's exact test, \*\*\*\*$P < 0.0001$; \*\*\*$P = 0.0004$ (*kremen2* CRISPR + *sox8* mRNA); \*\*$P = 0.0041$ (*sox8* CRISPR vs *sox8* CRISPR + *kremen2* mRNA); $P = 0.3223$ (control vs *sox8* CRISPR + *kremen2* mRNA); $P = 0.6913$ (*sox8* CRISPR vs kremen2 CRISPR); $P > 0.9999$ (*kremen2* CRISPR + *kremen2* CRISPR + *sox8* mRNA). Error bars: mean ± SD. $n = 27$ control embryos; $n = 15$ *sox8* CRISPR; $n = 23$ *sox8* CRISPR + *kremen2* mRNA; $n = 10$ *kremen2* CRISPR; $n = 15$ *kremen2* CRISPR + *sox8* mRNA and $n = 32$ *kremen2* CRISPR + *kremen2* mRNA (E–H) $N = 3$ independent experiments; Scale bars: 250 μm. For box plots in (C, E, G), boxes represent the interquartile range (IQR; 25th–75th percentile), with the center line indicating the median and whiskers extending to the minimum and maximum values.

**A** Control

**B** kremen2S CRISPR

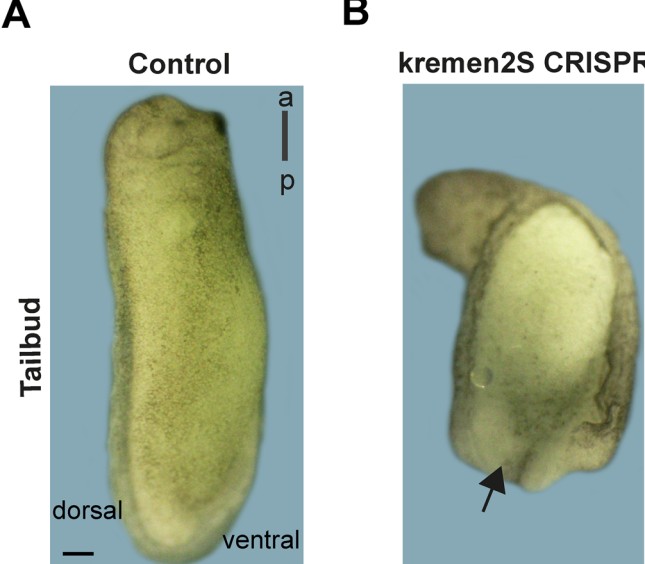

**Figure EV4.  *kremen2* knockdown by Cas7–11 in the ventrolateral mesoderm impairs embryonic development.**

Representative images of (**A**) control embryos and (**B**) *kremen2* CRISPR embryos at the tailbud stage, showing open blastopores (arrows), with shortened embryos on the anteroposterior axis. Scale bars: 250 μm.

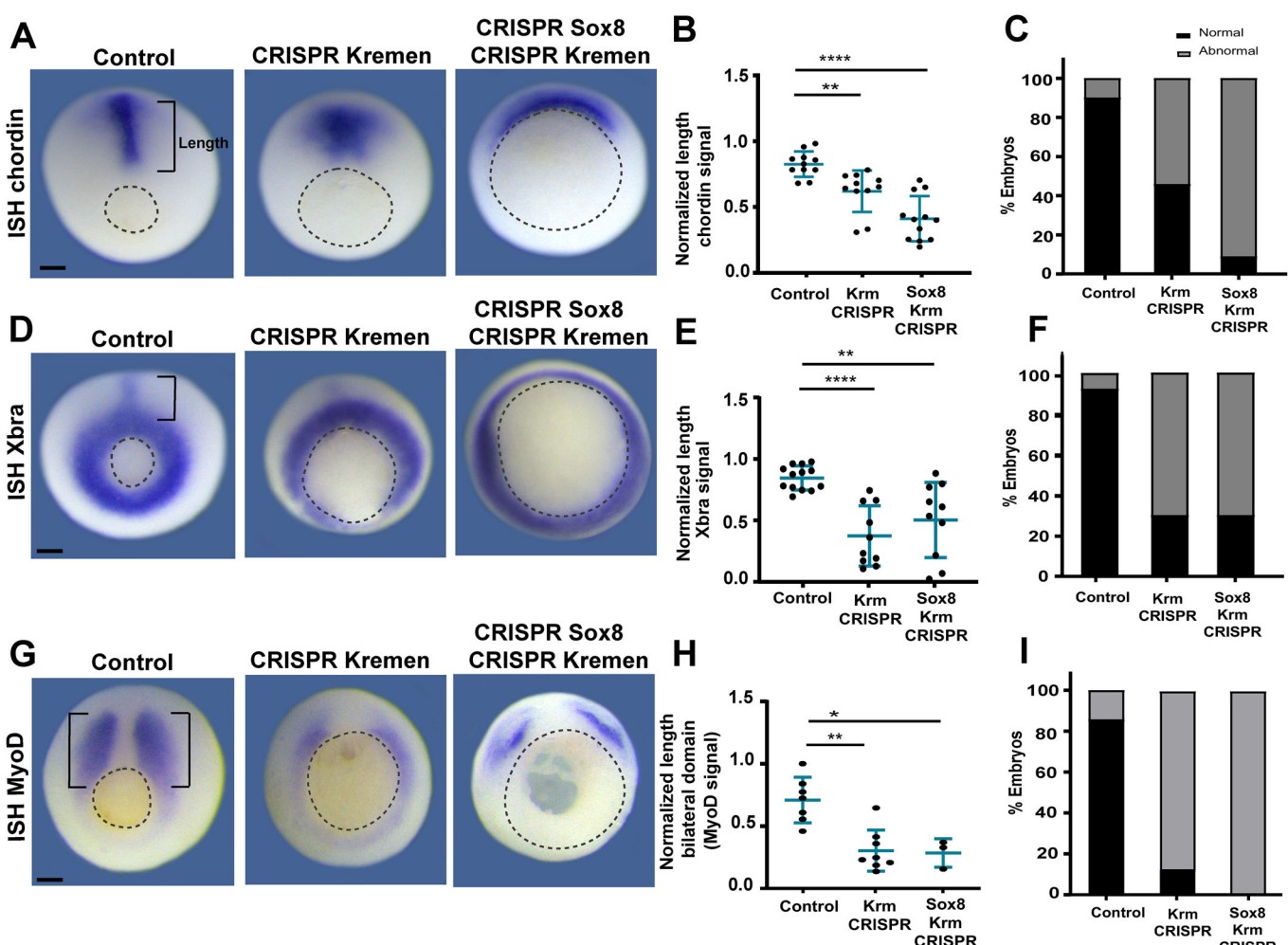

**Figure EV5. *kremen2* CRISPR and *sox8 kremen2* CRISPR double-knockdown disrupt the expression of mesodermal markers.**

(A, D, G) Whole-mount in situ hybridization (ISH) for different markers, *chordin* (A), *brachyury* (D), and *myoD* (G), revealed abnormal expression patterns of these markers in *kremen2* CRISPR and *sox8 kremen2* CRISPR double-knockdown embryos compared with control embryos. Dotted lines delineate the blastopore. (B, E, H) Quantification of marker expression length in the axial mesoderm revealed a significant reduction under CRISPR conditions. (B) Ordinary one-way ANOVA, **$P = 0.0081$; ****$P < 0.0001$. Error bars: mean ± SD. (E) Ordinary one-way ANOVA, **$P = 0.0028$; ****$P < 0.0001$. Error bars: mean ± SD. (H) Two-tailed Mann–Whitney test, **$P = 0.0022$ (control vs *kremen2* CRISPR) *$P = 0.0167$ (control vs *sox8 kremen2* CRISPR). Error bars: mean ± SD. (C, F, I) Percentages of normal and abnormal embryos in each condition (embryos displaying greater values than the baseline control length value are classified as normal). Sample sizes: ISH chordin: $n = 10$ control; $n = 11$ *kremen2* CRISPR; $n = 12$ *sox8 kremen2* CRISPR; *Xbra* ISH: $n = 13$ control; $n = 10$ *kremen2* CRISPR; $n = 10$ *sox8 kremen2* CRISPR ; *ISH MyoD*: $n = 7$ control; $n = 7$ *kremen2* CRISPR; $n = 3$ *sox8 kremen2* CRISPR. Scale bars: 250 µm.

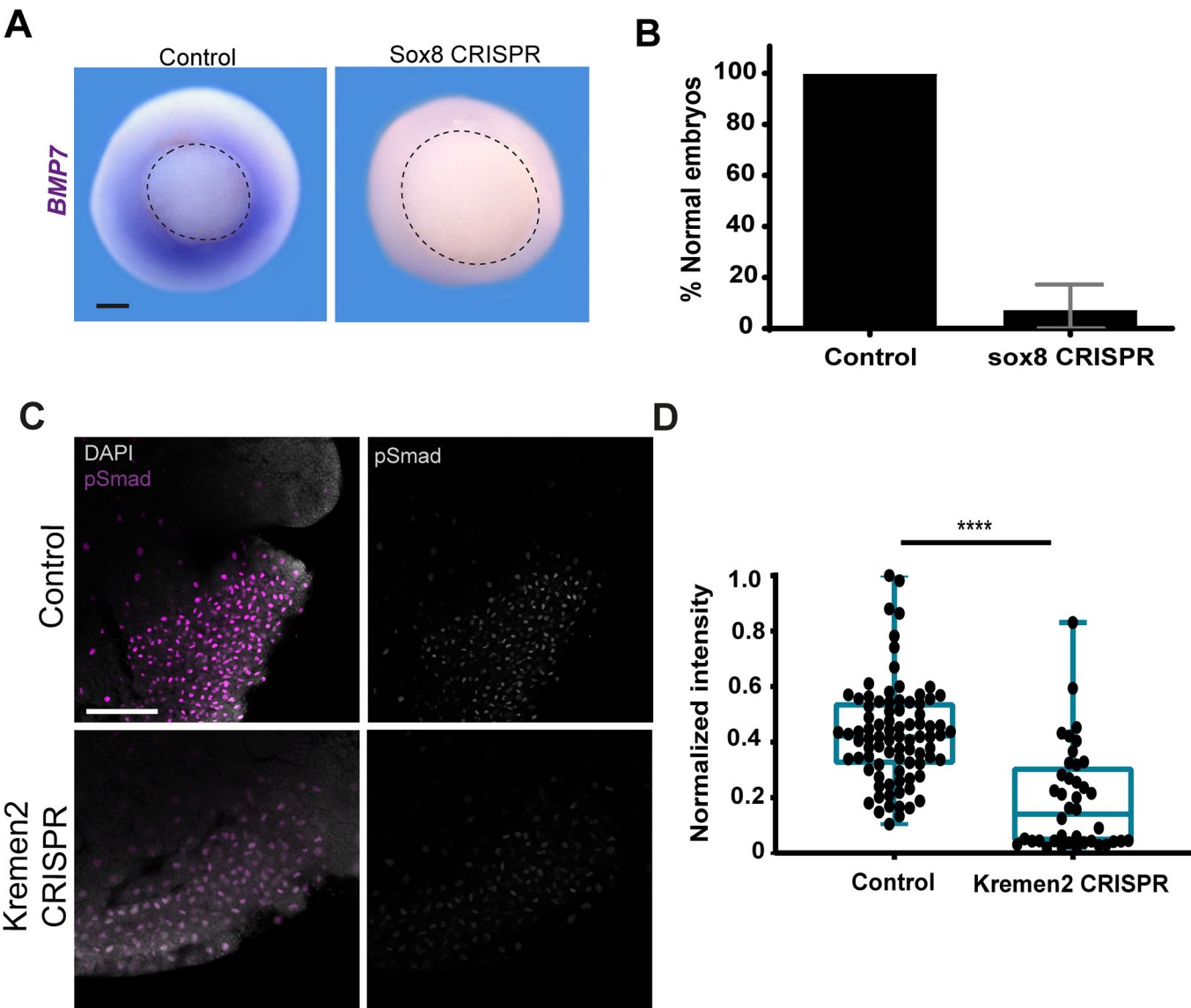

**Figure EV6. *BMP* signaling is perturbed under CRISPR conditions.**

(A) Whole-mount *ISH* for *bmp7* revealed reduced expression in *sox8* CRISPR embryos. Dotted lines delineate the blastopore. Scale bar: 250 μm. (B) Quantification of the percentage of embryos with a normal *bmp7* ISH signal. $N = 2$ independent experiments ($n = 5$ embryos control; $n = 7$ embryos *sox8* CRISPR). (C) pSmad staining levels are decreased in *kremen2* CRISPR conditions compared with controls. (D) Box plot showing the quantification of the nuclear signal in the ventral regions. Boxes represent the interquartile range (IQR; 25th–75th percentile), with the center line indicating the median and whiskers extending to the minimum and maximum values. Statistical significance was assessed by a two-tailed Mann–Whitney test, ****$P < 0.0001$. $n = 9$ control embryos (85 cells); $n = 4$ *kremen2* CRISPR (40 cells). Scale bar: 100 μm.

   