## [Peer Review File · EMBO Reports]

Sox8 is essential for vertebrate gastrulation

Elias Barriga, Sofia Moreira, and Artemis Korovesi

Corresponding author(s): *Elias Barriga (elias.barriga@tu-dresden.de)*

Review Timeline:

Submission Date:	10th Dec 24
Editorial Decision:	7th Jan 25
Editor's Email to Author:	15th Jan 25
Revision Received:	17th Jul 25
Editorial Decision:	13th Aug 25
Revision Received:	18th Sep 25
Accepted:	2nd Oct 25

Editor: Martina Rembold / Esther Schnapp

Transaction Report:

Dear Dr. Barriga

Thank you for the submission of your research manuscript to our journal. Three referees agreed to review your manuscript. So far, we have received two referee reports that are copied below. Given that both referees are in fair agreement that you should be given a chance to revise the manuscript, I would like to ask you to begin revising your study along the lines suggested by the referees.

Please note that this is a preliminary decision made in the interest of time, and that it is subject to change should the third referee offer very strong and convincing reasons for this. As soon as we receive the final report on your manuscript, we will forward it to you as well.

Please address all referee concerns in a complete point-by-point response. Acceptance of the manuscript will depend on a positive outcome of a second round of review. It is EMBO Reports policy to allow a single round of revision only and acceptance or rejection of the manuscript will therefore depend on the completeness of your responses included in the next, final version of the manuscript.

We realize that it is difficult to revise to a specific deadline. In the interest of protecting the conceptual advance provided by the work, we recommend a revision within 3 months (April 7). Please discuss the revision progress ahead of this time with the editor if you require more time to complete the revisions.

I am also happy to discuss the revision further via e-mail or a video call, if you wish.

*******IMPORTANT NOTE:**

We perform an initial quality control of all revised manuscripts before re-review. Your manuscript will FAIL this control and the handling will be delayed IN CASE the following APPLIES:

- 1) A data availability section providing access to data deposited in public databases is missing. If you have not deposited any data, please add a sentence to the data availability section that explains that.
- 2) Your manuscript contains statistics and error bars based on $n=2$. Please use scatter blots in these cases. No statistics should be calculated if $n=2$.

When submitting your revised manuscript, please carefully review the instructions that follow below. Failure to include requested items will delay the evaluation of your revision. *****

- 1) a .docx formatted version of the manuscript text (including legends for main figures, EV figures and tables). Please make sure that the changes are highlighted to be clearly visible.
- 2) individual production quality figure files as .eps, .tif, .jpg (one file per figure). Please download our Figure Preparation Guidelines (figure preparation pdf) from our Author Guidelines pages <https://www.embopress.org/page/journal/14693178/authorguide> for more info on how to prepare your figures.
- 3) a .docx formatted letter INCLUDING the reviewers' reports and your detailed point-by-point responses to their comments. As part of the EMBO Press transparent editorial process, the point-by-point response is part of the Review Process File (RPF), which will be published alongside your paper.
- 4) a complete author checklist, which you can download from our author guidelines (<<https://www.embopress.org/page/journal/14693178/authorguide>>). Please insert information in the checklist that is also reflected in the manuscript. The completed author checklist will also be part of the RPF.
- 5) Please note that all corresponding authors are required to supply an ORCID ID for their name upon submission of a revised manuscript (<<https://orcid.org/>>). Please find instructions on how to link your ORCID ID to your account in our manuscript tracking system in our Author guidelines (<<https://www.embopress.org/page/journal/14693178/authorguide#authorshipguidelines>>)
- 6) We replaced Supplementary Information with Expanded View (EV) Figures and Tables that are collapsible/expandable online.

A maximum of 5 EV Figures can be typeset. EV Figures should be cited as "Figure EV1, Figure EV2" etc... in the text and their respective legends should be included in the main text after the legends of regular figures.

7) Before submitting your revision, primary datasets (and computer code, where appropriate) produced in this study need to be deposited in an appropriate public database (see <<https://www.embopress.org/page/journal/14693178/authorguide#dataavailability>>). Specifically, we would kindly ask you to provide public access to the RNA-seq dataset.

The accession numbers and database should be listed in a formal "Data Availability " section (placed after Materials & Method) that follows the model below (see also <<https://www.embopress.org/page/journal/14693178/authorguide#dataavailability>>). Please note that the Data Availability Section is restricted to new primary data that are part of this study.

Data availability

Additional information on source data and instruction on how to label the files are available <<https://www.embopress.org/page/journal/14693178/authorguide#sourcedata>>.

10) Figure legends and data quantification:

- the name of the statistical test used to generate error bars and P values,
- the number (n) of independent experiments (please specify technical or biological replicates) underlying each data point,
- the nature of the bars and error bars (s.d., s.e.m.)

- If the data are obtained from n {less than or equal to} 5, show the individual data points in addition to the SD or SEM.
- If the data are obtained from n {less than or equal to} 2, use scatter blots showing the individual data points.

11) Our journal encourages inclusion of *data citations in the reference list* to directly cite datasets that were re-used and obtained from public databases. Data citations in the article text are distinct from normal bibliographical citations and should

directly link to the database records from which the data can be accessed. In the main text, data citations are formatted as follows: "Data ref: Smith et al, 2001" or "Data ref: NCBI Sequence Read Archive PRJNA342805, 2017". In the Reference list, data citations must be labeled with "[DATASET]". A data reference must provide the database name, accession number/identifiers and a resolvable link to the landing page from which the data can be accessed at the end of the reference. Further instructions are available at <<https://www.embopress.org/page/journal/14693178/authorguide#referencesformat>>.

12) All Materials and Methods need to be described in the main text using our 'Structured Methods' format. According to this format, the Methods section includes a Reagents and Tools Table (listing key reagents, experimental models, software and relevant equipment and including their sources and relevant identifiers) followed by a Methods and Protocols section describing the methods, ideally using a step-by-step protocol format. The aim is to facilitate adoption of the methodologies across labs. Please download and fill our Reagents and Tools Table template (.docx), which you can find in our author guidelines:

13) As part of the EMBO publication's Transparent Editorial Process, EMBO Reports publishes online a Review Process File to accompany accepted manuscripts. This File will be published in conjunction with your paper and will include the referee reports, your point-by-point response and all pertinent correspondence relating to the manuscript.

Kind regards,

Referee #1:

The segregation of dorsal and ventral fates during early animal development is a key contributor to establishing the body plan and organizing the specific cell movements during gastrulation that lead to the establishment of the three germ layers and all of the organ systems. Although this topic has been under study for nearly a century, molecular insights only became available in the 1990s and the topic has been under enthusiastic investigation ever since. This manuscript contributes to this literature by identifying a new transcriptional regulator in establishing the dorsal-ventral axis - Sox8 - and delineating at least one of its key roles - regulating the ventral expression of a Wnt inhibitor - Kremen2 - that prevents the encroachment of Wnt11 expression into the ventral side of the embryo. First, using RNAseq approaches, the authors identify high levels of sox8 expression in the ventral territory of gastrulae, which was confirmed by ISH analyses. Next, using standard localized microinjection approaches, the levels of sox8 mRNA were depleted on the ventral side using already validated Morpholino oligonucleotides. The authors then introduce, and describe in detail, a novel CRISPR based depletion technique that targets the 3'UTR of endogenous mRNAs (Cas7-11); the validation of this technique is extensively presented in the text and supplemental files, and the results correspond to previously published MO data. In addition to RNAseq, the authors supplement their molecular analyses with a straightforward morphological approach, measuring the size of the blastopore to indicate defects in gastrulation movements; this technique is highly quantifiable and is accepted in the field as an accurate indicator of such defects. Next, RNAseq analyses of Sox8 depleted (via Cas7-11) ventral domains demonstrate upregulation of known dorsalizing factors, particularly in the Wnt pathway, into the ventral domain - which is a major cause of gastrulation defects. Based on these data, the authors pursued analysis of one ventrally-expressed Wnt inhibitor - Kremen2 - and showed by ISH that its expression is reduced in Sox8 crispants. Using ChIPseq approaches, they show that there is a putative Sox8 binding site in the kremen2 promoter that in vivo binds Sox8 and in crispants shows reduced PolII binding, indicating loss of active transcription. Thus, the authors convincingly demonstrate that

Sox8 is required for normal gastrulation and normal dorsal-ventral gene expression, which they go on to further demonstrate in *Kremen2* crispants that phenocopy Sox8 crispants. Finally, epistasis experiments clearly demonstrate that Sox8 acts upstream of *Kremen2*, which acts to down-regulate the expression of *Wnt11b*, a dorsalizing factor.

This work addresses a major question in developmental biology: what are the key factors that regulate the segregation of dorsal and ventral fates during establishment of the body axes? There has been keen interest in this problem for a very long time, and this manuscript identifies a novel transcriptional player and delineates the immediate downstream targets by which it exerts its influence. The experiments are well designed, use a combination of cutting edge and classical approaches and employs impressive numbers of embryos, adequate triplicate experiments and replicas and convincing statistical analyses. The figures are clear, and include diagrams to help the non-expert to understand what was done. In addition, a new method for depleting endogenous mRNA levels in embryos is introduced and described in sufficient detail for readers to use this approach in their own labs.

The mechanisms by which gastrulation is achieved throughout the animal kingdom has been of central interest to the developmental biology community for over 100 years. A key aspect of this process is the segregation of dorsal and ventral fates, leading to the formation of the body plan and specific organs. In vertebrates, the discovery of the Organizer of dorsal fates (in the 1920s) and the subsequent molecular identification of the players on both the dorsal and ventral sides of the embryo (in the 1990s) has spurred great interest in filling in the molecular pathways that regulate this fundamental fate decision. This information has been of interest not only to developmental and molecular biologists, but also to stem cell biologists who are working towards creating organoids for tissue replacement therapies. The reported findings will also be of general interest to the molecular biology community for two reasons: a new mRNA knock-down technique is described in great detail and validated; and a new transcriptional regulator of Wnt signaling is presented. The Supplemental Material provides detailed information on how to perform Cas7-11 mRNA knockdown and its validation for the experiments presented in the main text.

The authors have done an exceptional job of doing the right experiments and providing robust analyses in terms of numbers of independent experiments, replicate trials and large numbers of embryos analyzed. Therefore, the conclusions based on these numbers and the authors' use of appropriate and rigorous statistical analyses are very convincing.

A minor weakness appears in Figure 4, in which perhaps the investigators could confirm the requirement of the PBS1 site in the *krmen2* promoter for Sox8 binding by using a reporter plasmid in vitro that contains a mutated PBS1 site. Alternatively, the authors could at least discuss this caveat.

The authors have provided an extensive reference list that adequately covers the relevant earlier literature on the already reported developmental role of Sox8, its activity in human cancers, and the process of dorsal-ventral fate determination in a variety of experimental model systems.

Minor points to address:

- 1) Figure S1A: The table has 3 enriched terms, not the 2 listed in the figure legend.
- 2) Figure S1B: There are Sox13S expression data available on Xenbase and in McGarey et al (2010) that should be included.
- 3) Figure S2A: please define the orange and white domains on the tailbud embryos.
- 4) Figure S3D: please define the purple stripes.
- 5) Figure S4A: please explain the relevance of including *Xbra* in the analysis, since this will stand independent of the printed manuscript.
- 6) Figure S5A and B: The numbers of independent trials and replicates on the figure and in the legend seem different. It is confusing, so please describe in more detail.
- 7) Figure 2: are control embryos uninjected or injected with Cas7-11 without gRNAs?
- 8) Figure 3H: the explanation of the figure is not adequately explained. What are the dotted lines, for example.
- 9) Figure 4A: please define the arrow in the Sox8 CRISPR image.
- 10) Figure 1 nicely indicates dorsal-ventral axis. Perhaps add this to all embryo ISH for those readers not familiar with *Xenopus* gastrulae?
- 11) There are places at which the English grammar is a bit off, especially in figure legends and methods in Supplemental material - a good proof-reading should be done.
- 12) In several places BMP is all caps and in others just the first letter is capitalized. For example, compare abstract with Significance Statement. Please be consistent. Same comment for Sox8 and other genes.
- 13) Page 2, line 59: "a homeobox factor that"; insert "factor"
- 14) Figure 7: define AP, D, V, and VP in the legend.

Referee #3:

Reviewer Comments to Author

In this study, the authors identify an unexpected role for Sox8 in establishing dorsal-ventral patterning during gastrulation. This is mediated by a dual role of Sox8 in promoting the expression of BMP-related factors along with a well-known Wnt signaling inhibitor, *kremen2*. This, in turn, contributes to restrict the expression of *Wnt11b* to ventral tissues. Given the importance of

establishing a proper DV axis in gastrulation, this study addresses an important question in biology. The paper is well-written, the data presented is of high quality and convincing, pointing towards a new function of Sox8 during gastrulation.

This work is of excellent quality, however, adding further analyses would, in my view, strengthen the key findings on the mechanism of action of Sox8 and clarify more broadly the mechanisms underlying AP patterning.

1. Do other organizer genes (e.g. *gsc*, *chordin*) show similarly defective localization upon Sox8 or *Kremen2* knockdown?
2. A more quantitative description of the spatial expansion of dorsal tissues is necessary for both Sox8 and *Kremen2* knockdowns and to compare the severity of the phenotypes between Sox8, *Kremen2* and the rescue experiments (see my next point).
3. It is unclear how much the role of Sox8 in promoting the expression of BMP-related factors is required for its function in DV patterning, given the observations with the *kremen2* mRNA rescue. Is the role of Sox8 in regulating BMP signaling therefore a more indirect consequence from regulation of Wnt signaling or rather a direct function, acting in parallel with the regulation of Wnt? To test this, a more quantitative analysis of the dorsal expansion phenotypes observed upon Sox8 and *Kremen* knockdown as compared to their combined knockdown is necessary.
4. It is unclear how the defects in blastopore closure are related with those in DV patterning. Is this a delay or blastopore closure fails completely upon Sox8 and *Kremen2* knockdown? How specific is this phenotype? Are there also other more specific defects in the morphogenesis of ventral tissues in Sox8 knockdown, for instance in convergence movements.
5. How does the severity of the Sox8 phenotype compare with that of the other well-known ventral determinant *Ventx1*? This should be discussed in the discussion. Since Sox8 expression is restricted to ventrolateral mesoderm, its function in DV patterning is likely to be more associated with the maintenance rather than establishment of the axis.
6. An important advance of this manuscript is the use of *Crispr Cas7-11* in vivo. As the authors point out, this can be very useful in other systems and I appreciated the detailed explanation of how to implement this method provided in the supplementary materials. This should be further expanded, and the key points should be brought into the main. Can the authors also comment more on off-targets or toxicity associated with the injection of these constructs? Have the authors also tested commercially-synthesized gRNAs and could they comment on this point as well?
7. While the authors provide validation for the Sox8 knockdown phenotype with a well-validated morpholino and show an effective rescue with mRNA, it would be very valuable to also compare these phenotypes with that of F0 *crispants*, which have become established in *xenopus* over recent years. This would provide a first comparison of the 3 different methods and resulting phenotypes in this species.

Date: 15th Jan 25 10:00:21

Last Sent: 15th Jan 25 10:00:21

Triggered By: Martina Rembold

From: m.rembold@emboreports.org

To: elias.barriga@tu-dresden.de

Subject: EMBOR: Obtained the third Referee Report for:

Message: EMBOR-2024-60964-T

Dear Elias,

We have meanwhile received the missing report from referee #2. For completeness, I copy all three reports below my signature.

As you will see, referee #2 considers the findings potentially very interesting but has also a number of concerns regarding the proposed role of Sox8 in functioning as ventral determinant by antagonizing Wnt signaling. I think that the concerns raised are all relevant and should be addressed experimentally. The report is very constructive and while addressing them will require quite some additional work, the experiments seem feasible and I feel that these will help to strengthen your indeed interesting observations.

As discussed earlier, we can extend the revision deadline if this is needed. I suggest to get in contact closer to the proposed resubmission date.

Kind regards,

Martina

Referee #1

The segregation of dorsal and ventral fates during early animal development is a key contributor to establishing the body plan and organizing the specific cell movements during gastrulation that lead to the establishment of the three germ layers and all of the organ systems. Although this topic has been under study for nearly a century, molecular insights only became available in the 1990s and the topic has been under enthusiastic investigation ever since. This manuscript contributes to this literature by identifying a new transcriptional regulator in establishing the dorsal-ventral axis - Sox8 - and delineating at least one of its key roles - regulating the ventral expression of a Wnt inhibitor - Kremen2 - that prevents the encroachment of Wnt11 expression into the ventral side of the embryo. First, using RNAseq approaches, the authors identify high levels of sox8 expression in the ventral territory of gastrulae, which was confirmed by ISH analyses. Next, using standard localized microinjection approaches, the levels of sox8 mRNA were depleted on the ventral side using already validated Morpholino oligonucleotides. The authors then introduce, and describe in detail, a novel CRISPR based depletion technique that targets the 3'UTR of endogenous mRNAs (Cas7-11); the validation of this technique is extensively presented in the text and supplemental files, and the results correspond to previously published MO data. In addition to RNAseq, the authors supplement their molecular analyses with a straightforward morphological approach, measuring the size of the blastopore to indicate defects in gastrulation movements; this technique is highly quantifiable and is accepted in the field as an accurate indicator of such defects. Next, RNAseq analyses of Sox8 depleted (via Cas7-11) ventral domains demonstrate upregulation of known dorsalizing factors, particularly in the Wnt pathway, into the ventral domain - which is a major cause of gastrulation defects. Based on these data, the authors pursued analysis of one ventrally-expressed Wnt inhibitor - Kremen2 - and showed by ISH that its expression is reduced in Sox8 crispants. Using ChIPseq approaches, they show that there is a putative Sox8 binding site in the kremen2 promoter that in vivo binds Sox8 and in crispants shows reduced PolII binding, indicating loss of active transcription. Thus, the authors convincingly demonstrate that Sox8 is required for normal gastrulation and normal dorsal-ventral gene expression, which they go on to further demonstrate in Kremen2 crispants that phenocopy Sox8 crispants. Finally, epistasis experiments clearly

demonstrate that Sox8 acts upstream of Kremen2, which acts to down-regulate the expression of Wnt11b, a dorsalizing factor.

This work addresses a major question in developmental biology: what are the key factors that regulate the segregation of dorsal and ventral fates during establishment of the body axes? There has been keen interest in this problem for a very long time, and this manuscript identifies a novel transcriptional player and delineates the immediate downstream targets by which it exerts its influence. The experiments are well designed, use a combination of cutting edge and classical approaches and employs impressive numbers of embryos, adequate triplicate experiments and replicas and convincing statistical analyses. The figures are clear, and include diagrams to help the non-expert to understand what was done. In addition, a new method for depleting endogenous mRNA levels in embryos is introduced and described in sufficient detail for readers to use this approach in their own labs.

The mechanisms by which gastrulation is achieved throughout the animal kingdom has been of central interest to the developmental biology community for over 100 years. A key aspect of this process is the segregation of dorsal and ventral fates, leading to the formation of the body plan and specific organs. In vertebrates, the discovery of the Organizer of dorsal fates (in the 1920s) and the subsequent molecular identification of the players on both the dorsal and ventral sides of the embryo (in the 1990s) has spurred great interest in filling in the molecular pathways that regulate this fundamental fate decision. This information has been of interest not only to developmental and molecular biologists, but also to stem cell biologists who are working towards creating organoids for tissue replacement therapies. The reported findings will also be of general interest to the molecular biology community for two reasons: a new mRNA knock-down technique is described in great detail and validated; and a new transcriptional regulator of Wnt signaling is presented. The Supplemental Material provides detailed information on how to perform Cas7-11 mRNA knockdown and its validation for the experiments presented in the main text.

The authors have done an exceptional job of doing the right experiments and

providing robust analyses in terms of numbers of independent experiments, replicate trials and large numbers of embryos analyzed. Therefore, the conclusions based on these numbers and the authors' use of appropriate and rigorous statistical analyses are very convincing.

A minor weakness appears in Figure 4, in which perhaps the investigators could confirm the requirement of the PBS1 site in the *krmn2* promoter for Sox8 binding by using a reporter plasmid in vitro that contains a mutated PBS1 site. Alternatively, the authors could at least discuss this caveat.

The authors have provided an extensive reference list that adequately covers the relevant earlier literature on the already reported developmental role of Sox8, its activity in human cancers, and the process of dorsal-ventral fate determination in a variety of experimental model systems.

Minor points to address:

- 1) Figure S1A: The table has 3 enriched terms, not the 2 listed in the figure legend.
- 2) Figure S1B: There are Sox13S expression data available on Xenbase and in McGarey et al (2010) that should be included.
- 3) Figure S2A: please define the orange and white domains on the tailbud embryos.
- 4) Figure S3D: please define the purple stripes.
- 5) Figure S4A: please explain the relevance of including Xbra in the analysis, since this will stand independent of the printed manuscript.
- 6) Figure S5A and B: The numbers of independent trials and replicates on the figure and in the legend seem different. It is confusing, so please describe in more detail.
- 7) Figure 2: are control embryos uninjected or injected with Cas7-11 without gRNAs?
- 8) Figure 3H: the explanation of the figure is not adequately explained. What are the dotted lines, for example.
- 9) Figure 4A: please define the arrow in the Sox8 CRISPR image.
- 10) Figure 1 nicely indicates dorsal-ventral axis. Perhaps add this to all embryo ISH for those readers not familiar with *Xenopus* gastrulae?
- 11) There are places at which the English grammar is a bit off, especially in figure legends and methods in Supplemental material - a good proof-

reading should be done.

12) In several places BMP is all caps and in others just the first letter is capitalized. For example, compare abstract with Significance Statement. Please be consistent. Same comment for Sox8 and other genes.

13) Page 2, line 59: "a homeobox factor that"; insert "factor"

14) Figure 7: define AP, D, V, and VP in the legend.

Referee #2

This manuscript reports a study on the function of the transcription factor Sox8 during late *Xenopus* gastrulation. Sox8 has been so far involved in defining the dorsal field for neural crest cells, here the authors focus on a potential earlier role on the ventral side. Note that this ventral expression pattern is visible in publicly available in situ images in xenbase, but here the authors have determined it by looking at gene expression ventral enrichment through transcriptomics.

As for the function, they report LOF by RNA depletion using CRISPR Cas7-11 technology, which they validate by comparing with classical antisense morpholinos. They observe a strong phenotype of failure of blastopore closure.

At the molecular level, they show a striking effect of Sox8 depletion on the transcriptome of the ventral region, with downregulation of components related to BMP signalling and upregulation of genes characteristic of the dorsal organizer, such as Gsc, chordin or noggin. This latter upregulation is quite impressive and rather unexpected. Sox8 depletion also leads to a milder upregulation of Wnt11b and a Wnt receptor, Frz7, and downregulation of a Wnt inhibitor, Kremen2.

The authors focus their analysis on these latter genes, show that Kremen2 is a direct target of Sox8, and that both Sox8 and Kremen2 LOF lead to a ventral expansion of Wnt11b. They interpret the data as evidence that Sox8 functions as a ventral determinant antagonizing Wnt signalling and thus indirectly favouring the ventral fate conferred by BMP signalling.

The report of a strong gastrulation phenotype upon LOF of a ventrally-enriched Sox8 is potentially quite interesting. The upregulation of dorsal organizer genes is also intriguing.

However, as it stands, this study is still preliminary/incomplete. It would require:

(A) A proper characterization of the gastrulation phenotype. It is impossible at the moment to evaluate the actual role of Sox8 on gastrulation. Showing simply failed blastopore closure is by far not sufficient.

(B) A thorough reinterpretation of the data and of Sox8 function. There are indeed major issues, as explained below.

The use of Cas7-11 seems to work very well, and will be a nice contribution to the field, although per se the report of this innovation is not sufficient to palliate for the current gaps and inconsistencies in the mechanistic analysis.

Here first some comments, followed by specific suggestions

1) Issues with the description of patterning during gastrulation:

I will try to briefly explain why the description of patterning during gastrulation is confusing, and actually incorrect. There is abundant literature and reviews on the subject.

a) Wnt signalling has a dorsalizing activity early on, due to maternal Wnt11/5a activating early canonical b-catenin. But this activity ceases at the onset of gastrulation.

b) During gastrulation, two main sources of canonical Wnt activity are: ventral Wnt8, which inhibits in particular the most dorsal axial fate (notochord), and a Wnt (Wnt11?) POSTERIOR activity, located at the late gastrula dorsal lip and contributing to set the AP axis both in the dorsal mesoderm and the prospective neuroderme. Posteriorizing Wnts are inhibited by more anteriorly located Wnt inhibitors (including Kremen).

c) This latter posterior activity is clearly the one repressed by Sox8 on the ventral lip (NOT globally the ventral side!)

d) Once gastrulation has started, the most dorsal trunk region (notochord) is formed through INHIBITION of Wnt (zygotic Wnt8), not activation. See abundant literature/reviews about this process.

e) In parallel, there is also a non-canonical Wnt11b/Frz7 dorsal activity, but this pathway is implicated in cell migration, not in DV patterning.

f) Thus, one is here dealing with something else than simply DV patterning. The model as drawn in figure 7 is simply wrong, starting with the various positions of the gene expressions.

2) Issues with LOF phenotype:

a) A blastopore closure phenotype is NOT a typical phenotype for defect in DV patterning. It has been known for decades that both dorsalized and ventralized embryos can complete "gastrulation", i.e. the blastopore closes. DV defects are much more about the presence/extent of dorsal and ventral structures.

b) One is probably dealing here with a different effect. One possibility is an effect directly on the blastopore closure, which is a specific morphogenetic movement (see elife paper by Ray Keller and colleagues). Actually, the possibility that this process indeed relies on signals located in the lip itself would be quite exciting per se.

I hope that these brief points will help the authors revisiting their interpretation.

Suggestion for major points to address:

- **Characterization of the gastrula phenotype:** Provide information about the internal structure of the late gastrula embryo.
- Map by in situ hybridization key markers of the various mesoderm regions (dorsal and ventral, but also anterior and posterior, i.e. prechordal and axial/paraxial).
- A late morphological phenotypic analysis could also help to evaluate whether there is also an impact on axis elongation. This should be already be visible at late neurula stage, even in embryos that have failed to close their blastopore.
- **The analysis of the molecular impact of Sox8 (and Kremen2) LOF is also incomplete:** That Kremen2 is a downstream target of Sox8 is well

demonstrated, no problem with that.

- However, Sox2 and Kremen2 LOF appear to affect Wnt11b EXPRESSION, while Kremen2 is not a repressor, but an inhibitor of Wnt activity. One would expect evidence for an effect on the Wnt signalling pathway activity.

- Related: What is exactly the Wnt activity one is talking about? Canonical b-catenin-dependent pathway, PCP pathway, non-canonical? Authors should experimentally address this question, which is here essential in reporting a new "patterning" activity during gastrulation.

- The model proposed suggests that the Sox8-Kremen2-Wnt11b "pathway" controls the dorsal organizer components (Goosecoid, xNot) and on BMP-dependent patterning. Yet there are no experimental data supporting this part of the model, beside RNAseq data. **Is the actual DV pattern of BMP activity affected?**

Some minimal experimental results, e.g. P-Smad1 staining for BMP signalling is expected. See above about characterization of the gastrulation phenotype.

Additional points:

- None of the transcriptomic data show any indications about variability/reproducibility/confidence: No stats, no SD???

- Goosecoid and Not mark distinct regions of the mesoderm.

Referee #3

Reviewer Comments to Author

In this study, the authors identify an unexpected role for Sox8 in establishing dorsal-ventral patterning during gastrulation. This is mediated by a dual role of Sox8 in promoting the expression of BMP-related factors along with a well-known Wnt signaling inhibitor, kremen2. This, in turn, contributes to restrict the expression of Wnt11b to ventral tissues. Given the importance of establishing a proper DV axis in gastrulation, this study addresses an important question in biology. The paper is well-written, the data presented is of high quality and convincing, pointing towards a new function of Sox8 during gastrulation.

This work is of excellent quality, however, adding further analyses would, in my view, strengthen the key findings on the mechanism of action of Sox8 and clarify more broadly the mechanisms underlying AP patterning.

1. Do other organizer genes (e.g. *gsc*, *chordin*) show similarly defective localization upon Sox8 or *Kremen2* knockdown?

2. A more quantitative description of the spatial expansion of dorsal tissues is necessary for both Sox8 and *Kremen2* knockdowns and to compare the severity of the phenotypes between Sox8, *Kremen2* and the rescue experiments (see my next point).

3. It is unclear how much the role of Sox8 in promoting the expression of BMP-related factors is required for its function in DV patterning, given the observations with the *kremen2* mRNA rescue. Is the role of Sox8 in regulating BMP signaling therefore a more indirect consequence from regulation of Wnt signaling or rather a direct function, acting in parallel with the regulation of Wnt? To test this, a more quantitative analysis of the dorsal expansion phenotypes observed upon Sox8 and *Kremen* knockdown as compared to their combined knockdown is necessary.

4. It is unclear how the defects in blastopore closure are related with those in DV patterning. Is this a delay or blastopore closure fails completely upon Sox8 and *Kremen2* knockdown? How specific is this phenotype? Are there also other more specific defects in the morphogenesis of ventral tissues in Sox8 knockdown, for instance in convergence movements.

5. How does the severity of the Sox8 phenotype compare with that of the other well-known ventral determinant *Ventx1*? This should be discussed in the discussion. Since Sox8 expression is restricted to ventrolateral mesoderm, its function in DV patterning is likely to be more associated with the maintenance rather than establishment of the axis.

6. An important advance of this manuscript is the use of Crispr Cas7-11 in vivo. As the authors point out, this can be very useful in other systems and I appreciated the detailed explanation of how to implement this method provided in the supplementary materials. This should be further expanded,

and the key points should be brought into the main. Can the authors also comment more on off-targets or toxicity associated with the injection of these constructs? Have the authors also tested commercially-synthesized gRNAs and could they comment on this point as well?

7. While the authors provide validation for the Sox8 knockdown phenotype with a well-validated morpholino and show an effective rescue with mRNA, it would be very valuable to also compare these phenotypes with that of F0 crispants, which have become established in xenopus over recent years. This would provide a first comparison of the 3 different methods and resulting phenotypes in this species.

Point-by-point response to reviewers

General comment

We truly thank the reviewers for their thoughtful and constructive feedback, which we believe has significantly improved the quality and clarity of our new manuscript. We experimentally addressed most of the points raised by the reviewers and included analyses and discussions that complement our previous and new observations. Below, we summarize (in blue) the key revisions and improvements made in response to the reviewers' comments:

1. **Expanded phenotypic characterization:** We now present a thorough characterization of the gastrulation phenotype. This includes a more detailed analysis of the internal structure of the gastrula, live imaging and trajectory analysis of involuting ventral cells movements and new *in situ* hybridization (ISH) for mesoderm markers. In addition to this we analysed Wnt and BMP activities by immunofluorescence in ventral tissues. These experiments offer a more complete understanding of the observed gastrulation defects.
2. **Mechanistic Insights into BMP Regulation:** We performed co-injections and observed that *bmp* mRNA does not rescue the defects caused by *sox8* knockdown, whereas *kremen2* mRNA does. This complement our evidence showing that Sox8 directly regulates *kremen2* expression and aligns with published data showing that, although BMP is required for proper gastrulation, BMP loss-of-function phenotypes do not recapitulate those observed upon *sox8* or *kremen2* knockdowns. Thus, our new results suggest that the observed BMP downregulation is likely a secondary effect of Wnt signalling modulation via the Kremen2, rather than a primary driver of blastopore closure defects. We carefully rephrased this, as we believe that a dedicated project would ultimately dissect these possibilities.

Thus, we have reconsidered our conclusions and propose that Sox8 contribute to gastrulation by modulating anterior-posterior patterning via Kremen2 expression and biasing Wnt activity to dorsal tissues.

Additional note: we add here the token to access our sequencing data in GEO (dataset GSE302963)
Token: [REDACTED]

Specific comments:

Please find below our point-by-point response letter to the referees' comments. For clarity, our answers are written in blue characters and referee's comments are shown in black.

Referee #1:

The segregation of dorsal and ventral fates during early animal development is a key contributor to establishing the body plan and organizing the specific cell movements during gastrulation that lead to the establishment of the three germ layers and all of the organ systems. Although this topic has been under study for nearly a century, molecular insights only became available in the 1990s and the topic has been under enthusiastic investigation ever since. This manuscript contributes to this literature by identifying a new transcriptional regulator in establishing the dorsal-ventral axis - Sox8 - and delineating

at least one of its key roles - regulating the ventral expression of a Wnt inhibitor - Kremen2 - that prevents the encroachment of Wnt11 expression into the ventral side of the embryo. First, using RNAseq approaches, the authors identify high levels of sox8 expression in the ventral territory of gastrulae, which was confirmed by ISH analyses. Next, using standard localized microinjection approaches, the levels of sox8 mRNA were depleted on the ventral side using already validated Morpholino oligonucleotides. The authors then introduce, and describe in detail, a novel CRISPR based depletion technique that targets the 3'UTR of endogenous mRNAs (Cas7-11); the validation of this technique is extensively presented in the text and supplemental files, and the results correspond to previously published MO data. In addition to RNAseq, the authors supplement their molecular analyses with a straightforward morphological approach, measuring the size of the blastopore to indicate defects in gastrulation movements; this technique is highly quantifiable and is accepted in the field as an accurate indicator of such defects. Next, RNAseq analyses of Sox8 depleted (via Cas7-11) ventral domains demonstrate upregulation of known dorsalizing factors, particularly in the Wnt pathway, into the ventral domain - which is a major cause of gastrulation defects. Based on these data, the authors pursued analysis of one ventrally-expressed Wnt inhibitor - Kremen2 - and showed by ISH that its expression is reduced in Sox8 crispants. Using ChIPseq approaches, they show that there is a putative Sox8 binding site in the kremen2 promoter that in vivo binds Sox8 and in crispants shows reduced PolII binding, indicating loss of active transcription. Thus, the authors convincingly demonstrate that Sox8 is required for normal gastrulation and normal dorsal-ventral gene expression, which they go on to further demonstrate in Kremen2 crispants that phenocopy Sox8 crispants. Finally, epistasis experiments clearly demonstrate that Sox8 acts upstream of Kremen2, which acts to down-regulate the expression of Wnt11b, a dorsalizing factor.

This work addresses a major question in developmental biology: what are the key factors that regulate the segregation of dorsal and ventral fates during establishment of the body axes? There has been keen interest in this problem for a very long time, and this manuscript identifies a novel transcriptional player and delineates the immediate downstream targets by which it exerts its influence. The experiments are well designed, use a combination of cutting edge and classical approaches and employs impressive numbers of embryos, adequate triplicate experiments and replicas and convincing statistical analyses. The figures are clear, and include diagrams to help the non-expert to understand what was done. In addition, a new method for depleting endogenous mRNA levels in embryos is introduced and described in sufficient detail for readers to use this approach in their own labs.

The mechanisms by which gastrulation is achieved throughout the animal kingdom has been of central interest to the developmental biology community for over 100 years. A key aspect of this process is the segregation of dorsal and ventral fates, leading to the formation of the body plan and specific organs. In vertebrates, the discovery of the Organizer of dorsal fates (in the 1920s) and the subsequent molecular identification of the players on both the dorsal and ventral sides of the embryo (in the 1990s) has spurred great interest in filling in the molecular pathways that regulate this fundamental fate decision. This information has been of interest not only to developmental and molecular biologists, but also to stem cell biologists who are working towards creating organoids for tissue replacement therapies. The reported findings will also be of general interest to the molecular biology community for two reasons: a new mRNA knock-down technique is described in great detail and validated; and a new transcriptional

regulator of Wnt signaling is presented. The Supplemental Material provides detailed information on how to perform Cas7-11 mRNA knockdown and its validation for the experiments presented in the main text.

The authors have done an exceptional job of doing the right experiments and providing robust analyses in terms of numbers of independent experiments, replicate trials and large numbers of embryos analyzed. Therefore, the conclusions based on these numbers and the authors' use of appropriate and rigorous statistical analyses are very convincing.

A: We greatly appreciate the reviewer's positive comments regarding the significance of our findings as well as the positive comments on the robustness of our data and design of our experiments. Below we detail our response to their comments:

Point 1: A minor weakness appears in Figure 4, in which perhaps the investigators could confirm the requirement of the PBS1 site in the *kremen2* promoter for Sox8 binding by using a reporter plasmid in vitro that contains a mutated PBS1 site. Alternatively, the authors could at least discuss this caveat.

A: We thank the reviewer for highlighting this comment. The experiments are interesting as the presence of a wildtype exogenous PBS1 sequence would report transcriptional activation while the mutated exogenous PBS1 would elicit no signal, suggesting that Sox8 binding site is required to recruit transcriptional machineries. We did not perform this precise experiment, but our analyses offer a complementary approach to it. Briefly, in addition to our experiments that show Sox8-GFP enrichment in the *kremen2* PBS1 site, we generate ChIP extracts from wild-type as well as from Sox8 knockdown samples. These experiments show that Polymerase II is only enriched in the wild-type condition, when compared to the extract obtained from Sox8 knockdown embryos. In addition to this, we now added a new ChIP-PCR experiment in which the binding of Sox8 to a second putative site was not observed (Figure 1 Reviewer 1). Since this putative site has similar but not identical sequence with PBS1, our new results confirm the preference of Sox8 to bind into *kremen2* PBS1 and the recruitment of polymerase as a readout of transcriptional activity to this binding site.

We thank again the reviewer and apologise for

Figure 1 Rev 01: ChIP-PCR analysis for another putative binding site do not show a significant increase of Sox8-GFP when comparing with control GFP, suggesting that binding of PBS1 site to Sox8-GFP might be specific. Note that while PBS1 sequence is AACATA, this second site sequence is TACAATG.

not including the precise suggested experiment. We have added now the new data (Appendix Figure S3 B) and discuss this point in the revised manuscript (lines 189).

Minor points to address:

1) Figure S1A: The table has 3 enriched terms, not the 2 listed in the figure legend.

A: Figure S1A (now Appendix Figure S1A) legend was amended accordingly.

2) Figure S1B: There are Sox13S expression data available on Xenbase and in McGarey et al (2010) that should be included.

A: We added this reference in the revised version (line 94) and highlighted in the Appendix Figure 1B.

3) Figure S2A: please define the orange and white domains on the tailbud embryos.

A: According to the assessment of our conclusions, we have removed these domains in our new figure version.

4) Figure S3D: please define the purple stripes.

A: In figure S3D (now Figure EV2D), "The purple stripes represent target injections in the neural crest" (description was included in the figure legend).

5) Figure S4A: please explain the relevance of including Xbra in the analysis, since this will stand independent of the printed manuscript.

A: We chose to include *xbra* (*brachyury*) because it is a general mesodermal marker. As such, we used it as a normalization control between conditions. Since *xbra* expression is not affected it also indicates that our defects are not related to general mesoderm induction. Note that another general mesoderm marker could have also work.

6) Figure S5A and B: The numbers of independent trials and replicates on the figure and in the legend seem different. It is confusing, so please describe in more detail.

A: We have now corrected the legend in new Figure 5 and Appendix Figure S3. We also added a third ChIP-PCR repeat for Sox8-GFP vs GFP experiment.

7) Figure 2: are control embryos uninjected or injected with Cas7-11 without gRNAs?

A: We apologize for not clarifying this earlier. We used controls injected with Cas7-11+DxRed without gRNAs. It is now clarified in the new version.

8) Figure 3H: the explanation of the figure is not adequately explained. What are the dotted lines, for example.

A: Done: "Schematic illustration of the observed *wnt11b* expression predominant phenotypes in each condition. Dashed lines represent the embryo's equatorial region, dividing the dorsal (above) and ventral (below) regions. The shaded purple area indicates *wnt11b* expression."

9) Figure 4A: please define the arrow in the Sox8 CRISPR image.

A: Done: "Arrow points out to the reduction of *kremen2* signal in the ventral region."

10) Figure 1 nicely indicates dorsal-ventral axis. Perhaps add this to all embryo ISH for those readers not familiar with *Xenopus gastrulae*?

A: Done. We added the scheme to figures where we think this might be more helpful for the readers (for instance, in Figure 5A).

11) There are places at which the English grammar is a bit off, especially in figure legends and methods in Supplemental material - a good proof-reading should be done.

A: We have proofread the text to improve its clarity.

12) In several places BMP is all caps and in others just the first letter is capitalized. For example, compare abstract with Significance Statement. Please be consistent. Same comment for Sox8 and other genes.

A: Done.

13) Page 2, line 59: "a homeobox factor that"; insert "factor"

A: We removed this sentence from the new introduction.

14) Figure 7: define AP, D, V, and VP in the legend.

A: We thank the reviewer for pointing out this aspect. Other comments lead us to remove this figure.

Referee #2 This manuscript reports a study on the function of the transcription factor Sox8 during late *Xenopus* gastrulation. Sox8 has been so far involved in defining the dorsal field for neural crest cells, here the authors focus on a potential earlier role on the ventral side. Note that this ventral expression pattern is visible in publicly available in situ images in xenbase, but here the authors have determined it by looking at gene expression ventral enrichment through transcriptomics. As for the function, they report LOF by RNA depletion using CRISPR Cas7-11 technology, which they validate by comparing with classical antisense morpholinos. They observe a strong phenotype of failure of blastopore closure. At the molecular level, they show a striking effect of Sox8 depletion on the transcriptome of the ventral region, with downregulation of components related to BMP signalling and upregulation of genes characteristic of the dorsal organizer, such as *Gsc*, *chordin* or *noggin*. This latter upregulation is quite impressive and rather unexpected. Sox8 depletion also leads to a milder upregulation of *Wnt11b* and a *Wnt* receptor, *Frz7*, and downregulation of a *Wnt* inhibitor, *Kremen2*. The authors focus their analysis on these latter genes, show that *Kremen2* is a direct target of Sox8, and that both Sox8 and *Kremen2* LOF lead to a ventral expansion of *Wnt11b*. They interpret the data as evidence that Sox8 functions as a ventral determinant antagonizing *Wnt* signalling and thus indirectly favouring the ventral fate conferred by BMP signalling.

The report of a strong gastrulation phenotype upon LOF of a ventrally-enriched Sox8 is potentially quite interesting. The upregulation of dorsal organizer genes is also intriguing.

However, as it stands, this study is still preliminary/incomplete. It would require:

(A) A proper characterization of the gastrulation phenotype. It is impossible at the moment to evaluate the actual role of Sox8 on gastrulation. Showing simply failed blastopore closure is by far not sufficient.

(B) A thorough reinterpretation of the data and of Sox8 function. There are indeed major issues, as explained below.

The use of Cas7-11 seems to work very well, and will be a nice contribution to the field, although per se the report of this innovation is not sufficient to palliate for the current gaps and inconsistencies in the mechanistic analysis.

Here first some comments, followed by specific suggestions

1) *Issues with the description of patterning during gastrulation: I will try to briefly explain why the description of patterning during gastrulation is confusing, and actually incorrect. There is abundant literature and reviews on the subject.*

a) *Wnt signalling has a dorsalizing activity early on, due to maternal Wnt11/5a activating early canonical b-catenin. But this activity ceases at the onset of gastrulation.*

b) *During gastrulation, two main sources of canonical Wnt activity are: ventral Wnt8, which inhibits in particular the most dorsal axial fate (notochord), and a Wnt (Wnt11?) POSTERIOR activity, located at the late gastrula dorsal lip and contributing to set the AP axis both in the dorsal mesoderm and the prospective neuroderme. Posteriorizing Wnts are inhibited by more anteriorly located Wnt inhibitors (including Kremen).*

However, a number of efficient Xwnt8 antagonists, including secreted Frizzled-related proteins (FRPs) and Dickkopf-1 (Dkk-1) (Hoppler et al. 1996; Finch et al. 1997; Leyns et al. 1997; Mayr et al. 1997; Salic et al. 1997; Glinka et al. 1998; Xu et al. 1998), failed to trigger dorsal development in ventral blastomeres, suggesting that Xwnt8 is not required for the maintenance of ventral fates.

c) *This latter posterior activity is clearly the one repressed by Sox8 on the ventral lip (NOT globally the ventral side!)*

d) *Once gastrulation has started, the most dorsal trunk region (notochord) is formed through INHIBITION of Wnt (zygotic Wnt8), not activation. See abundant literature/reviews about this process.*

e) *In parallel, there is also a non-canonical Wnt11b/Frz7 dorsal activity, but this pathway is implicated in cell migration, not in DV patterning.*

f) *Thus, one is here dealing with something else than simply DV patterning. The model as drawn in figure 7 is simply wrong, starting with the various positions of the gene expressions.*

2) *Issues with LOF phenotype:*

a) *A blastopore closure phenotype is NOT a typical phenotype for defect in DV patterning. It has been known for decades that both dorsalized and ventralized embryos can complete "gastrulation", i.e. the blastopore closes. DV defects are much more about the presence/extent of dorsal and ventral structures.*

b) *One is probably dealing here with a different effect. One possibility is an effect directly on the blastopore closure, which is a specific morphogenetic movement (see elife paper by Ray Keller and colleagues). Actually, the possibility that this process indeed relies on signals located in the lip itself would be quite exciting per se.*

I hope that these brief points will help the authors revisiting their interpretation.

A: We thank the reviewer for the insightful comments as the points described above, in combination with the questions raised in their specific points, have prompted us to design experiments whose results led us to complement and reinterpret our conclusions. In brief, we analysed cell movements and found that they are affected in our treatment. Consequently, axis elongation was also affected as shown by morphological analyses at later stages and *in situ* hybridization against axial and paraxial markers. As predicted by the reviewer, morphological analyses at later stages (tailbud) did not reveal any ventralizing phenotype; instead we observed that about 60% of the embryos displayed mild to dramatic phenotypes (exhibiting blastopore open, shorten anterior-posterior axis and, in the most severe cases, exagastulation). Indeed, BMP loss-of-function experiments does not lead to such strong phenotypes, but mostly to a loss of ventral features at later stages. Additionally, our experiments show that BMP

overexpression cannot rescue *Sox8* knockdown associated defects. Thus, and as stated by the reviewer, these results confirmed that we are dealing with something else than just an impact in DV boundary. In consequence, we have changed our interpretation about the role of *Sox8* in gastrulation and departed from the idea of a requirement for DV axis establishment. Instead, we emphasise the idea that the

observed defects are related to an impact in cell movements and AP patterning. We truly believe that the manuscript is now stronger thanks to their comments. Explanations/answers are provided below in our point-by-point response:

Suggestion for major points to address:

Point 1: *Characterization of the gastrula phenotype: Provide information about the internal structure of the late gastrula embryo.*

A: We acknowledge the reviewer suggestion and proceeded to a more detailed characterization of the internal structure through histological sagittal sections. In this context, the most striking difference observed between control and *sox8* CRISPR was the defective internalization of the yolk plug and an increase in thickness/area of ventral tissues, indicating an accumulation of cells in the ventral region close to the ventral lip (new Fig. 3A). To explore the origin of this phenotype, we performed live imaging of control and *sox8* CRISPR embryos. We started by characterizing the blastopore closure through time, detecting a significant blastopore closure delay in *sox8* CRISPR (new EV1A,B). Next, we tracked the motion of superficial involuting cells in the ventral side (new Fig. 3B-D) and measured cell directionality and velocity. We found that while the velocity is slightly affected, the persistence of cell migration in the ventral lip is drastically reduced upon *sox8* knockdown (new Fig. 3B-D). Together, our results suggest *Sox8* is required for the internalization of ventral tissues. These results are consistent with axis elongation defects observed in Fig. EV1 and disrupted localization of several *ISH* axis elongation markers (see new Fig.3E-M). We added these new figures to the revised version of the manuscript (see Figure 1, Rev02, new Fig. 3 and new Fig.EV1) and described this results in lines 129-142 of the *Results and Discussion* section.

Point 2: *Map by in situ hybridization key markers of the various mesoderm regions (dorsal and ventral, but also anterior and posterior, i.e. prechordal and axial/paraxial).*

A: Following reviewer suggestion, we analysed the expression patterns of several markers associated with distinct mesoderm regions using whole-mount *in situ* hybridization (*ISH*); 1) *chordin* (*chrd*), expressed in the organizer and the axial mesoderm; 2) brachyury (*xbra*) which is expressed in the presumptive mesodermal cells around the blastopore and then in the axial mesoderm (Smith et al, 1991); and 3) myogenic differentiation 1 (*myoD*), which marks the paraxial mesoderm (Della Gaspera et al, 2012; Harvey, 1992). Collectively, our *ISH* results suggest that downregulation of *sox8* and/or *kremen2* lead to pronounced defects in patterning and elongation of dorsal, anterior and posterior mesoderm elongation, without obvious effects on the overall expression levels of these markers (new Figure 3E-M and new Figure EV5A-I). These defects are consistent with the observed impaired morphogenetic movements, strongly involved in midline elongation and in anterior-posterior axis development (Carron & Shi, 2016; Keller & Shook, 2008). We added these results to new figures of the revised manuscript (new Figure 3 and new Figure EV5) and discussed accordingly in the new text (line 138-142 and lines 200-206).

Point 3: *A late morphological phenotypic analysis could also help to evaluate whether there is also an impact on axis elongation. This should be already be visible at late neurula stage, even in embryos that have failed to close their blastopore.*

A: As predicted by the reviewer many embryos failed to close their blastopore, so this analysis was challenging. However, embryos exhibiting milder phenotypes appear shortened, indicating defects in anterior-posterior axis elongation (see new Figure EV1C-E). This was also observed upon *Kremen2* loss-of-function (new Figure EV4). Both Wnt and BMP signaling pathways are known to play roles in axis elongation (Van Itallie et al, 2022;Sander et al, 2007). Thus, the disruption of Wnt and BMP signaling in CRISPR mutants of *sox8* and *kremen2* likely contributes to the observed defects in axis elongation, thereby providing a more likely explanation to the observed phenotypes than our initial assumption about DV related defects.

Point 4: *The analysis of the molecular impact of Sox8 (and Kremen2) LOF is also incomplete: That Kremen2 is a downstream target of Sox8 is well demonstrated, no problem with that.*

- *However, Sox2 and Kremen2 LOF appear to affect Wnt11b EXPRESSION, while Kremen2 is not a repressor, but an inhibitor of Wnt activity. One would expect evidence for an effect on the Wnt signalling pathway activity.*

- *Related: What is exactly the Wnt activity one is talking about? Canonical b-catenin-dependent pathway, PCP pathway, non-canonical? Authors should experimentally address this question, which is here essential in reporting a new "patterning" activity during gastrulation.*

A: *Kremen2 is known to be involved in the canonical Wnt/ β -catenin pathway (Rothbächer & Lemaire, 2002). So to address the referee's points, we performed β -catenin immunofluorescence, using nuclear accumulation of β -catenin as a readout of Wnt activation (Fagotto & Brown, 2009). In control embryos, nuclear β -catenin signal in the ventral mesoderm is low. In contrast, sagittal sections of *sox8* CRISPR and *kremen2* CRISPR embryos revealed increased nuclear β -catenin signal in the ventral mesoderm, suggesting that canonical Wnt signaling is upregulated in these conditions. Despite this increase in Wnt/ β -catenin activity, we cannot exclude the involvement of non-canonical Wnt activity in the phenotype we observed. A new figure with representative confocal images and quantifications of nuclear intensity in ventral cells is now shown in the new version of the manuscript as new Fig.4 I, J and new Fig. 6E,F. Results are discussed in lines 167-171 and 211-213.*

Point 5: *The model proposed suggests that the Sox8-Kremen2-Wnt11b "pathway" controls the dorsal organizer components (Goosecoid, xNot) and on BMP-dependent patterning. Yet there are no experimental data supporting this part of the model, beside RNAseq data. Is the actual DV pattern of BMP activity affected?*

Some minimal experimental results, e.g. P-Smad1 staining for BMP signalling is expected. See above about characterization of the gastrulation phenotype.

A: *We thank the reviewer for this comment. Sox8-Kremen2-Wnt11b is not a pathway and we have amended that along the text. In accordance to a more precise characterisation of the gastrulation phenotype, we have also performed immunofluorescence against P-Smad1 and observed a clear reduction in ventral territories upon Sox8 or Kremen2 knockdown. This new information was added in two new figures (Figure 4K,L and Figure EV6). Also, and as exposed in the introductory paragraph of this letter and explained to reviewer 3 in response to their point #2 and 3, this decrease may not be the driver of the observed phenotypes, as *bmp* mRNA injection did not rescue *sox8* CRISPR related effects. In contrast, *kremen2* mRNA clearly rescued Sox8-related defects (please see Reviewer 3 points 2 and 3). In light of our new results arising from your comments we have removed our model as indeed was not accurate.*

Additional points:

Point 6: None of the transcriptomic data show any indications about variability/reproducibility/confidence: No stats, no SD???

A: We have now added it (Appendix Fig. S2).

Point 7: *Gooseoid* and *Not* mark distinct regions of the mesoderm

A: In light of our new results and to avoid confusion we removed our schematics where *gsc* and *not* were shown in the same territory.

Referee #3:

Reviewer Comments to Author

In this study, the authors identify an unexpected role for Sox8 in establishing dorsal-ventral patterning during gastrulation. This is mediated by a dual role of Sox8 in promoting the expression of BMP-related factors along with a well-known Wnt signaling inhibitor, kremen2. This, in turn, contributes to restrict the expression of Wnt11b to ventral tissues. Given the importance of establishing a proper DV axis in gastrulation, this study addresses an important question in biology. The paper is well-written, the data presented is of high quality and convincing, pointing towards a new function of Sox8 during gastrulation. This work is of excellent quality, however, adding further analyses would, in my view, strengthen the key findings on the mechanism of action of Sox8 and clarify more broadly the mechanisms underlying AP patterning.

A: We sincerely acknowledge the referee for their constructive comments and suggestions, which have significantly improved our manuscript.

Point 1: Do other organizer genes (e.g. *gsc*, *chordin*) show similarly defective localization upon Sox8 or *Kremen2* knockdown?

A: In response to the referee's suggestion, we expanded our analysis to include the expression patterns not only of *chordin* (*chrd*), but also *brachyury* (*xbra*) and the paraxial mesoderm marker *myoD*. We attempted *gsc in situ* hybridisations but the probe did not work in our hands. *In situ* hybridisation analyses revealed that the knockdown of Sox8 and/or *Kremen2* affected the expression pattern of these markers (new Figure 3E-M and new Fig. EV5). Indeed, and as suggested by the reviewer, these defects elicited problems with AP patterning of the axial and paraxial mesoderm. These results, together with new morphological analyses (new Fig. EV1 and Fig. EV4), internal gastrula structure analysis and cell movement tracking (new Fig. 3) have prompted us to refine our conclusions about how Sox8 contribute to gastrulation. We now propose that Sox8 contribute to gastrulation by allowing the expression of *kremen2* which in turn limit the expansion of Wnt-related genes to ventral territories. This favours a

correct blastopore closure and AP patterning at later stages. We included this new data in our revised manuscript (lines 129-142 and lines 200-206).

Point 2: *A more quantitative description of the spatial expansion of dorsal tissues is necessary for both Sox8 and Kremen2 knockdowns and to compare the severity of the phenotypes between Sox8, Kremen2 and the rescue experiments (see my next point).*

A: Thanks for this point. Briefly, we performed additional quantifications of the area of Wnt11b signal across different conditions: Control, *sox8* CRISPR, *kremen2* CRISPR and rescues. Consistent with the conclusions drawn from measuring Wnt11b signal intensity below the equatorial region of embryos, we observed that the area of Wnt11b expression is expanded in both *sox8* CRISPR and *kremen2* CRISPR conditions (see new Figure EV3G, H). As the rescue phenotypes relate also to the point below, our reply to that point is included in Point 3.

Point 3: *It is unclear how much the role of Sox8 in promoting the expression of BMP-related factors is required for its function in DV patterning, given the observations with the kremen2 mRNA rescue. Is the role of Sox8 in regulating BMP signaling therefore a more indirect consequence from regulation of Wnt signaling or rather a direct function, acting in parallel with the regulation of Wnt? To test this, a more quantitative analysis of the dorsal expansion phenotypes observed upon Sox8 and Kremen knockdown as compared to their combined knockdown is necessary.*

A: We analysed the penetrance of Sox8 and Kremen2 knockdowns and we found no significant differences between these treatments. But it is worth mentioning that Sox8 knockdown was often more aggressive and the double knockdown elicited a stronger defect, as shown when analysing *chordin in situ* hybridisations (Fig3.E–M and new FigEV5). In terms of the effect over BMP levels, we performed new experiments that led us to propose that the impact of Sox8 in BMP levels is rather a more indirect consequence from regulation of Wnt signalling. In brief, our observations about the effect of Sox8 knockdown over BMP expression raised the possibility that the role of Sox8 in gastrulation is related to the regulation of BMP signalling. However, co-injection of *bmp* mRNA did not rescue the defects caused by *sox8* CRISPR injection (new Fig. 6H, J; Fig. EV3C, D). On the other hand, co-injection of *sox8* CRISPR 7-11 alongside *kremen2* mRNA was sufficient to rescue both blastopore closure and the *wnt11b* expression pattern (new Fig. 6G-I, M-O and Fig. EV3 C-H). These findings strongly suggest that BMP downregulation may not be the primary cause of the observed defects upon Sox8 knockdown but rather a secondary effect of Wnt signalling modulation via Kremen2 transcriptional control. We have added these new experiments and discussed accordingly in the revised version of the manuscript (lines 217-226 and 258-267).

Point 4: *It is unclear how the defects in blastopore closure are related with those in DV patterning. Is this a delay or blastopore closure fails completely upon Sox8 and Kremen2 knockdown? How specific is this phenotype? Are there also other more specific defects in the morphogenesis of ventral tissues in Sox8 knockdown, for instance in convergence movements.*

A: We appreciate the referee's comment. Following this comment, we performed live imaging of *sox8* knockdowns and evaluated the blastopore closure over time. Our analysis reflected that rather than a delay, the blastopore fails to close in a consistent manner (new Fig. EV1 and EV4B). These defects are rescued by re-introducing *sox8* or *kremen2* mRNAs confirming the specificity of our observations. About 60% of the embryos arrive to tailbud stages with mild to drastic defects (new Fig. EV1C-E). We complemented these experiments with cell tracking of labelled nuclei and observed that convergence

cell movements are affected (new Fig. 3 B-D). This complemented our *in situ* hybridisation analyses showing alterations in AP patterning (point 1 of this reviewer). Thus, we have reassessed our interpretations about the impact of Sox8 in DV patterning. In fact, issues with DV patterning would elicit a loss of dorsal or ventral features at tailbud stages (as it is the case for *ventx1* loss-of-function, please see point 5 below), but instead we observed shortened embryos. These new observations strongly suggest that the observed defects are more related to problems with movements which are required for cell convergence and AP elongation. We have added this new information in our revised version (lines 129-142 and 200-206).

Point 5: *How does the severity of the Sox8 phenotype compare with that of the other well-known ventral determinant Ventx1? This should be discussed in the discussion. Since Sox8 expression is restricted to ventrolateral mesoderm, its function in DV patterning is likely to be more associated with the maintenance rather than establishment of the axis.*

A: This is a very good point that nicely complement point 4. No blastopore closure defects were reported after downregulation of *ventx1/2* (Sander et al, 2007). Depletion of this and other ventral markers often leads to a dorsalisation of embryos at later stages, which is characterised by the lack of ventral features. Sox8 and Kremen2 depletion, instead, displayed a marked failure of blastopore closure and defects in axis elongation at later stages. This was consistent with defective convergence movements and the patterning of axial and mesodermal markers. Thus, and as exposed in point 4, the observed defects upon Sox8 and Kremen2 depletion are more related to AP elongation. This point has been discussed in the revised manuscript (lines 258-267).

Point 6: *An important advance of this manuscript is the use of Crispr Cas7-11 in vivo. As the authors point out, this can be very useful in other systems and I appreciated the detailed explanation of how to implement this method provided in the supplementary materials. This should be further expanded, and the key points should be brought into the main. Can the authors also comment more on off-targets or toxicity associated with the injection of these constructs? Have the authors also tested commercially-synthesized gRNAs and could they comment on this point as well?*

A: We sincerely appreciate this point. To address concerns about the potential toxicity of Cas7-11 mRNA and guide injections, we conducted control experiments. For Cas7-11 mRNA, we evaluated gastrulation (blastopore closure) at the concentration used (800 pg/blastomere; Cas7-11 and guides concentrations can be refined with dose dependent curves, if needed). No significant adverse effects were observed (Figure 2 Rev03). Similarly, we injected only the guides at their standard concentration and found no significant gastrulation defects, as demonstrated in images and quantifications (Figure 2 Rev03). Likewise, these treatments did not affect embryo viability when compared to uninjected controls. In addition to this, we tried commercially-synthesised gRNAs. Similar to our findings with transcribed gRNA guides, we did not observe significant toxicity or death defects. Indeed, phenotypes were achieved with lower concentrations of guides. This is relevant as it can reduce even more potential toxicity defects. It can also help with potential concentration related variability—that can appear from the quantification of transcribed guides in different devices (Nanodrop, Q-bit, etc). Costs can be also reduced and time. We added this point in the revised version (please see Appendix Supplementary Method lines 299-301).

Figure 2 Rev03: Injecting Cas7-11 and guides alone do not impair gastrulation A) Blastopore closure in uninjected and embryos injected with Cas7-11 at a concentration of 800pg/blastomere look similar; B) Quantification of blastopore closure in injected vs uninjected conditions; ns-non significant

7. While the authors provide validation for the *Sox8* knockdown phenotype with a well-validated morpholino and show an effective rescue with mRNA, it would be very valuable to also compare these phenotypes with that of F0 crispants, which have become established in *Xenopus* over recent years. This would provide a first comparison of the 3 different methods and resulting phenotypes in this species.

A: We appreciate the reviewer's suggestion to include a third method for comparison. We agree that the use of F0 crispants in *Xenopus* has indeed become routinely used. However, we would like to emphasize that F0 crispants, designed for genome targeting and generating null alleles, may not serve as a directly comparable tool to CRISPR-Cas7-11 or morpholinos (MOs). While a positive result (recapitulating phenotypes) could be informative, a negative result (no comparable phenotype) may not be conclusive due to potential genomic insertion or compensation mechanisms, such as homolog upregulation, which could mask the phenotype of interest (Salanga & Salanga, 2021). The fundamental difference between Cas9, which modifies DNA, and Cas7-11, which targets RNA, could also contribute to variability in phenotypic outcomes. Cas9-induced genomic alterations impact all subsequent RNA transcripts, whereas Cas7-11 only affects pre-existing RNA molecules. Still, we think that the reviewer comment was important so to strengthen our conclusions, as we employed CRISPR-Cas13 (Kushawah et al, 2020; Hernandez-Huertas), a knockdown approach that has been validated in our lab (Ferreira et al, 2025). Using this method, we successfully downregulated *Sox8* and recapitulated the gastrulation phenotype observed with both *sox8* CRISPR-Cas7-11 and *Sox8* MO (Figure 3 Rev03). Moreover, we recapitulated the phenotypes in neural crest induction (detected by *ISH* for *snail2*) (Figure 4 Rev03). With three well-controlled knockdown approaches yielding consistent results, we believe this provides strong evidence supporting the role of *Sox8* in blastopore closure and embryonic patterning during gastrulation.

Figure 4 Rev03: CRISPR Cas13 - mediated Sox8 knockdown in the ventrolateral mesoderm reproduces CRISPR Cas7-11 results. A) Schematic representation of targeting injections; B) Percentage of embryos displaying a normal ISH signal of *sox8* in Control and CRISPR Cas13 Sox8. Two-sided Fischer's exact test, *** $p = 0.0010$; Error bars: mean with SD; N= 3 independent experiments; C) sqRT-PCR showing depletion of Sox8 in CRISPR 13 condition; D) Representative phenotypes for Control, *sox8* CRISPR 13 and rescue by reintroducing *sox8* mRNA in embryos. B (blastopore); E) Box and whiskers representation of the normalized ratio of blastopore area/embryo area in different conditions. Kruskal–Wallis multiple comparison test, **** $p < 0.0001$; ns- non-significant. N= 3 independent experiments. F) Percentage of embryos exhibiting normal in each condition. Two-sided Fischer's exact test; **** $p < 0.0001$; Error bars: mean with SD. Scale bars: 250 μm .

Figure 5 Rev03: Sox8 CRISPR Cas13 phenocopies *sox8* CRISPR Cas7-11 and *sox8* MO phenotypes. (A-C) Semi-quantitative RT-PCR of stage 21 heads (early neurulation) confirms the depletion of *sox8*. The purple stripes represent target injections in the neural crest. Mann-Whitney test, ** $p=0.0022$; $N=3$ independent experiments. (D) CRISPR Cas13 was used to deplete *sox8* in the neural crest, where its expression was previously characterized. *sox8* CRISPR Cas13 phenocopies previous *sox8* MO phenotypes in the neural crest, such as a delay in neural crest induction (detected by ISH for *snail2*). (E) Quantification of the percentage of embryos with normal signals for *sox8* or *snail2*. Two-sided Chi-square test, *** $p=0.0007$; * $p=0.0409$. Error bars: mean \pm SD; $N=3$ independent experiments ($n=13$ control embryos (ISH *snail2*); $n=9$ *sox8* CRISPR Cas7-11 (ISH *snail2*); $n=24$ control embryos (ISH *sox8*); $n=21$ *sox8* CRISPR (ISH *Sox8*));

References

- Carron C & Shi D-L (2016) Specification of anteroposterior axis by combinatorial signaling during *Xenopus* development. *WIREs Developmental Biology* 5: 150–168
- Della Gaspera B, Armand A-S, Sequeira I, Chesneau A, Mazabraud A, Lécolle S, Charbonnier F & Chanoine C (2012) Myogenic waves and myogenic programs during *Xenopus* embryonic myogenesis. *Developmental Dynamics* 241: 995–1007
- Fagotto F & Brown CM (2009) Detection of Nuclear β -catenin in *Xenopus* Embryos. In *Wnt Signaling*, Vincan E (ed) pp 363–380. Totowa, NJ: Humana Press
- Ferreira F, Moreira S, Zhao M & Barriga EH (2025) Stretch-induced endogenous electric fields drive directed collective cell migration in vivo. *Nat Mater*: 1–9
- Harvey RP (1992) MyoD protein expression in *Xenopus* embryos closely follows a mesoderm induction-dependent amplification of *MyoD* transcription and is synchronous across the future somite axis. *Mechanisms of Development* 37: 141–149
- Hernandez-Huertas L Optimized CRISPR-RfxCas13d system for RNA targeting in zebrafish embryos. *OPEN ACCESS*: 32
- Keller R & Shook D (2008) Dynamic determinations: patterning the cell behaviours that close the amphibian blastopore. *Phil Trans R Soc B* 363: 1317–1332
- Kushawah G, Hernandez-Huertas L, Abugattas-Nuñez del Prado J, Martinez-Morales JR, DeVore ML, Hassan H, Moreno-Sanchez I, Tomas-Gallardo L, Diaz-Moscoso A, Monges DE, *et al* (2020) CRISPR-Cas13d Induces Efficient mRNA Knockdown in Animal Embryos. *Developmental Cell* 54: 805-817.e7
- Rothbächer U & Lemaire P (2002) Crème de la Kremen of Wnt signalling inhibition. *Nat Cell Biol* 4: E172–E172
- Salanga CM & Salanga MC (2021) Genotype to Phenotype: CRISPR Gene Editing Reveals Genetic Compensation as a Mechanism for Phenotypic Disjunction of Morphants and Mutants. *Int J Mol Sci* 22: 3472
- Sander V, Reversade B & De Robertis EM (2007) The opposing homeobox genes Goosecoid and Vent1/2 self-regulate *Xenopus* patterning. *Development* 134: 2955–2965
- Smith JC, Price BM, Green JB, Weigel D & Herrmann BG (1991) Expression of a *Xenopus* homolog of Brachyury (T) is an immediate-early response to mesoderm induction. *Cell* 67: 79–87
- Van Itallie ES, Field CM, Mitchison TJ & Kirschner MW (2022) Wnt11 family dependent morphogenesis during frog gastrulation is marked by the cleavage furrow protein anillin *Developmental Biology*

Dear Dr. Barriga,

Thank you for the submission of your revised manuscript. I have taken over its handling since my colleague Martina is currently out of office. We have now received the enclosed reports from the referees and I am happy to say that all support its publication. Referees 2 and 3 still have a few more minor suggestions that I would like you to address in the ms text before we can proceed with the official acceptance of your paper.

A few editorial requests will also need to be addressed:

- Your ms has 6 main figures and will therefore be published as a full article with separate results and discussion sections. Please separate both sections. Please also remove the "Outlook" subheading, which is not supported.
- Please remove the figures from the ms file; only the figure legends need to be included at the end of the ms file.
- Please correct the conflict of interest subheading to "Disclosure and Competing Interests Statement"
- Please remove the authors credits from the ms file. All credits need to be entered during online ms submission.
- There is discrepancy between the FUNDING INFO in our online submission system and the ms file. Please check carefully and correct. All funding info must be present in both places. (Cluster of Excellence Physics of Life of TU Dresden, Instituto Gulbenkian de Ciência (IGC) and Fundação Calouste Gulbenkian (FCG), start-up grant I-411133.01, Gulbenkian Institute for Molecular Medicine (GIMM); this one might be a discrepancy or the number is missing in eJP: 94978)
- "Supplementary" is not a correct callout; and callouts for Appendix Figures S4-6 are missing, please add and correct.
- The APPENDIX FILE needs to be submitted as PDF and should not have Methods even if they are supplementary (we don't use "Supplementary" anymore); the table of content needs to have each item listed with its page number; Appendix figures are also provided as separate files and these need to be removed.
- During our routine check of ms images 2 possible image reuses were detected that are not listed in the figure legends: between Figure 2E and Figure EV3A; and between Figure 4K and Figure EV6C. Please clarify and correct.

* Figure Legends - Comments *

- Please note that the legends for figures 3, EV5 is not provided in the sequential manner. This needs to be rectified.
- Please note that the exact p values are not provided in the legends of figures 2D, F, G; 3D; 4J, L; 5B, F, H; 6B, C, F; EV1 B, D, E; EV2 B, C, F, H; EV3 B, C, D, E, F, G, H; EV5 B, E, H; EV6 D. Please provide exact p-values as reasonable.
- Please note that the box plots need to be defined in terms of minima, maxima, centre, bounds of box and whiskers, and percentile in the legends of figures 2F; 3A, C, D; 4J, L; 5F, H; 6B, F; EV2 B, EV3 C, E, G.
- Please note that the error bars are not defined in the legends of figures 1D; EV2 C, F; EV3 B, F, H; EV5 B, E, H. "
- Please note that the scale bar is missing for figures 6H-Q
- Please note that the dotted borders are not defined in the legend of figures 6G-Q; EV1 A, C; EV5 A, D, G; EV6 A . This needs to be rectified.

EMBO press papers are accompanied online by A) a short (1-2 sentences) summary of the findings and their significance, B) 2-3 bullet points highlighting key results and C) a synopsis image that is exactly 550 pixels wide and 200-600 pixels high (the height is variable). The synopsis image should provide a sketch of the major findings, like a graphical abstract. Please note that text needs to be readable at the final size. Please send us this information along with the final manuscript.

Referee #1:

The authors have made extensive modifications to this manuscript, including important new experimental data, that suitably address the previous reviews.

This work is of high impact and includes convincing sample sizes, controls and statistical analyses, giving high confidence in the interpretations.

Referee #2:

The revised manuscript has been significantly modified. It includes several key new data that are clean and consistent, and support a much more convincing model. The identification of Sox8 as a new transcriptional regulator acting on the ventral side of the embryo is an important discovery, as for years most of the research has focused on the dorsal part of the gene network, which provided a blatantly incomplete picture. The data of the present study illustrate the importance of the ventral contribution in proper patterning and morphogenesis of the gastrula embryo. The changes in the distribution of key markers in response to Sox8 depletion are striking, and the functional linking between Sox8 and the inhibitor Kremen2 fully supported. The dramatic effect of this ventral gene on the general process of gastrulation is very intriguing, and this study should impel renewed interest in digging further in this so far largely ignored aspect, not only in amphibians, but in all vertebrate embryos. Altogether, the authors have done a great job, and this study should be of high interest for the Developmental Biology community.

One last small issue, which I raised in the first review but did not fully come through: The study unambiguously shows a strong impact of modulating Sox8 and Kremen2 on the expression of Wnt11b. The issue is that the authors amalgamate this Wnt11b with the early maternal Wnt11b locate dorsally upon cortical rotation, and responsible for setting the initial dorsalizing center. But this maternal component is an mRNA is irreversibly located in the dorsal cells during early cleavage. Since they show here differences in gene EXPRESSION, they CANNOT be dealing with MATERNAL Wnt11b, but they are necessarily looking at ZYGOTIC Wnt11b. Zygotic Wnt11b expression is well established, and although initially presented as triggering the non-canonical pathway, it can certainly also activate β -catenin signalling, as now supported by the new data of the manuscript. I would urge the author to do this small correction, as currently presented, this aspect will add unnecessary confusion to the field.

Referee #3:

I appreciate the additional experiments, analyses and clarifications provided by Moreira, et al in response to the referee comments. The revised manuscript now includes a revised model for how Sox8 functions in gastrulation. While the data provided for BMP overexpression in the Sox8 downregulation background is not, in my view, entirely conclusive, with a closer comparison of BMP signaling in the ventral domain upon modulation of canonical versus non-canonical Wnt signaling still missing. This could, for example, clarify the phenotype difference between the double Sox8/Kremen2 knockdowns versus the singles or help test whether the complex phenotypes observed upon Sox8 downregulation could reflect a dual role in regulating both canonical and non-canonical Wnt signaling.

Nonetheless, the authors strengthened their previous data supporting the role of Sox8 in regulating Wnt signaling by: i) showing enhanced nuclear β -catenin in ventral tissues following Sox8 or Kremen2 downregulation; and ii) more detailed analyses of the phenotypes upon Sox8 or Kremen2 knockdowns, which now point more towards a role in AP patterning and morphogenesis. Since it is still not clear what is the primary morphogenetic defect upon the Sox8 or Kremen2 downregulations, a more focused discussion of these phenotypes (and respective rescues) in light of the literature and proposed model would be valuable in my view. For instance, how connected are the different morphogenetic defects observed - can the defects in blastopore closure per se result in a shortened AP axis or, vice-versa? Or instead these phenotypes are likely to be independent defects, pointing to a morphogenetic function of Sox8 in regulating AP axis extension, as well as a separate function in controlling blastopore closure.

Overall, the paper is of high-quality, uncovers a novel role for Sox8 in gastrulation and the authors have substantially improved the manuscript, with additional technical details with the CRISPR7-11 system and validations with CRISPR13 being very welcomed/useful for a wide community.

Dear Dr. Barriga,

Thank you for the submission of your revised manuscript. I have taken over its handling since my colleague Martina is currently out of office. We have now received the enclosed reports from the referees and I am happy to say that all support its publication. Referees 2 and 3 still have a few more minor suggestions that I would like you to address in the ms text before we can proceed with the official acceptance of your paper.

A **few editorial requests** will also need to be addressed:

- Your ms has 6 main figures and will therefore be published as a full article with separate results and discussion sections. Please separate both sections. Please also remove the "Outlook" subheading, which is not supported.

A: Done.

- Please remove the figures from the ms file; only the figure legends need to be included at the end of the ms file.

A: Done.

- Please correct the conflict of interest subheading to "Disclosure and Competing Interests Statement"

A: We have now corrected.

- Please remove the authors credits from the ms file. All credits need to be entered during online ms submission.

A: Done as requested.

- There is discrepancy between the FUNDING INFO in our online submission system and the ms file. Please check carefully and correct. All funding info must be present in both places. (Cluster of Excellence Physics of Life of TU Dresden, Instituto Gulbenkian de Ciência (IGC) and Fundação Calouste Gulbenkian (FCG), start-up grant I-411133.01, Gulbenkian Institute for Molecular Medicine (GIMM); this one might be a discrepancy or the number is missing in eJP: 94978)

A: Done as requested.

- "Supplementary" is not a correct callout; and callouts for Appendix Figures S4-6 are missing, please add and correct.

A: Corrected.

- The APPENDIX FILE needs to be submitted as PDF and should not have Methods even if they are supplementary (we don't use "Supplementary" anymore); the table of content needs to have each item listed with its page number; Appendix figures are also provided as separate files and these need to be removed.

A: We moved the Methods section from the Appendix file to the main manuscript and shortened the content to fit the new format. All Appendix Figures were removed, with only their legends retained in the appendix file. The table of contents was updated accordingly.

Note that the protocol for CRISPR 7-11 that we wanted to make available to the community with tips and troubleshooting is now gone, after these changes. Would it be possible to upload it as a separate method article or is there any other option that you consider appropriate to deliver this information to the public?

- During our routine check of ms images 2 possible image reuses were detected that are not listed in the figure legends: between Figure 2E and Figure EV3A; and between Figure 4K and Figure EV6C. Please clarify and correct.

A: Embryo from Figure EV3A was replaced to avoid redundancy with Figure 2E. Moreover, control Figure EV6C was replaced to avoid redundancy with Figure 4K (control).

* Figure Legends - Comments*

- Please note that the legends for figures 3, EV5 is not provided in the sequential manner. This needs to be rectified.

A: We have now corrected it.

- Please note that the exact p values are not provided in the legends of figures 2D, F, G; 3D; 4J, L; 5B, F, H; 6B, C, F; EV1 B, D, E; EV2 B, C, F, H; EV3 B, C, D, E, F, G, H; EV5 B, E, H; EV6 D. Please provide exact p-values as reasonable.

A: Exact *P* values were now added. In cases where $P < 0.0001$, Graphpad Prism does not report exact values below 0.0001.

- Please note that the box plots need to be defined in terms of minima, maxima, centre, bounds of box and whiskers, and percentile in the legends of figures 2F; 3A, C, D; 4J, L; 5F, H; 6B, F; EV2 B, EV3 C, E, G.

A: We provided this information now.

- Please note that the error bars are not defined in the legends of figures 1D; EV2 C, F; EV3 B, F, H; EV5 B, E, H. "

A: We provided this information now.

- Please note that the scale bar is missing for figures 6H-Q

A: The scale bar in these figures is the same as the first image, so we added the information of the scale bar for the figures G–L.

- Please note that the dotted borders are not defined in the legend of figures 6G-Q; EV1 A, C; EV5 A, D, G; EV6 A. This needs to be rectified.

A: We provided this information now.

EMBO press papers are accompanied online by A) a short (1-2 sentences) summary of the findings and their significance, B) 2-3 bullet points highlighting key results and C) a synopsis image that is exactly 550 pixels wide and 200-600 pixels high (the height is variable). The synopsis image should provide a sketch of the major findings, like a graphical abstract. Please note that text needs to be readable at the final size. Please send us this information along with the final manuscript.

Please see separate file One sentence summary and Bullet points (Related Manuscript file).

Referee #1:

The authors have made extensive modifications to this manuscript, including important new experimental data, that suitably address the previous reviews. This work is of high impact and includes convincing sample sizes, controls and statistical analyses, giving high confidence in the interpretations.

A: We thank the reviewer for their insightful comments, which have greatly strengthened and improved our work.

Referee #2:

The revised manuscript has been significantly modified. It includes several key new data that are clean and consistent, and support a much more convincing model. The identification of Sox8 as a new transcriptional regulator acting on the ventral side of the embryo is an important discovery, as for years most of the research has focused on the dorsal part of the gene network,

which provided a blatantly incomplete picture. The data of the present study illustrate the importance of the ventral contribution in proper patterning and morphogenesis of the gastrula embryo. The changes in the distribution of key markers in response to Sox8 depletion are striking, and the functional linking between Sox8 and the inhibitor Kremen2 fully supported. The dramatic effect of this ventral gene on the general process of gastrulation is very intriguing, and this study should impel renewed interest in digging further in this so far largely ignored aspect, not only in amphibians, but in all vertebrate embryos. Altogether, the authors have done a great job, and this study should be of high interest for the Developmental Biology community. One last small issue, which I raised in the first review but did not fully come through: The study unambiguously shows a strong impact of modulating Sox8 and Kremen2 on the expression of Wnt11b. The issue is that the authors amalgamate this Wnt11b with the early maternal Wnt11b locate dorsally upon cortical rotation, and responsible for setting the initial dorsalizing center. But this maternal component is an mRNA is irreversibly located in the dorsal cells during early cleavage. Since they show here differences in gene EXPRESSION, they CANNOT be dealing with MATERNAL Wnt11b, but they are necessarily looking at ZYGOTIC Wnt11b. Zygotic Wnt11b expression is well established, and although initially presented as triggering the non-canonical pathway, it can certainly also activate β -catenin signalling, as now supported by the new data of the manuscript. I would urge the author to do this small correction, as currently presented, this aspect will add unnecessary confusion to the field.

A: We thank the reviewer for this valuable comment and apologize for not having fully addressed this point in the previous version (please see line 165).

Referee #3:

I appreciate the additional experiments, analyses and clarifications provided by Moreira, et al in response to the referee comments. The revised manuscript now includes a revised model for how Sox8 functions in gastrulation. While the data provided for BMP overexpression in the Sox8 downregulation background is not, in my view, entirely conclusive, with a closer comparison of BMP signaling in the ventral domain upon modulation of canonical versus non-canonical Wnt signaling still missing. This could, for example, clarify the phenotype difference between the double Sox8/Kremen2 knockdowns versus the singles or help test whether the complex phenotypes observed upon Sox8 downregulation could reflect a dual role in regulating both canonical and non-canonical Wnt signaling.

A: We fully agree with the reviewer that, beyond canonical Wnt signaling, non-canonical Wnt signaling may also contribute to the phenotype observed upon *sox8* knockdown. Because the potential contribution of non-canonical Wnt signaling remains highly relevant, we have added a paragraph in the Discussion to explicitly address this possibility (lines 246).

Nonetheless, the authors strengthened their previous data supporting the role of Sox8 in regulating Wnt signaling by: i) showing enhanced nuclear B-catenin in ventral tissues following Sox8 or Kremen2 downregulation; and ii) more detailed analyses of the phenotypes upon Sox8 or Kremen2 knockdowns, which now point more towards a role in AP patterning and morphogenesis. Since it is still not clear what is the primary morphogenetic defect upon the

Sox8 or Kremen2 downregulations, a more focused discussion of these phenotypes (and respective rescues) in light of the literature and proposed model would be valuable in my view. For instance, how connected are the different morphogenetic defects observed - can the defects in blastopore closure per se result in a shortened AP axis or, vice-versa? Or instead these phenotypes are likely to be independent defects, pointing to a morphogenetic function of Sox8 in regulating AP axis extension, as well as a separate function in controlling blastopore closure.

A: We thank the reviewer for this insightful comment and have incorporated the suggested aspects into the Discussion section (lines 274)

Overall, the paper is of high-quality, uncovers a novel role for Sox8 in gastrulation and the authors have substantially improved the manuscript, with additional technical details with the CRISPR7-11 system and validations with CRISPR13 being very welcomed/useful for a wide community.

Dr. Elias Barriga
TU Dresden
Cluster of Excellence Physics of Life (PoL),
ArnoldStr 18
Dresden, Saxony
Germany

Dear Elias,

I am very pleased to accept your manuscript for publication in the next available issue of EMBO reports. Thank you for your contribution to our journal.

Kind regards,

Martina
